# The CIP2A-TOPBP1 axis facilitates mitotic DNA repair via MiDAS and MMEJ

Peter R. Martin [1] ✉, Jadwiga Nieminuszczy [1], Zuza Kozik[1], Nihal Jakub[1], Szymon Kowalski[1,2], Maxime Lecot [3], Julia Vorhauser[1], Karen A. Lane [1], Alexandra Kanellou[1], Jörg Mansfeld [1], Laurence H. Pearl[4,5], Antony W. Oliver [5], Jessica A. Downs [1], Jyoti S. Choudhary [1], Matthew Day [4,6] & Wojciech Niedzwiedz [1] ✉

Mitotic DNA double-strand breaks (DSBs) accumulate in response to replication stress or BRCA1/2 deficiency posing a significant threat to genome stability as repair by non-homologous end-joining (NHEJ) and homologous recombination (HR) is largely inactivated in mitosis. Instead, mitotic cells rely on alternative repair processes such as microhomology-mediated end-joining (MMEJ) and mitotic DNA synthesis (MiDAS). How these mitotic DNA repair pathways are functionally regulated remains unclear. Here we reveal that the CIP2A-TOPBP1 complex plays an essential regulatory role by facilitating the mitotic recruitment of both SMX complex components and Polθ to mitotic chromatin. Recruitment of the SMX complex components is driven by CDK1-dependent phosphorylation of SLX4 at Thr1260, enabling its interaction with TOPBP1 BRCT domains 1/2, thereby promoting MiDAS. Concurrently, CIP2A promotes efficient mitotic localisation of Polθ to facilitate MMEJ. The simultaneous functional disruption of both MiDAS and MMEJ pathways upon CIP2A loss provides rationale for the synthetic lethality observed in BRCA1 or 2-deficient cells. These findings position the CIP2A-TOPBP1 axis as a central regulatory hub for mitotic DNA repair, highlighting therapeutic opportunities in tumours characterised by HR deficiency or elevated replication stress.

DNA double-strand breaks (DSBs) in mitotic cells may persist from interphase due to replication stress or may occur directly in mitosis[1]. If left unrepaired, these DSBs lead to gross chromosome breakage, genomic rearrangements, and polyploidy, all hallmarks of cancer[2–5]. Established anti-cancer therapeutic approaches exploit this by induction of replication-associated DNA damage, driving cells with unrepaired DNA into mitosis, promoting mitotic catastrophe and cell death[1,6–9]. However, clinical resistance to these therapeutic modalities is a growing issue and underscores the gap in our mechanistic understanding of DNA repair outside of interphase.

Cells deficient in homologous recombination (HR), experience heightened replication stress, and are particularly prone to the accumulation of mitotic DNA DSBs. Recent studies have identified the role of DNA polymerase theta (Polθ) mediated microhomology-mediated end joining (MMEJ) and mitotic DNA synthesis (MiDAS) as critical repair pathways in mitosis, functioning redundantly to HR[5,10–12]. These pathways present a targetable vulnerability in tumours that exhibit heightened DNA replication stress, including those that are HR deficient[1]. Despite the potential to target the mechanisms of DSB repair in mitosis, they remain inadequately defined, limiting our

[1]Division of Cell and Molecular Biology, The Institute of Cancer Research, London, UK. [2]Department of Pharmacology, Faculty of Medicine, Medical University of Gdańsk, Gdańsk, Poland. [3]Faculty of Medicine of Rennes, University of Rennes, Rennes, France. [4]Division of Structural Biology, The Institute of Cancer Research, London, UK. [5]Genome Damage and Stability Centre, University of Sussex, Brighton, UK. [6]Centre of Molecular Cell Biology, Queen Mary University of London, London, UK. ✉e-mail: peter.martin@icr.ac.uk; wojciech.niedzwiedz@icr.ac.uk

ability to fully exploit these mitotic mechanisms for anti-cancer therapy[1,6–8,13–17].

The SMX (SLX1-SLX4-XPF-ERCC1-MUS81-EME1) tri-nuclease complex is essential for genome stability and is activated in a phosphorylation-dependent manner at the G2/M boundary[18]. It plays an evolutionary conserved role in resolving replication and recombination intermediates and facilitating MiDAS to ensure proper chromosomal segregation[11,12,18,19]. SLX4 serves as a central scaffold coordinating the assembly and activity of SMX components at DNA damage sites[18,20]. Mutations in *SLX4* (*FANCP*) cause Fanconi anaemia, a disorder characterised by developmental abnormalities, anaemia, and increased cancer risk[21–23]. Its function is tightly regulated by kinases to prevent premature activation, as uncontrolled activity—such as Cyclindependent kinase 1 (CDK1) de-repression by Wee1-like protein kinase (WEE1) inhibition—promotes aberrant cleavage of replication forks, leading to chromosomal damage[20]. Together, this underscores the importance of the SMX complex and its regulation in the maintenance of genome stability.

TOPBP1, is an essential adaptor protein, with key roles in the maintenance of genome stability, largely attributed to its nine BRCA1 C-terminal (BRCT) domains, which facilitate phosphorylationdependent protein-protein interaction throughout the cell cycle, including in mitosis[24–27]. TOPBP1's functions are closely regulated by CDK1 and Polo-like kinase 1 (PLK1) -dependent phosphorylation events[5,28–31]. Notably, TOPBP1 interacts with DNA polymerase theta (Polθ), facilitating MMEJ-mediated DSB repair during mitosis[10]. TOPBP1 also plays a role in regulating the BTR (BLM-TOP3a-RMI1/2) dissolvase complex, supporting chromosomal segregation through phosphorylation events driven by CDK1 and PLK1[30]. Studies have also shown that the interaction between yeast orthologs of TOPBP1 (Dbp11) and SLX4 (Slx4) is mediated by CDK1-driven phosphorylation[28], a mechanism that is conserved in human cells, where the interaction is facilitated through SLX4 phosphorylation at Thr1260[28,29]. However, the role of the TOPBP1-SLX4 interaction in mitotic repair, especially in the context of HR deficiency, remains undefined.

Recently, a mitotic-specific DNA repair complex consisting of MDC1, TOPBP1 and CIP2A has been identified, which functions to tether broken chromosomes to prevent fragmentation and mitotic catastrophe[32,33]. This mechanism is independent of Polθ, POLD3, or the MRN (MRE11-NBN-RAD50) complex components, although their loss increases mitotic DNA damage[32]. CIP2A and TOPBP1, interact directly and exhibit mutual dependence for chromatin recruitment during mitosis[13]. Recent studies have underscored CIP2A and its interaction with TOPBP1 as a highly penetrant mitotic-specific synthetic lethal (SL) target in BRCA1/2 deficient cells[1,13]. The mechanistic underpinning of this SL interaction is unknown; although, it has been suggested that CIP2A's role in the mitotic DNA tethering complex may be a contributing factor[8,32,33]. However, this explanation is challenged by the observation that MDC1 loss, another key DNA tethering factor, does not induce the same SL in BRCA1/2 deficiency[34,35]. An alternative hypothesis is that CIP2A is a key regulator of yet undefined mitotic DNA repair. However, although the interaction between TOPBP1 and Polθ is essential for MMEJ-dependent DSB repair in mitosis[5], CIP2A appears to be dispensable for MMEJ-driven telomere fusions, even though it is critical in regulating TOPBP1 recruitment to chromatin[1,13,36,37]. This suggests that other targetable factors/mechanisms may drive the synthetic lethality between CIP2A-TOPBP1 and BRCA1/2 deficiency. Given the limited understanding of mitotic DSB repair mechanisms and the ongoing clinical trials targeting Polθ, defining the precise role of the CIP2A-TOPBP1 axis in DNA repair is critical for improving the treatment of cancers.

To address this, we conducted a comprehensive, unbiased Co-IP/MS analysis that mapped the TOPBP1 interactome across different cell cycle phases. Importantly, we identify key interactions in mitosis with SMX complex components (SLX4, ERCC1, XPF, EME1, MUS81), CIP2A, MDC1, and PLK1. We determine that TOPBP1 is required for the

recruitment of SMX component recruitment in BRCA1/2 deficiency. We identify CDK1-dependent phosphorylation of SLX4 at Thr1260 as critical for interaction with BRCT1 and 2 of TOPBP1. This interaction functions to recruit SLX4, MUS81, and ERCC1 to mitotic chromatin marked by the CIP2A-TOPBP1 complex in response to replication stress. Cells with an endogenous SLX4 Thr1260Ala mutation show defective MiDAS, leading to increased genome instability, indicated by elevated micronuclei, γH2AX and 53BP1 levels.

Critically, our study uncovers a genetic interaction network that defines the CIP2A-TOPBP1 complex as a regulatory hub that facilitates mitotic DNA repair via MiDAS and MMEJ. The CIP2A-TOPBP1 complex achieves this by recruiting not only components of the SMX complex (SLX4, MUS81 and ERCC1) but also Polθ to mitotic chromatin. Concurrently, we demonstrate that CIP2A loss impairs both break-induced replication (BIR)-like /MiDAS and MMEJ. Pharmacological inhibition of Polθ, combined with loss of the TOPBP1-SLX4 interaction, exacerbates genome instability and reduces cellular proliferation under replication stress. Notably, SLX4, Polθ, and CIP2A are essential for cellular proliferation in BRCA1/2-deficient cells and expression of a minimal SLX4 fragment containing Thr1260 is also sufficient to impair proliferation in BRCA1/2 deficient cells. This study subsequently provides a mechanistic framework that underpins the SL observed upon CIP2A-TOPBP1 loss in cells with BRCA1/2 mutations. Consequently, these findings further emphasise the mitotic CIP2A-TOPBP1 axis as a promising therapeutic target in BRCA1/2-deficient cancers and those with elevated DNA replication stress, while highlighting actionable avenues for anti-cancer therapy development.

## Results

### Cell cycle-specific proteomics reveals TOPBP1 interactors in mitosis

TOPBP1 is overexpressed in multiple cancers, including breast cancer, and its overexpression in this setting is associated with an aggressive phenotype[38–40]. Moreover, single-nucleotide polymorphisms (SNPs) in *TopBP1* that promote higher protein expression have been associated with an increased risk of breast and endometrial cancers[41,42]. These tumours are burdened with *Brca1/2* mutations, resulting in significant homologous recombination deficiency (HRD)[43–45]. With growing interest in exploiting mitotic DSB repair as a therapeutic vulnerability in HR deficient tumours, we hypothesised that the function of TOPBP1 in mediating protein-protein interactions throughout the cell cycle may be especially critical during mitosis[1,5,10]. To explore this, we conducted an unbiased, cell-cycle stage-specific, co-immunoprecipitation LC-MS analysis to identify proteins uniquely enriched during S-phase and mitosis (Fig. 1a–c and Supplementary Fig. 1a). This approach allowed us to map the mitotic TOPBP1 interactome, revealing proteins preferentially enriched in M-phase compared to asynchronous or S-phase cells (Fig. 1b and c), including previously reported factors PLK1, MDC1, and CIP2A[1,13,37,46]. Notably, we identified elevated interaction between TOPBP1 and members of the SMX complex in mitosis. These findings reveal changes in TOPBP1 complexes throughout the cell-cycle and provide insights into its roles throughout the cell-cycle.

To further validate and explore the functional implications of the TOPBP1 interactome, we performed an additional quantitative Tandem Mass Tag (TMT) mass-spectrometry analysis by isobaric labelling of TOPBP1 Co-IPs from asynchronous and M phase synchronised cells (Fig. 1d). This analysis highlighted a subset of proteins significantly enriched in mitosis, including key DNA repair proteins, such as PLK1, CIP2A, RHNO1, EME1, SLX4, ERCC4 and SLX4IP, supporting the notion that TOPBP1 plays a crucial role in orchestration of a network of DNA repair complexes specifically during mitosis (Fig. 1d).

To gain understanding of the biological processes associated with the TOPBP1 interactome, we conducted a gene ontology (GO) enrichment analysis on the identified proteins in our TMT analysis (Fig. 1e). The analysis of mitotic interactions revealed a significant enrichment for processes related to DNA repair, mitotic cell cycle, chromosome

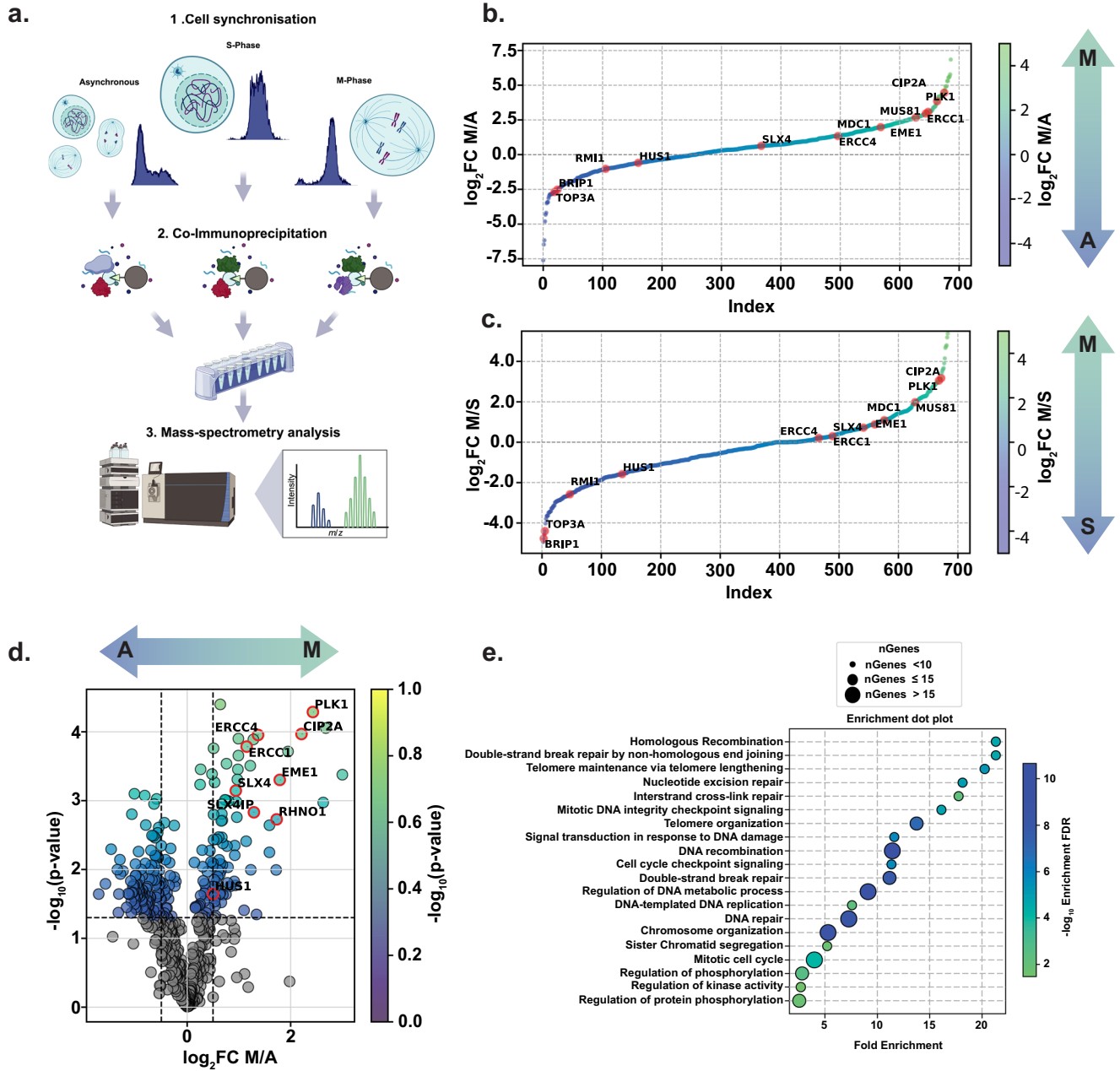

**Fig. 1 | Enrichment of TOPBP1 interactions with SMX complex components during mitosis. a** Schematic representation of endogenous TOPBP1 Co-Immunoprecipitation (Co-IP) experiments across different cell cycle phases from HEK293TN cells. Created in BioRender. Martin, P. (2025) https://BioRender.com/n14w633. **b** Dot plot illustrating mean $\log_2$ fold change (FC) of proteins detected in TOPBP1 Co-IP samples from M phase synchronised 'M' versus asynchronous 'A' cells, by label-free mass-spectrometry. **c** similar analysis as in (**b**), comparing TOPBP1 Co-IP samples from M phase 'M' synchronised cells and S phase 'S' synchronised cells. Samples analysed in (**a–c**) were from three independent experiments. **d** Volcano plot of TMT quantitative mass spectrometry analysis of TOPBP1 Co-IP samples, showing mean $\log_2$ fold change and the statistically significant enrichment of interactions in asynchronous 'A' versus M phase 'M' cells. Samples were from three independent experiments and statistical significance was determined by two tailed t-test. **e** Gene Ontology (GO)-term enrichment analysis of the mitotic TOPBP1 interactome showing -$\log_{10}$ enrichment false discovery rate (FDR) and fold enrichment. Source data are provided as a Source Data file.

organisation, and protein phosphorylation. Notably, pathways, such as DSB repair, HR and non-homologous end joining were among the enriched categories, supporting a role of TOPBP1 in promoting aspects of mitotic DNA repair and maintenance of genome stability (Fig. 1e). To gain phenotypic insight into the role of TOPBP1 in supporting genome stability in mitosis, we employed a TOPBP1-degron system[47]. HCT116 cells expressing osTIR1 that were CRISPR edited at the endogenous locus to express TOPBP1-mAID-Clover were exposed to IAA to elicit acute degradation of TOPBP1. Consistent with the crucial role of TOPBP1 in mitosis, its acute temporal depletion in G2/M, followed by release into mitosis, led to a marked increase in anaphase abnormalities, including DAPI bridges, DNA laggards, and PICH-marked ultra-fine anaphase bridges (UFBs) (Supplementary Fig. 1b–d). Taken together, we hypothesize that TOPBP1 may play an essential role in mitotic genome stability through the identified mitotic protein-protein interactions.

Notably, in our mass-spectrometry interactome analysis, we observed significant enrichment of the majority of the components of the SMX complex (ERCC4/XPF, ERCC1, EME1, MUS81 and SLX4 (Fig. 1b–d)) indicating a functional role of the TOPBP1-SMX complex interaction that may maintain genome integrity. Therefore, we sought

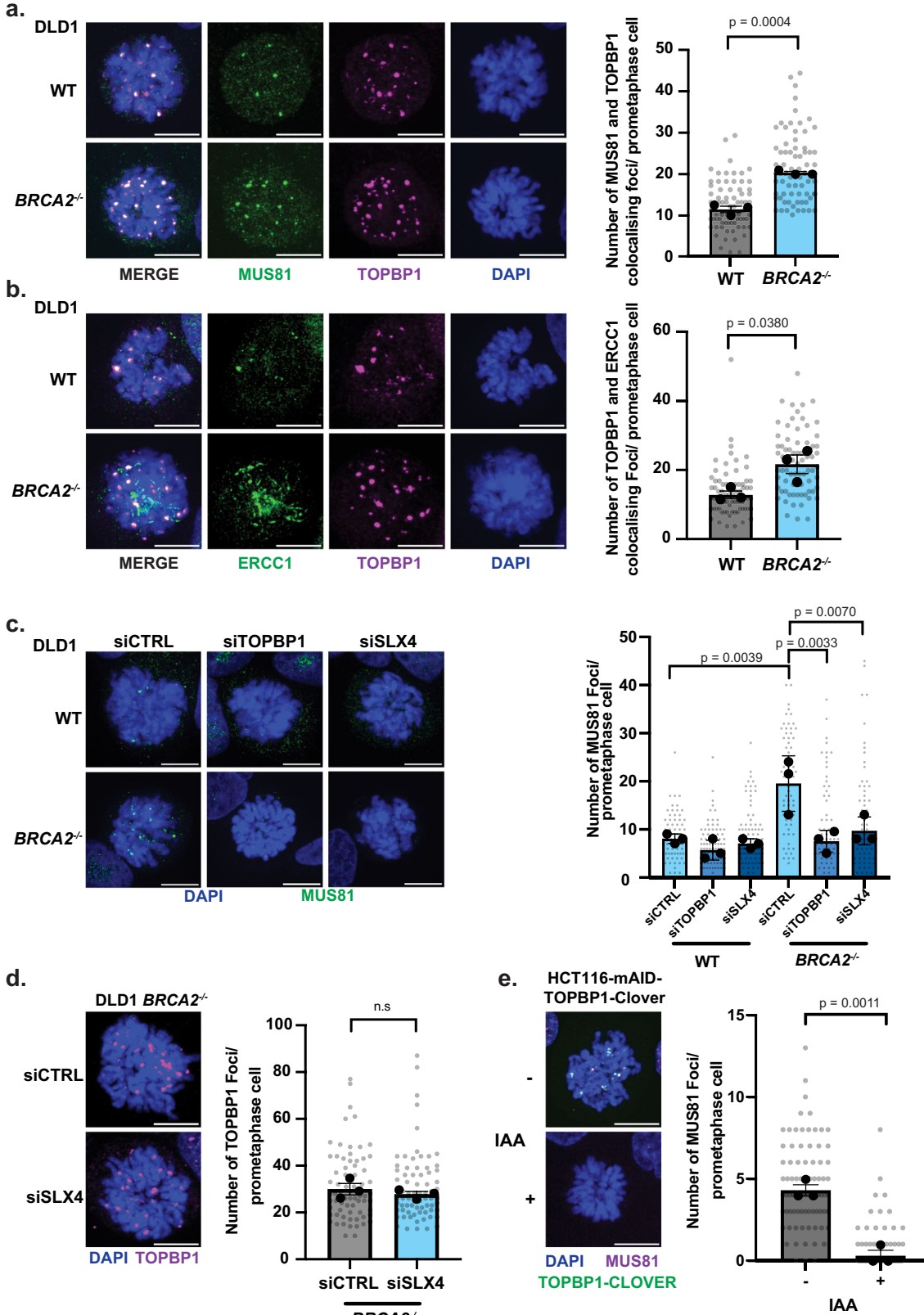

to validate these interactions by eGFP-TOPBP1 co-immunoprecipitation from human cell extracts and found mitotic enrichment of endogenous components of the SMX complex (SLX4, MUS81 and ERCC1 (Supplementary Fig. 1e; lysates were treated with benzonase to prevent DNA bridging)). Thus, we conclude that TOPBP1 and its interaction with components of the SMX complex is elevated in mitosis.

## TOPBP1 co-localises with SMX components and is required for SLX4-MUS81 recruitment in mitosis

BRCA1 and 2 loss is associated with increased DNA replication stress and the accumulation of unresolved replication intermediates in mitosis[1]. To determine whether BRCA2-deficient cells demonstrate reliance on TOPBP1-associated functions we analysed TOPBP1

**Fig. 2 | Elevation of mitotic TOPBP1 dependent recruitment of SMX complex components in BRCA2 deficiency. a** Representative images and a dot plot showing the number of MUS81 and TOPBP1 colocalising foci in DLD1 and DLD1 *BRCA2^-/-^* prometaphase cells (WT: *n* = 74 and *BRCA2^-/-^*: *n* = 73, from three independent experiments). **b** Representative images and a dot plot showing the number of ERCC1 and TOPBP1 colocalising foci in DLD1 and DLD1 *BRCA2^-/-^* cells (WT: *n* = 73, *BRCA2^-/-^*: *n* = 74, from three independent experiments). **c** Representative images and a dot plot showing the number of MUS81 localisation in DLD1 WT and DLD1 *BRCA2^-/-^* prometaphase cells, treated with siCTRL, siTOPBP1 or siSLX4 (DLD1 (siCTRL: *n* = 75, siTOPBP1: *n* = 82, siSLX4: *n* = 86) and DLD1 *BRCA2^-/-^* (siCTRL: *n* = 65, siTOPBP1: *n* = 69, siSLX4: *n* = 64) from three independent experiments).

**d** Representative images and a dot plot showing TOPBP1 foci in DLD1 *BRCA2^-/-^* prometaphase cells (siCTRL: *n* = 72 and siSLX4: *n* = 73, from three independent experiments). **e** Representative images and a dot plot showing number of MUS81 foci in prometaphase HCT116-TOPBP1-mAID-Clover cells with and without 2 h IAA (500 μM) dependent degradation of TOPBP1 in the presence of 40 ng/ml nocodazole ("-": *n* = 81 and "+": *n* = 84, from three independent experiments). Statistical significance in (**a,b,d,e**) was determined by two-tailed unpaired t-test. Statistical significance in c was determined by the two-way ANOVA test. In a-e individual measurements are represented by grey dots, whereas black dots indicate medians of each experiment, and the mean is presented as bars with error bars showing SEM. Scale bars are equivalent to 10 μm. Source data are provided as a Source Data file.

recruitment in mitotic cells. We observed an increase in the incidence of TOPBP1 foci in prometaphase DLD1 *BRCA2^-/-^* cells, indicating elevated requirement for mitotic TOPBP1 function in these cells (Supplementary Fig. 2a). Consistent with this, and with an upstream role of MDC1 in facilitating TOPBP1 localisation to mitotic chromatin, we observed elevated MDC1 foci formation in mitosis in BRCA1- and BRCA2-deficient cells as well as aphidicolin-treated RPE1 cells compared to respective controls (Supplementary Fig. 2b–d). In line with this, MDC1 knockdown markedly reduced TOPBP1 foci in prometaphase in BRCA1- and BRCA2-deficient cells (Supplementary Fig. 2e–h), but not in aphidicolin-treated RPE1 cells relative to controls (Supplementary Fig. 2i and j), aligning with observations reported by the Durocher lab for CIP2A recruitment under similar conditions[1].

To gain insight into whether BRCA2 deficient cells demonstrate dependency on the association of TOPBP1 and SMX complex components in mitosis we analysed MUS81, ERCC1 and SLX4 colocalisation with TOPBP1 in prometaphase DLD1 WT and *BRCA2^-/-^* cells (Fig. 2a and b and Supplementary Fig. 3a). As such, MUS81, ERCC1 and SLX4 frequently colocalize with TOPBP1 in prometaphase cells (Fig. 2a and b and Supplementary Fig. 3a). This was similarly observed in BRCA1-deficient cells for SLX4 (Supplementary Fig. 3b) or MUS81 (Supplementary Fig. 3c) colocalisation with TOPBP1 in prometaphase. Notably, we observed elevated colocalisation of MUS81, ERCC1 and SLX4 in BRCA1/2 deficient cells when compared to their WT control (Fig. 2a and b, Supplementary Fig. 3a–c). Similarly, treatment of non-cancerous RPE1 cells with aphidicolin, to induce replicative stress, resulted in elevated mitotic colocalisation of SLX4, MUS81 or ERCC1 with TOPBP1 in prometaphase (Supplementary Fig. 3d–f). These findings indicate that TOPBP1 and components of the SMX complex associate in mitosis in response to replication stress induced by aphidicolin, BRCA1 or BRCA2 deficiency, consistent with their defined roles in repair of under-replicated DNA or persistent recombination intermediates[11,12,18]. In agreement with their established functions in responding to replication stress[27,48–51], we also detected elevated formation of TOPBP1, SLX4, and MUS81 foci during interphase in BRCA2-deficient DLD1 cells compared to wild-type controls, as well as in non-cancerous RPE1 cells treated with aphidicolin (Supplementary Fig. 4a–f). Importantly, quantitative analysis revealed a significantly increased percentage of colocalisation events between SLX4/MUS81 and TOPBP1 in mitosis compared to interphase, both in BRCA2-deficient DLD1 cells and RPE1 cells treated with or without aphidicolin (Supplementary Fig. 5a–d). This indicates enhanced recruitment or stabilisation of these complexes in mitosis, consistent with a critical role for TOPBP1-SMX interactions in resolving replication-associated lesions persisting into mitosis[1,12,29]. Interestingly, analysis of Cyclin A2 positive interphase cells also revealed an elevation of SLX4 and MUS81 foci upon TOPBP1 depletion in BRCA2-deficient DLD1 cells and aphidicolin-treated RPE1 cells (Supplementary Fig. 5e–h), suggesting a potential compensatory engagement of these nucleases independently of TOPBP1 in the S/G2 cell cycle phase under conditions of heightened replicative stress[48,51,52].

Given the observed elevation in MUS81-TOPBP1 colocalisation in *BRCA2^-/-^* cells, we sought to ascertain the requirement of TOPBP1 and SLX4 in mediating MUS81 localisation in wild type and *BRCA2^-/-^* DLD1 prometaphase cells (Fig. 2c). Consistently, TOPBP1 depletion by siRNA resulted in a reduction in MUS81 foci formation in both WT and *BRCA2^-/-^* DLD1 cells compared to siRNA control treated cells (Fig. 2c). Similarly, SLX4 depletion, resulted in defects in its localisation akin to those observed with TOPBP1 depletion (Fig. 2c, Supplementary Fig. 6a, b). Consistently, depletion of TOPBP1 in *BRCA1^-/-^* cells impaired MUS81 localisation in prometaphase (Supplementary Fig. 6c and d). However, SLX4 depletion in *BRCA2^-/-^* or *BRCA1^-/-^* cells did not impair TOPBP1 localisation to prometaphase chromatin, suggesting SLX4 and MUS81 act downstream of TOPBP1 (Fig. 2d, Supplementary Fig. 6b, e and f). Together, this data suggests that recruitment of the SMX complex to chromatin in mitosis in BRCA1/2 deficiency is mediated primarily by TOPBP1 and is subsequently stabilised by SLX4.

To determine whether TOPBP1 is required specifically in mitosis for localisation of MUS81, we employed the HCT116-TOPBP1-mAID-Clover degron system to facilitate acute depletion of TOPBP1, by addition of Indole-3-acetic acid (IAA) for 2 h, followed by analysis of prometaphase cells[47] (Supplementary Fig. 1d and Supplementary Fig. 6g). In the presence of TOPBP1 we observed robust formation of MUS81 foci in prometaphase cells (Fig. 2e). In contrast, cells acutely depleted of TOPBP1 demonstrated a near total abolishment of MUS81 foci formation in prometaphase (Fig. 2e), demonstrating TOPBP1 is required in mitosis for MUS81 recruitment.

## The N-terminal module of TOPBP1, containing BRCT 0-1-2, mediates interaction with the SMX complex in a phospho-dependent manner

TOPBP1 consists of nine BRCT domains, which facilitate phosphorylation-dependent protein-protein interactions in a number of distinct complexes[24,27] (Fig. 3a). To gain insight into the mitotic function of TOPBP1-SMX, we sought to map binding site(s) mediating its interaction with components of the SMX complex. To achieve this, we expressed C-terminal truncation mutants of TOPBP1 in mitotic synchronised HEK293TN cells and performed co-immunoprecipitations (Fig. 3b and Supplementary Fig. 7a). Strikingly, all TOPBP1 truncation mutants tested, including BRCT0-2, were sufficient to bind SLX4, MUS81 and ERCC1 of the SMX complex (Fig. 3b). However, interaction between TOPBP1 and the BTR complex components BLM and TOP3a were significantly reduced in cells expressing the BRCT 0-2 and 0-3 mutants, consistent with their previously characterised binding site in BRCT 5 of TOPBP1 (Fig. 3b)[31]. Importantly, N-terminal truncation of TOPBP1, which removed BRCT 0-2 but maintained all other domains of TOPBP1, was sufficient to disrupt interaction between TOPBP1 and components of the SMX complex in mitotically synchronised cells (Fig. 3c). Therefore, BRCT 0-2 of TOPBP1 is both necessary and sufficient to mediate its interaction with components of the SMX.

Based on these observations, we hypothesised that the TOPBP1-SMX interactions may be direct and driven by canonical phosphorylation-dependent – BRCT interactions[25,26]. Bioinformatic

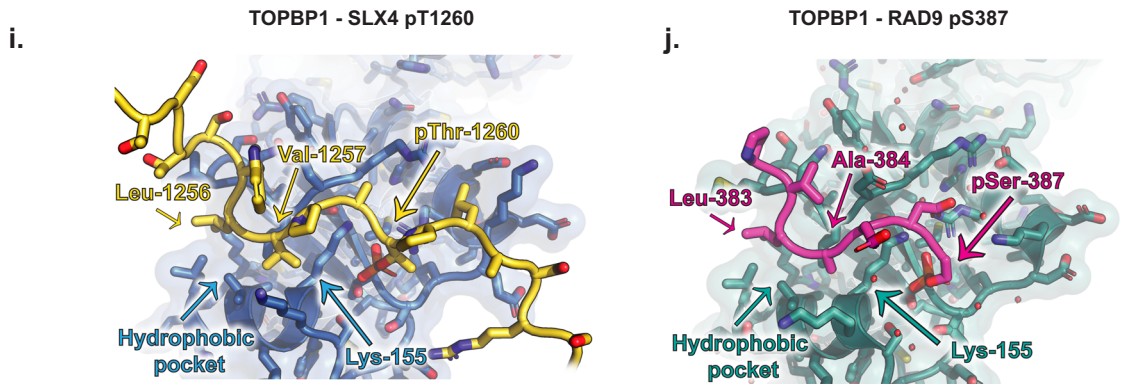

analysis of the SLX4 sequence, scanning for defined BRCT1 binding consensus motifs '[VIL]-[AVIL]-x-x-[ST]-P'[26] identified two CDK consensus motifs centred on the highly conserved residues Thr1260 and Thr1476 (Fig. 3d and Supplementary Fig. 7b). Notably, phosphorylation of SLX4 Thr1260 and Thr1476 have been previously identified in human cells[53].

Using purified TOPBP1 protein fragments[26], we conducted in vitro fluorescence polarisation analysis in the presence of SLX4 peptides containing phospho-Thr1260 (pThr1260) and phospho-Thr1476 (pThr1476). We observed that the pThr1260 SLX4 peptide, interacted with a GST-tagged TOPBP1 fragment containing BRCTs 0-1-2 while the Thr1476-containing phosphorylated peptide did not bind

**Fig. 3 | SLX4 interacts with BRCT1/2 of TOPBP1 via pT1260 in mitotic cells.**
**a** eGFP-TOPBP1 domain architecture schematic depicting relative positions of
TOPBP1 BRCA1 C-terminal (BRCT) domains 0–8 and its ATR activation domain
(AAD). **b** Western blot analysis of BLM, TOP3A, SLX4, MUS81 and ERCC1 in eGFP-
TOPBP1 truncation mutant Co-IPs from HEK293TN cells transiently transfected
with WT or C-terminally truncated eGFP-TOPBP1 expression constructs followed by
18 h 100 ng/ml nocodazole synchronisation from Supplementary Fig. 7a. Mock Co-
IP from non-transfected HEK293TN cells was conducted as negative control (CTRL).
**c** Western blot analysis of SLX4, MUS81 and ERCC1 in eGFP-TOPBP1 Co-IPs from
HEK293TN cells transiently transfected with WT or N-terminally truncated Δ314
(3–8) eGFP-TOPBP1 expression constructs followed by 18 h 100 ng/ml nocodazole
synchronisation. Mock Co-IP from non-transfected HEK293TN cells was conducted
as negative control (CTRL). For all IP experiments 1% of input was used for analysis
by western blot of input lysate. **d** Domain architecture of SLX4 depicting relative
positions of the MUS312/MEI9 interaction-like (MLR), Broad-complex, Tramtrack,
and Bric-a-brac (BTB), SAF-A/B, Acinus, and PIAS (SAP) and Conserved C-terminal
Domain (CCD) domains and the relative position of the identified pT1260, TOPBP1

BRCT 1 recognition motif. **e** Fluorescence polarisation analysis of recombinant
TOPBP1 BRCT0-1-2, BRCT 4-5 and BRCT 7-8 domain containing fragments in the
presence of a fluorescein tagged SLX4 pT1260-containing peptide. **f** Fluorescence
polarisation analysis of recombinant TOPBP1 BRCT0-1-2 fragment in the presence
of fluorescein tagged SLX4 pT1260-containing peptide with or without lambda (λ)
phosphatase treatment. **g** Fluorescence polarisation analysis of recombinant
TOPBP1 BRCT0-1-2 WT or conserved lysine 'K' to glutamic acid 'E' mutations in the
BRCT1 (K155E) or 2 (K250E) or in BRCT 1 + 2 (K155E + K250E) of the protein frag-
ment, in the presence of fluorescein tagged SLX4 pT1260 containing peptide (**e**–**g**
contains data from three independent experiments, error bars display SEM, for all
FP experiments raw data was fit with a specific and non-specific component, with
the non-specific component subtracted before plotting). **h** Table showing the dis-
sociation constant of TOPBP1 BRCT 0-1-2 WT or K250E with SLX4 pT1260. **i** TOPBP1
BRCT 0-1-2 and SLX4 pT260 interaction modelled in AlphaFold 3. **j** TOPBP1 BRCT 0-
1-2 and RAD9 pS387 crystal structure (PDB: 6HM5). Source data are provided as a
Source Data file.

(Fig. 3e and Supplementary Fig. 7b). Furthermore, no interaction was
detected between TOPBP1 fragments containing BRCTs 4-5 or 7-8 for
either the SLX4-pThr1260 or -pThr1476 peptides (Fig. 3e and Supple-
mentary Fig. 7b). Treatment of the pThr1260 peptide with lambda
phosphatase prevented the interaction with BRCT 0-1-2 protein
(Fig. 3f), demonstrating the interaction is phosphorylation-dependent.

Tandem BRCT domains contain conserved lysine residues that
interact with the phosphate group of a modified protein ligand, and
mutating this lysine drastically reduces ligand-binding affinity[24,26,27,37].
In TOPBP1, Lys155 in BRCT1 and Lys250 in BRCT2 are highly conserved.
Introducing K155E, or K155E + K250E, mutations within BRCTs 1 and 2,
disrupted the interaction between the TOPBP1 BRCT 0-1-2 containing
fragment and the phosphorylated SLX4 pThr1260 peptide while the
K250E mutant was able to bind with comparable $K_d$ to WT (Fig. 3g, h).
This demonstrated a preference of the pThr1260 peptide for BRCT1 of
TOPBP1. Modelling of the TOPBP1 BRCT 0-1-2 SLX4 pThr1260 inter-
action in Alphafold 3 supported the structural basis of the interaction
(Fig. 3i). When comparing the modelled interaction to the previously
characterised crystal structure of the TOPBP1 BRCT 0-1-2 and RAD9
pS387 interaction[26], we observed strong similarities in BRCT1-
dependent phospho-ligand recognition (Fig. 3i and j). As such, the
phosphate group of SLX4 pThr1260 is predicted to be recognised by
the conserved triplet residues Thr114, Arg121 and Lys155 in the BRCT1
domain of TOPBP1, as demonstrated experimentally for the RAD9
pSer387 interaction (Fig. 3i and j). Furthermore, several of the inter-
actions predicted N-terminal of the phosphate group residues are
conserved with the hydrophobic side chains of SLX4 Leu1256 and
Val1257, taking the place of RAD9 residues Leu383 and Ala384,
observed sitting in the hydrophobic pocket on the surface that
determines the specificity of BRCT1 (Fig. 3i and j)[26]. Together, this
provides a putative structural basis of the TOPBP1 BRCT 0-1-2 – SLX4
pThr1260 interaction.

To further validate the human TOPBP1 BRCT1 interaction with
SLX4 pThr1260, we expressed eGFP-tagged full-length TOPBP1 con-
taining lysine-to-glutamic acid mutations in BRCT domains 1, 2, and
combined 1 and 2. We expressed these proteins in mitotic synchro-
nised cells and carried out co-immunoprecipitation analysis from the
chromatin fraction or whole cell lysates (Fig. 4a and Supplementary
Fig. 8a). We validated this approach by probing for MDC1, a TOPBP1-
binding protein that interacts with BRCTs 1 and 2[24]. As expected,
mutation of BRCT1 or BRCT2 specifically prevented MDC1 binding
(Supplementary Fig. 8a)[13].

We next assessed the ability of these TOPBP1 mutants to interact
with SLX4, MUS81 and ERCC1. Consistent with our fluorescence
polarisation data, we found that the K155E/K250E double mutant,
which disrupts both BRCT1 and BRCT2, resulted in reduced TOPBP1-
SMX complex (SLX4, MUS81, ERCC1) interactions isolated from

mitotic chromatin (Fig. 4a and Supplementary Fig. 8a). This suggests
that while BRCT1 is critical for the interaction, BRCT2 likely provides
additional stabilising interactions or facilitates indirect binding of
TOPBP1 or SLX4. We also observed reduced recruitment of TOPBP1 to
mitotic chromatin upon mutation of K155E/K250E (Fig. 4a).

To address the role of SLX4 pThr1260 in mediating interaction
with TOPBP1 and components of the SMX complex, we generated a
Thr1260Ala eGFP-SLX4 mutant expression construct and performed
co-immunoprecipitations from mitotic synchronised cells after tran-
sient expression. Strikingly, mutating SLX4 Thr1260 to alanine abol-
ished binding with TOPBP1 but had no discernible effect on the
interaction with MUS81 (Fig. 4b), providing evidence for a TOPBP1-
independent interaction mechanism for these proteins.

Therefore, we conclude that the TOPBP1-SLX4 interaction
involves direct binding of phosphorylated SLX4-Thr1260 to TOPBP1
BRCTs 1 and 2. In addition, we establish that MUS81 and ERCC1 bind to
the BRCT1/2 module of TOPBP1 in mitotic cells (Fig. 4a), suggesting
assembly of the SMX complex components on chromatin via interac-
tions with the BRCT1 and 2 domains of TOPBP1.

## CDK1-dependent phosphorylation of threonine 1260 of SLX4 promotes its interaction with TOPBP1

The mitotic kinase CDK1 is maximally active at the G2-M transition[54–56]
and SLX4 Thr1260 resides within a predicted minimal CDK consensus
site[28]. We hypothesised that the TOPBP1-SLX4 interaction is likely
regulated by CDK1. To test this, we raised a phospho-specific antibody
against a peptide encompassing phosphorylated SLX4-Thr1260. Con-
sistent with mitotic-specific phosphorylation at SLX4-Thr1260, the
antibody preferentially detected phosphorylation in cells isolated by
mitotic shake off, which was abolished by phosphatase treatment,
indicating its phospho-specificity (Supplementary Fig. 8b). This phos-
phorylation was reduced when cells were acutely treated with a CDK1
inhibitor (CDK1i) after mitotic synchronisation, indicating that CDK1 is
required for SLX4 phosphorylation (Fig. 4c and d).

To ascertain the functional consequences of disruption of this
phosphorylation, we examined the interaction between eGFP-SLX4
and TOPBP1, MUS81, or ERCC1 after acute mitotic inhibition of CDK1
(Fig. 4c). Acute mitotic inhibition of CDK1 reduced SLX4 phosphor-
ylation of Thr1260 and binding of TOPBP1 but did not affect its inter-
action with MUS81 or ERCC1 (Fig. 4c). Similarly, CDK1i treatment
hindered the interaction between eGFP-TOPBP1 and SLX4 in reciprocal
immunoprecipitation experiments but did not affect the interaction
between TOPBP1 and components of the BTR complex (Fig. 4d).
Notably, CDK1i treatment disrupted the interaction between TOPBP1
and MUS81 or ERCC1, suggesting that these interactions may also be
phosphorylation-dependent and CDK1-regulated (Fig. 4d). Con-
sistently, we observed a marked reduction in SLX4 Thr1260

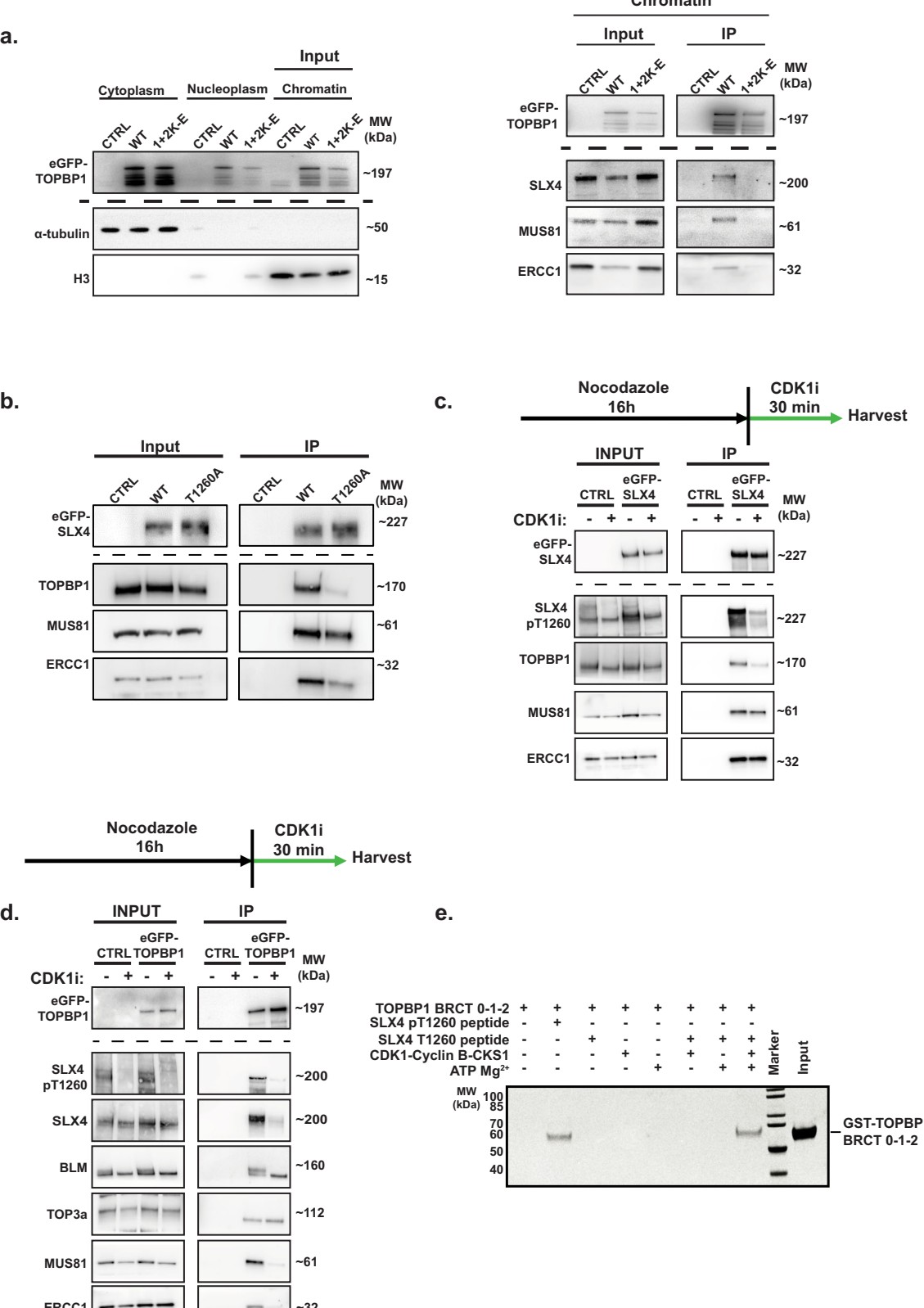

phosphorylation following acute CDK1 inhibition, in an eGFP-TOPBP1 Co-IP (Fig. 4d). Moreover, co-immunoprecipitation of eGFP-SLX4 from chromatin revealed that acute inhibition of CDK1 markedly impaired chromatin residency of eGFP-SLX4 and MUS81 (Supplementary Fig. 8c), providing evidence that CDK1-dependent phosphorylation is essential for stable recruitment of the SMX complex to mitotic chromatin and its assembly. These findings indicate that stable chromatin residency of the SMX complex during mitosis depends on intact BRCT1/2 domains of TOPBP1 and CDK1-dependent phosphorylation likely via SLX4.

To corroborate the direct phosphorylation of SLX4 by CDK1, we reconstituted the CDK1-SLX4 phosphorylation, TOPBP1 interaction

**Fig. 4 | Mitotic interaction of the SMX complex components and TOPBP1 is driven by CDK1 activity. a** Western blot analysis of fractionated HEK293TN cells treated with 100 ng/ml of nocodazole for 18 hours, 24 hours after transient transfection with eGFP-TOPBP1 WT, or 1 + 2 (K155E + K250E) expression constructs and western blot analysis of SLX4, MUS81 and ERCC1 in Co-IPs from the chromatin fraction of these cells. Mock Co-IP from non-transfected HEK293TN cells was conducted as negative control (CTRL). **b** Western blot analysis of TOPBP1, MUS81 and ERCC1 in eGFP-SLX4 Co-IPs from the chromatin fraction of HEK293TN cells treated with 100 ng/ml of nocodazole for 18 h after transient transfection of eGFP-SLX4 WT and T1260A expression constructs. Mock Co-IP from non-transfected HEK293TN cells was conducted as negative control (CTRL). **c** Western blot analysis of CDK1 dependent phosphorylation of SLX4 T1260 and interaction with MUS81, ERCC1 and TOPBP1 in eGFP-SLX4 Co-IPs. HEK293TN cells were transfected with an eGFP-SLX4 expression construct and 24 h later were synchronised for 18 h with 100 ng/ml nocodazole. This was followed with or without 7 μM RO-3306 (CDK1i) treatment for 30 min, prior to collection. Mock Co-IP from non-transfected HEK293TN cells act as negative control (CTRL). **d** As in C, but reciprocal eGFP-TOPBP1 Co-IP analysis. For all IP experiments 1% of input was used for analysis by western blot of input lysate. **e** SDS-PAGE of in vitro reconstitution of CDK1-Cyclin B-CKS1 driven phosphorylation of a SLX4 T1260 containing biotinylated peptide and its phosphorylation-dependent interaction with a recombinant TOPBP1 BRCT 0-1-2 containing fragment, demonstrated through peptide pull down. Source data are provided as a Source Data file.

cascade in vitro using purified recombinant CDK1-Cyclin B-CKS1 (Fig. 4e and Supplementary Fig. 8f), GST-TOPBP1 BRCTs 0-1-2, and a biotinylated-SLX4 Thr1260 peptide in the presence of ATP and $Mg^{2+}$. As such, only streptavidin pull-down of the biotinylated-SLX4 Thr1260 peptide in the presence of TOPBP1 BRCTs 0-1-2, ATP, $Mg^{2+}$ and recombinant CDK1-Cyclin B-CKS1 facilitated the detection of the SLX4 Thr1260 peptide- TOPBP1 BRCT 0-1-2 interaction, comparable to incubation of a pThr1260 peptide with TOPBP1 BRCT 0-1-2 (Fig. 4e). This experiment confirmed that direct phosphorylation of SLX4 by CDK1-Cyclin B-CKS1, may facilitate the mitotic SLX4-TOPBP1 interaction (Fig. 4e).

Consistent with this, our immunofluorescence microscopy analysis of RPE1 p53[-/-] cells treated with aphidicolin followed by acute CDK1i treatment demonstrated an impairment of MUS81 and TOPBP1 foci formation in prometaphase cells (Supplementary Fig. 8d and e), Therefore, mitotic phosphorylation of SLX4 by CDK1 regulates its interaction with TOPBP1 and this interaction is mediated by the direct binding of phospho-Thr1260 to the BRCT 0-1-2 domains of TOPBP1.

### Threonine 1260 of SLX4 is required for recruitment of components of the SMX complex to chromatin during mitosis and to promote genome stability

To establish the functional role of the mitotic CDK1-driven TOPBP1-SLX4 interaction in SLX4 recruitment, we generated SLX4[-/-] cells by CRISPR/Cas9-mediated genome editing and complemented them with inducible wild-type (WT) eGFP-SLX4 or the Thr1260Ala mutant that cannot be phosphorylated to mediate interaction with TOPBP1. Subsequently, we induced expression by the addition of doxycycline (Supplementary Fig. 9a), then treated these cells without or with aphidicolin, to induce replication stress. As TOPBP1 is complexed with CIP2A in mitosis[1,13] we assessed SLX4 colocalisation with CIP2A on prometaphase chromatin (Fig. 5a).

We observed that WT SLX4 was recruited to prometaphase chromatin in response to replicative stress induced by aphidicolin and colocalised with CIP2A (Fig. 5a). In contrast to WT SLX4, the Thr1260Ala mutant displayed total abrogation of chromatin recruitment in mitosis (Fig. 5a). Importantly, the Thr1260Ala SLX4 mutant did not hinder CIP2A recruitment to prometaphase chromatin (Supplementary Fig. 9b) and displayed comparable competency to WT in its ability to form foci in interphase cells (Supplementary Fig. 9c).

To assess the functional role of the TOPBP1-SLX4 interaction in mitotic recruitment of MUS81 and ERCC1, we generated cells harbouring the SLX4 Thr1260Ala mutation at the endogenous locus by CRISPR-mediated knock-in (Supplementary Fig. 10a–c). Following the induction of replication stress with aphidicolin, we examined colocalisation of CIP2A with MUS81 or ERCC1 (Fig. 5b and Supplementary Fig. 11a). In two independent clones of Thr1260Ala knock-in cells, the colocalisation of MUS81 and ERCC1 with CIP2A was significantly disrupted, in stark contrast to WT cells (Fig. 5b and Supplementary Fig. 11a).

Given that SLX4 and MUS81 are required for MiDAS we hypothesised that the TOPBP1-SLX4 interaction may be required to facilitate this mitotic DNA repair mechanism[11,12,57]. Therefore, we treated WT and Thr1260Ala mutant cells with aphidicolin and monitored EdU incorporation in prometaphase cells (Fig. 5c). We observed that EdU incorporation was significantly reduced in two independently derived Thr1260Ala mutant clones, when compared to WT cells, indicating an impairment of MiDAS in SLX4 Thr1260Ala cells (Fig. 5c). Consistent with this, cells expressing the SLX4 Thr1260Ala mutant, displayed elevated levels of genome instability, as indicated by micronuclei formation (Fig. 5d). SLX4[-/-] cells displayed a more pronounced phenotype than the Thr1260Ala knock-in clones (Fig. 5d), reflecting the mitosis-specific function of Thr1260 phosphorylation as opposed to the broader role of SLX4 in DNA repair during both interphase and mitosis.

Collectively, these findings demonstrate that the functional role of the TOPBP1-SLX4 interaction in human cells is to facilitate mitotic recruitment of the SMX complex components, SLX4, MUS81, and ERCC1, to sites of DNA damage marked by CIP2A-TOPBP1 during mitosis, to drive MiDAS and safeguard genome stability.

### CIP2A regulates SLX4 and Polθ recruitment to orchestrate mitotic DNA repair

The role of CIP2A in the regulation of DNA repair activity remains relatively unknown. We hypothesised that CIP2A may act upstream of SLX4 given that SLX4 depletion did not alter TOPBP1 recruitment and the Thr1260Ala mutation did not impair mitotic CIP2A localisation (Figs. 2d, 5a, b and Supplementary Fig. 11a). To further substantiate this model, we first examined the colocalisation of MUS81 and TOPBP1 foci in CIP2A knockout cells (Supplementary Fig. 13a). Consistently, CIP2A[-/-] cells exhibited a near-complete absence of mitotic MUS81 and TOPBP1 foci, and consequently an impairment of MUS81-TOPBP1 colocalisation was observed (Fig. 6a). Next, we analysed the mitotic-specific recruitment of eGFP-SLX4 WT and its colocalisation with TOPBP1 in prometaphase cells where CIP2A was present or depleted using validated siRNA[1,13](Supplementary Fig. 11b). Consistent with our hypothesis, we observed that SLX4 colocalised with TOPBP1 in mitosis (Supplementary Fig. 11b), and a near-complete absence of SLX4 and TOPBP1 colocalising foci on prometaphase chromatin in CIP2A-depleted cells (Supplementary Fig. 11b and c). As expected, we also observed a reduction in TOPBP1 foci formation on prometaphase chromatin in CIP2A-depleted cells (Supplementary Fig. 11c and d). Analysis of CIP2A localisation demonstrated that the majority of CIP2A is present in the cytoplasm of Cyclin A positive or negative interphase cells, with striking localisation to mitotic chromatin (Supplementary Fig. 12), in further support of a mitotic-specific role in this context.

Given the established roles of SLX4 and MUS81 in MiDAS and their impaired recruitment we observed in CIP2A deficient cells, we analysed MiDAS by measuring EdU incorporation. This analysis revealed a significant defect in MiDAS in two independent CIP2A[-/-] clones compared to WT parental cells (Fig. 6b and Supplementary Fig. 13a). As BRCA2 deficient cells demonstrate an elevated reliance on MiDAS (Supplementary Fig. 13b)[58], we analysed MiDAS in DLD1 BRCA2[-/-] cells depleted of CIP2A by siRNA (Fig. 6c and Supplementary Fig. 13c). Upon

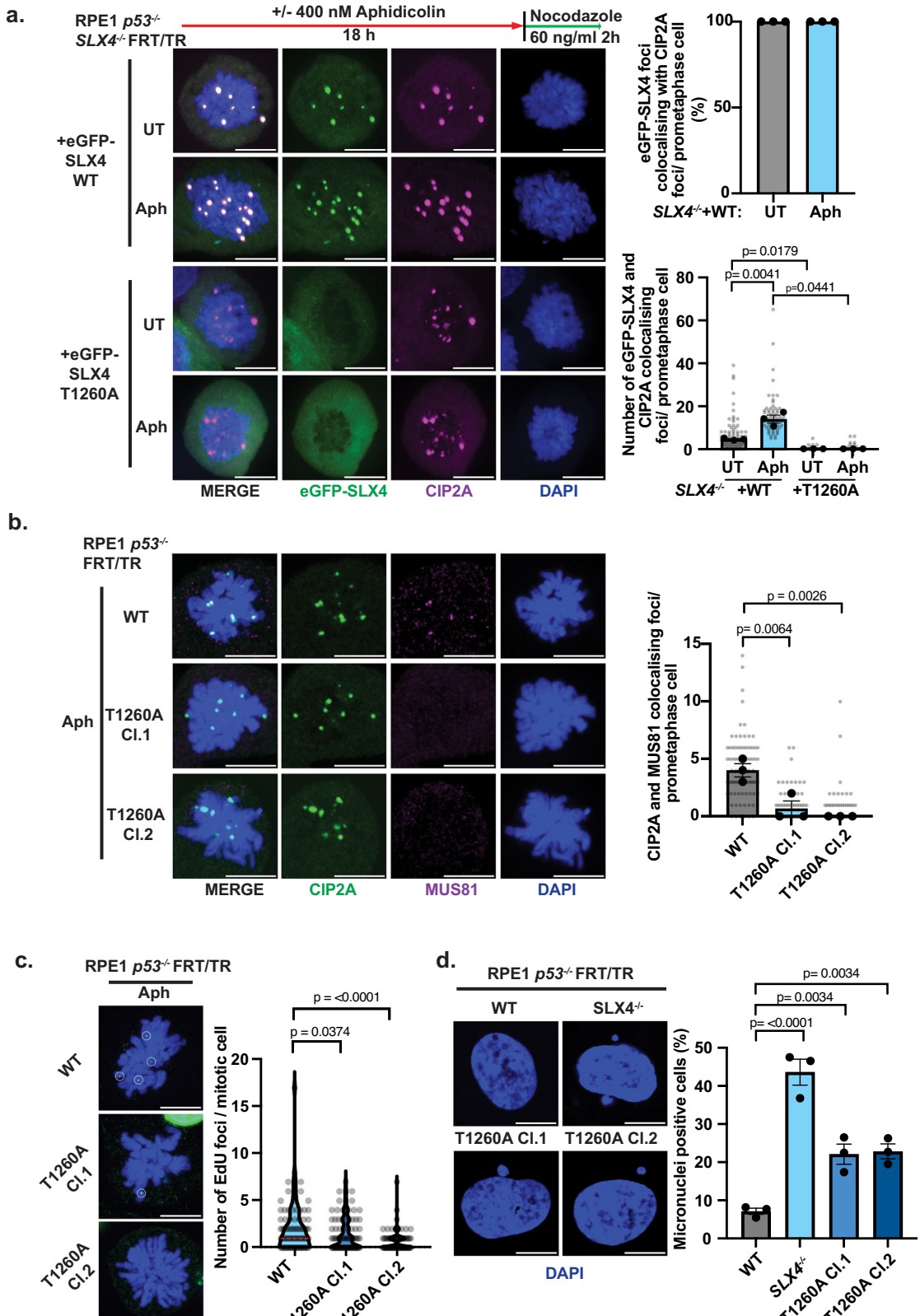

CIP2A depletion we observed a reduction in EdU incorporation in CIP2A-depleted BRCA2-deficient cells (Fig. 6c), highlighting a pivotal role for CIP2A in facilitating MiDAS in *BRCA2*[-/-]. Since MiDAS operates through a break-induced replication (BIR)-like mechanism[19], we sought to strengthen this observation by investigating the role of CIP2A in BIR-dependent DNA repair utilising a BIR reporter (Fig. 6d)[59,60]. As

expected, depletion of POLD3 or SLX4, known drivers of MiDAS, significantly impaired efficiency of DSB repair via BIR when compared to siCTRL-treated cells (Fig. 6d, Supplementary Fig. 13d and e). In line with our prediction, CIP2A depletion also significantly reduced BIR efficiency (Fig. 6d, Supplementary Fig. 13d and e), providing direct evidence that CIP2A is critical for the repair of DSBs by BIR.

**Fig. 5 | Mitotic localisation of the SMX complex components is CIP2A-TOPBP1 dependent and facilitates unscheduled DNA synthesis to safeguard genome stability. a** Representative images, a bar plot showing the mean percentage of eGFP-SLX4 WT foci colocalising with CIP2A per RPE1 *p53*[-/-] *SLX4*[-/-] FRT/TR +eGFP-SLX4 WT/ T1260A prometaphase cell after induction with 10 ng/ml doxycycline for 24 h followed by treatment with or without 400 nM/ 18 h aphidicolin followed by synchronisation with 60 ng/ml nocodazole for 2 h (UT: *n* = 85, Aph: *n* = 76), and a dot plot showing the number of eGFP-SLX4 WT or eGFP-SLX4 T1260A foci colocalising with CIP2A foci per prometaphase cell after treatment as above (eGFP-SLX4 WT (UT: *n* = 85, Aph: *n* = 76) and eGFP-SLX4 T1260A (UT: *n* = 70, Aph: *n* = 76)) from three independent experiments, statistical significance was determined by two-way ANOVA. Grey dots represent individual measurements, black dots represent the medians of each experiment and bars show the mean with error bars showing SEM. **b** Representative images and a dot plot showing CIP2A and MUS81 colocalising foci in RPE1 *p53*[-/-] FRT/TR WT and SLX4 T1260A knock in prometaphase cells treated with 400 nM aphidicolin followed by synchronisation with 60 ng/ml nocodazole

for 2 h (Parental WT: *n* = 80, T1260A Cl.1: *n* = 76, T1260A Cl.2 *n* = 85, from three independent experiments, statistical significance was determined by one-way ANOVA test). Grey dots represent individual measurements, black dots represent the medians of each experiment and bars show the mean with error bars showing SEM. **c** Representative images and violin plot of the number of EdU foci in RPE1 *p53*[-/-] FRT/TR WT and T1260A prometaphase cells (Parental WT: *n* = 90, T1260A Cl.1: *n* = 87, T1260A Cl.2 *n* = 82, from three independent experiments, statistical significance was determined by two-sided Mann-Whitney U test). **d** Representative images and bar plot of the mean of the percentage of micronuclei-positive RPE1 *p53*[-/-] FRT/TR WT, *SLX4*[-/-] and SLX4 T1260A knock in cells (Parental WT *n* = 336, *SLX4*[-/-] *n* = 346, T1260A Cl.1: *n* = 384, T1260A Cl.2 *n* = 377 from three independent experiments, statistical significance was determined by one-way ANOVA), black dots represent the mean of each experiment and bars show the mean with error bars showing SEM. Scale bars in (**a**–**d**) are equivalent to 10 μm. Source data are provided as a Source Data file.

To support these findings, we investigated if CIP2A directly localises to sites of DNA damage in mitosis by analysing colocalisation of CIP2A with γH2AX, a marker of DNA damage, in RPE1 cells treated with aphidicolin and in BRCA2 deficient DLD1 cells. Aphidicolin-treated RPE1 *p53*[-/-] and BRCA2 deficient cells displayed a striking increase in CIP2A–γH2AX colocalising foci in mitosis (Fig. 6e and Supplementary Fig. 13f), supporting a role of CIP2A in the mitotic DNA damage response. Consistent with a coordinated response to mitotic DNA damage, we also observed a significant increase in colocalising γH2AX and SLX4 foci in aphidicolin treated RPE1 *p53*[-/-] cells and in BRCA2 deficient DLD1 cells when compared to their respective untreated or WT control (Fig. Supplementary Fig. 13g and h). The same observation was made for colocalisation between γH2AX and MUS81 in RPE1 *p53*[-/-] cells treated with aphidicolin (Supplementary Fig. 13i). Importantly, we also detected colocalisation between CIP2A and FANCD2 foci on mitotic chromatin in response to aphidicolin treatment in RPE1 *p53*[-/-] cells (Fig. 6f). Given the well-established role of FANCD2 in localising to under-replicated DNA structures, fragile sites during mitosis and in facilitating MiDAS[61–64], this finding suggests that CIP2A acts together with TOPBP1 and SLX4 to facilitate the resolution of replication-associated lesions that persist into mitosis.

Recently, Gelot et al. demonstrated that Polθ phosphorylation by PLK1 facilitates its interaction with TOPBP1, driving mitotic MMEJ[5]. Given the mutual dependency between CIP2A and TOPBP1 for chromatin recruitment during mitosis[1,13], we hypothesised that CIP2A may also regulate mitotic MMEJ by facilitating the mitotic recruitment of Polθ to mitotic chromatin. To test this hypothesis, we generated DLD1 *BRCA2*[-/-] cells that express eGFP-tagged Polθ in an inducible manner and analysed its recruitment to mitotic chromatin following CIP2A depletion (Supplementary Fig. 14a). In siCTRL-treated cells, eGFP-Polθ was recruited to prometaphase chromatin; however, this recruitment was reduced in CIP2A-depleted cells (Fig. 6g and Supplementary Fig. 14a). To assess the role of CIP2A in MMEJ, we used an U2OS cell line expressing an MMEJ reporter construct (Fig. 6h, Supplementary Fig. 14b and c)[65]. Depletion of Polθ in this system led to a significant defect in MMEJ efficiency, validating our approach. Strikingly, we observed that CIP2A depletion led to a marked impairment in MMEJ (Fig. 6h, Supplementary Fig. 14b and c), which is consistent with the observed defective recruitment of Polθ upon CIP2A depletion (Fig. 6g and Supplementary Fig. 14a). Together these findings demonstrate a critical role for CIP2A in facilitating not only BIR-like MiDAS but also Polθ-mediated MMEJ during mitosis.

Next, we examined the impact of CIP2A loss (Supplementary Fig. 14d) in comparison to the combined disruption of the TOPBP1-SLX4 interaction (required for MiDAS) and Polθ (required for MMEJ) inhibition (Polθi) on genome stability in response to replication stress induced by aphidicolin. We observed an increase in micronuclei formation in response to Polθ inhibition, although this increase was less

pronounced than in SLX4 Thr1260Ala mutant cells (Fig. 7a). Notably, inhibition of Polθ (required for MMEJ) in the SLX4 Thr1260Ala mutant cells led to an even greater increase in micronuclei formation, indicating an additive effect. Supporting this, SLX4 Thr1260Ala mutant cells also displayed significantly increased genome instability as indicated by elevated γH2AX and 53BP1 foci formation, which was further exacerbated upon Polθ inhibition (Supplementary Fig. 14e and f). Consistent with the role of CIP2A in facilitating both MMEJ and MiDAS repair in mitosis, its depletion in WT cells resulted in similar levels of micronuclei formation as observed in SLX4 Thr1260Ala cells treated with Polθ (Fig. 7a and Supplementary Fig. 14d).

Analysis of cellular proliferation in the presence of aphidicolin revealed that SLX4 Thr1260Ala mutant knock-in cells displayed proliferation defects that were exacerbated by Polθi treatment (Fig. 7b). In line with this, depletion of SLX4 in combination with Polθi resulted in a broadly similar loss of proliferative capacity compared to the single depletion of CIP2A alone in BRCA2 deficient DLD1 cells (Fig. 7c and Supplementary Fig. 15a). Moreover, knockdown of BRCA1 or BRCA2 in SLX4 Thr1260Ala mutant cells resulted in further reduced proliferation compared to SLX4 Thr1260Ala alone (Supplementary Fig. 15b–d), consistent with the requirement for the SLX4-TOPBP1 interaction in resolving persistent replication stress arising from BRCA1/2 deficiency[1]. Notably, this proliferation defect was further exacerbated by Polθ inhibition (Supplementary Fig. 15b–d), corroborating our earlier observations with aphidicolin treatment and highlighting the cooperative roles of MiDAS and MMEJ in mitigating replication-associated DNA damage, to permit proliferation. Consistent with BRCA1/2 deficient cells being reliant on the TOPBP1-SLX4 interaction to facilitate MiDAS and safeguard genome stability, we also observed that induction of expression of a minimal SLX4 Thr1260 containing fragment in DLD1 *BRCA2*[-/-] (Supplementary Fig. 15e) or SUM149PT *BRCA1*[-/-] (Supplementary Fig. 15f) cells was sufficient to impair cellular proliferation.

We conclude that CIP2A is essential for facilitating mitotic DNA double-strand break repair through multiple mitotic DNA repair pathways, including coordination of BIR-like processes through the TOPBP1-SLX4 axis and MMEJ through Polθ. This work identifies a critical role for CIP2A in maintaining genome stability during mitosis through DNA repair and provides a mechanistic basis for its synthetic lethality upon loss in BRCA1/2-deficient cells (Fig. 7d).

## Discussion

Recent studies underscore the importance of mitosis as a critical phase for the maintenance of genome integrity and highlight the pivotal role of the CIP2A-TOPBP1 axis in promoting chromatin tethering[1,8,32,33]. Although direct evidence for CIP2A's role in regulating mitotic DNA repair remains limited[1,13], recent studies have elucidated mechanisms that safeguard genome stability during mitosis. One such mechanism

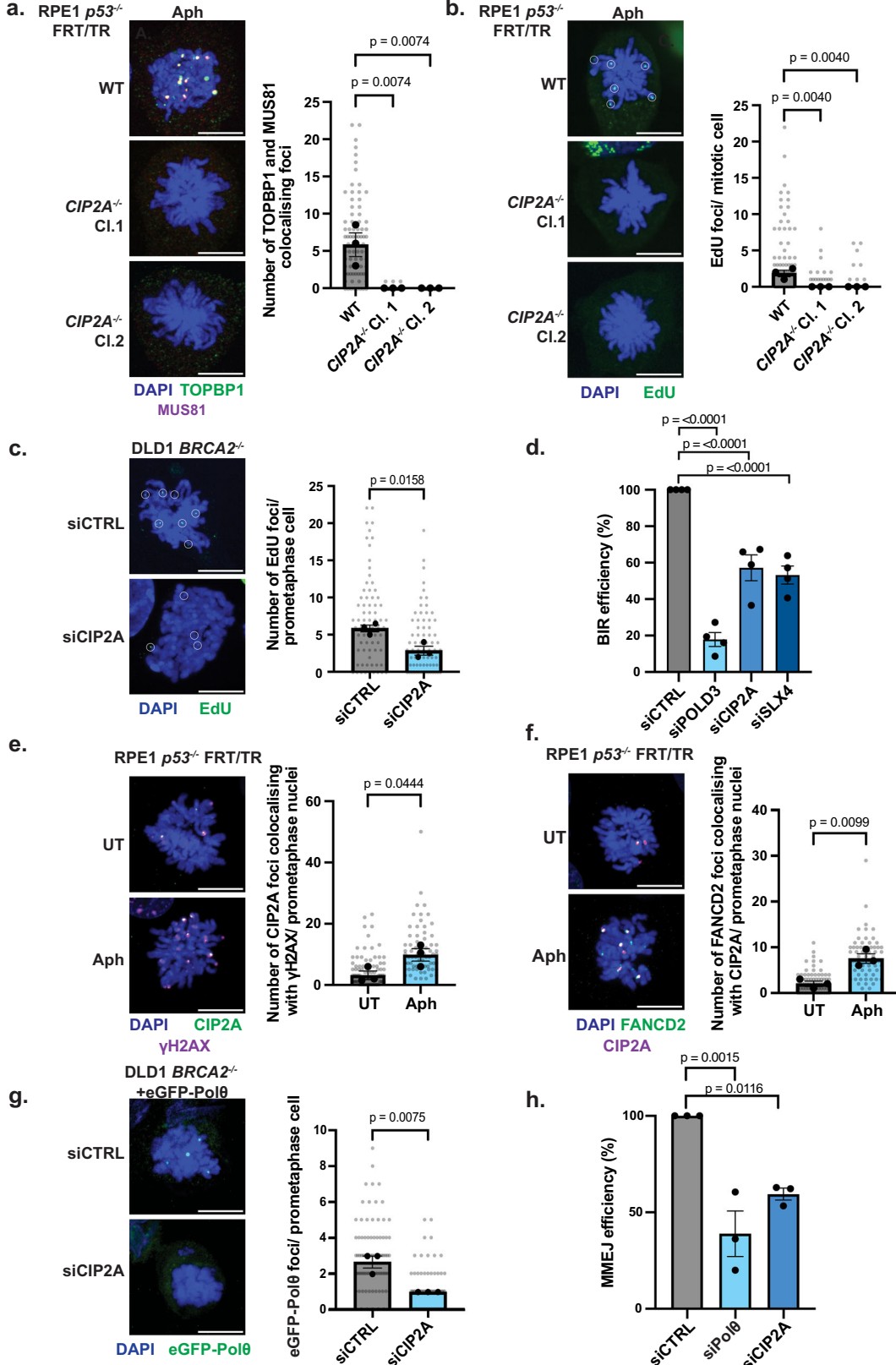

involves PLK1-dependent phosphorylation of Polθ, which enhances its interaction with TOPBP1 and facilitates its recruitment to mitotic chromosomes, thereby promoting mitotic MMEJ[5]. Additionally, RHNO1 has been shown to regulate mitotic MMEJ, representing another pathway that contributes to genome integrity during mitosis[5,10]. Research from the Fox laboratory further implicates Polθ

and FANCD2 as epistatic partners in the repair of DSBs during mitosis[66,67]. Their coordinated activity helps prevent genome instability in *Drosophila melanogaster* mitotic chromosomes, suggesting a collaborative role for FANCD2 and Polθ in maintaining mitotic genome stability[66,67]. Notably, elegant work from several laboratories has demonstrated that disrupting the CIP2A-TOPBP1 and TOPBP1-Polθ

**Fig. 6 | CIP2A is required for the regulation of redundant mitotic repair pathways. a** Representative images and dot plot showing the number of TOPBP1–MUS81 colocalising foci in RPE1 $p53^{-/-}$ FRT/TR WT or $CIP2A^{-/-}$ prometaphase cells following 400 nM aphidicolin for 18 h and synchronisation with 60 ng/ml nocodazole for 2 h (WT: $n = 78$; $CIP2A^{-/-}$ Cl.1: $n = 79$; Cl.2: $n = 80$; three independent experiments). **b** Representative images and dot plot of EdU foci in prometaphase RPE1 $p53^{-/-}$ WT or $CIP2A^{-/-}$ cells after following 400 nM aphidicolin followed by 30 min synchronisation with 60 ng/ml nocodazole (WT: $n = 77$; Cl.1: $n = 69$; Cl.2: $n = 74$; three experiments). **c** EdU foci in prometaphase DLD1 $BRCA2^{-/-}$ cells treated with siCTRL or siCIP2A (siCTRL: $n = 75$; siCIP2A: $n = 74$; three experiments; significance determined by two-tailed unpaired t-test). **d** Bar plot showing break-induced replication (BIR) efficiency after sgRNP-CAS9 cleavage of an I-SceI site in the pBIR-GFP reporter U2OS cell line treated with siCTRL, siPOLD3, siCIP2A, or siSLX4 (four experiments). **e** Representative images and dot plot of CIP2A–γH2AX colocalising foci in RPE1 $p53^{-/-}$FRT/TR WT cells untreated or treated with 400 nM aphidicolin for 18 h followed by synchronisation with 60 ng/ml nocodazole for 2 h (UT: $n = 82$; Aph: $n = 69$; three experiments; two-tailed unpaired t-test). **f** Representative images and dot plot of FANCD2–CIP2A colocalising foci in RPE1 $p53^{-/-}$ WT cells under the same conditions as (**e**) (UT: $n = 75$; Aph: $n = 70$; three experiments; two-tailed unpaired t-test). **g** eGFP-Polθ foci in prometaphase DLD1 $BRCA2^{-/-}$ eGFP-Polθ cells treated with doxycycline (100 ng/ml, 24 h) 48 h after siRNA treatment with siCTRL or siCIP2A for (siCTRL: $n = 86$; siCIP2A: $n = 83$; three experiments; two-tailed unpaired t-test). **h** Bar plot of alternative end-joining (Alt-EJ)/microhomology-mediated end joining (MMEJ) efficiency after sgRNP-Cas9 cleavage of an I-SceI site in EJ2 reporter U2OS cells treated with siCTRL, siPolθ, or siCIP2A (three experiments). In (**a,b,c,e,f,g**), grey dots represent individual measurements, black dots indicate medians per experiment, and bars show mean ± SEM. Scale bars: 10 μm. Statistical significance: a, b, d, and h by one-way ANOVA; others by two-tailed unpaired t-test. Source data are provided as a Source Data file.

---

interactions selectively impairs the growth of BRCA1/2-deficient cells. Despite these recent advances, the underlying molecular mechanism driving the SL between CIP2A loss and BRCA1/2 deficiency remains unclear, since chromosome tethering does not fully explain this phenotype, given that MDC1, which plays a role in mitotic chromosome tethering, does not exhibit synthetic lethality with BRCA1/2 loss[32–35].

In this study, we demonstrate that the CIP2A-TOPBP1 complex plays a vital role in mitotic DSB repair in conditions of elevated and persistent DNA replication stress. We show that the mitotic kinase CDK1 mediates phosphorylation of SLX4 at Thr1260, which is essential for its interaction with BRCT1/2 of TOPBP1, facilitating the recruitment of SMX complex components to DNA lesions marked by CIP2A-TOPBP1 in mitosis. Our analysis reveals that recruitment of SLX4 and MUS81 is largely independent of TOPBP1 outside of mitosis, indicative of alternative and potentially compensatory mechanisms[48,50–52]. Consistently, our chromatin immunoprecipitation and immunofluorescence microscopy data confirm that acute inhibition of CDK1 impairs SMX complex component chromatin residency in mitosis, emphasising mitotic phosphorylation-dependent regulation of this process. These results define CIP2A-TOPBP1 as a central spatial and temporal coordinator of mitotic DNA repair essential for genome integrity. This phosphorylation event initiates SMX component interaction with TOPBP1 and assembly on chromatin to promote BIR-like MiDAS and consequently genome stability during mitosis. Given that phosphorylation of Slx4 at S486A enhances its interaction with Topbp1 in yeast[28,29], and SLX4 T1260 in humans[28,68], our work functionally characterises an evolutionarily conserved regulatory mechanism promoting DSB repair through BIR/MiDAS. Importantly, disruption of this mechanism likely underpins the synthetic lethality observed with SLX4 or MUS81 loss in BRCA2 deficiency[58].

Interestingly, our use of the BIR reporter system employs Cas9 to initiate cleavage, often attributed to SLX4 and MUS81 activity[12,18,59,69], raising the question of the precise stage in BIR that CIP2A and SLX4 may be required. We speculate, based on our SLX4-MUS81 recruitment, mitotic EdU incorporation and BIR reporter data, that CIP2A-TOPBP1-SLX4 are implicated in the regulation of multiple stages of BIR-like repair. This may include replication fork cleavage, D-loop resolution and regulation of additional reinvasion cleavage cycles, ahead of POLD3 synthesis[11,12,18,68,70,71]. While our data directly implicate CIP2A in the regulation of BIR-like MiDAS in mitosis, our BIR reporter experiments were conducted in asynchronous cells. This also raises the additional possibility that CIP2A may impact BIR-like repair in *cis* or in a *trans* manner outside of mitosis, or alternatively that a large proportion of BIR activity assayed by this reporter is mitosis dependent[59].

Our findings indicate that CIP2A also facilitates the recruitment of Polθ to mitotic chromatin to promote Polθ-dependent MMEJ. Consequently, CIP2A depletion compromises Polθ, SMX component and TOPBP1 recruitment to mitotic chromatin, significantly impairing both MiDAS and MMEJ efficiency. Accordingly, simultaneous disruption of Polθ-mediated MMEJ (pharmacological inhibition), and SLX4-mediated MiDAS (Thr1260Ala mutation) leads to elevated micronuclei, increased DNA damage markers (γH2AX and 53BP1) and impaired cellular proliferation when challenged with aphidicolin or BRCA1/2 deficiency. These observations advocate that SLX4 and Polθ act as functionally independent downstream effectors of the CIP2A-TOPBP1 complex, acting non-epistatically to maintain genome stability and support cellular proliferation. In support of our observations in human cells Carvajal-Garcia et al. defined that combined loss of Polθ, SLX4 and BRCA2 hypersensitised *D. melanogaster* cells to ionising radiation and replication stress induced by camptothecin[72]. However, we do not rule out the possibility that Polθ-mediated MMEJ and SLX4-TOPBP1-driven mechanisms may act in a coordinated, rather than fully independent manner to repair DNA lesions that persist into mitosis[66,67,73].

A notable implication of our findings and recent work from Gelot et al. is the coordinated assembly of components of the SMX complex and Polθ in mitosis through TOPBP1. Specifically, CDK1-driven phosphorylation of SLX4 at Thr1260 facilitates its interaction with the BRCT1/2 of TOPBP1, promoting the recruitment of the SMX complex components to sites of DNA damage. Concurrently, PLK1-mediated phosphorylation of Polθ at S1482, S1486, S1488 and S1493 enables its interaction with BRCT 7/8 of TOPBP1 and mitotic chromatin association[5]. This highlights a tightly regulated, multi-step process essential for mitotic DNA repair. This underscores CIP2A-TOPBP1 as a multifunctional regulatory hub integrating kinase signalling to control spatial and temporal regulation of mitotic genome maintenance.

CIP2A depletion in mouse embryonic fibroblasts (MEFs) does not affect telomere fusions dependent on MMEJ in the absence of TRF2, suggesting species/context-specific distinctions in the regulation of Polθ-mediated MMEJ[5,10,36,66]. Indeed, alternative mechanisms may predominate in specific contexts[36]. Nonetheless, our study provides evidence for the critical role of CIP2A in coordinating MiDAS and MMEJ-dependent DNA repair during mitosis, while additional CIP2A-TOPBP1-independent mechanisms may operate concurrently during S phase or mitosis[10,74,75]. In agreement, our analysis shows that SLX4 and MUS81 are recruited to chromatin in interphase cells in the absence of TOPBP1.

Together, these findings provide a mechanistic rationale for targeting the CIP2A-TOPBP1 axis, as well as downstream repair pathways involving SLX4 and Polθ, particularly in tumours that exhibit elevated replicative stress. We speculate that CIP2A may act to provide replication stress tolerance through stabilisation of phosphorylation-dependent protein-protein interactions with TOPBP1 or phosphorylation driven repair signalling[7]. Given the ability of TOPBP1 to oligomerise and sustain multiple protein-protein interactions via distinct BRCT domains in higher order complexes that may contain distinct sub-complexes, this would in turn enable the establishment of an expansive hub that functions to stabilise genome integrity[1,13,24,27,37,68,76].

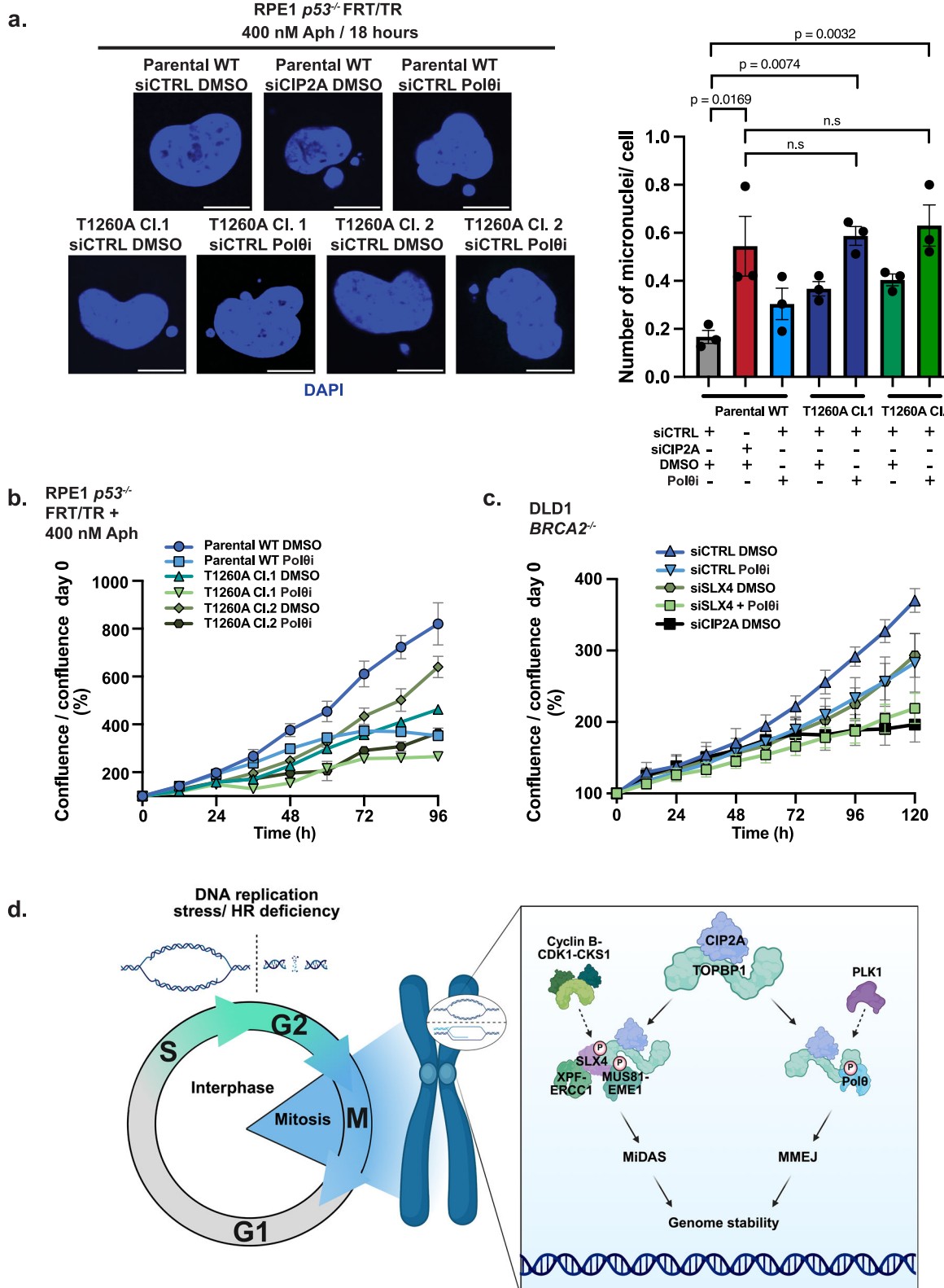

Together with their protein partners, CIP2A-TOPBP1 may aid in the regulation of DNA repair at distinct regions of the genome in mitosis. Given our observation that CIP2A colocalises with FANCD2 and γH2AX in mitosis in response to replication inhibition, this may provide insight into the distinct regions of the genome where CIP2A and TOPBP1 respond to enhance repair of persistent DNA replication stress-associated lesions[61,63,64,66,68]. Future research aimed at unravelling the precise dynamics and full complement of mitotic DNA repair

mechanisms, will be crucial for unlocking the therapeutic potential associated with targeting these pathways.

## Methods

### Cell lines and compound treatments

HCT116-TOPBP1-mAID-Clover cells were a generous gift from D.K. Cortez[47]. RPE1 *p53*[-/-] FRT/TR cells were kindly provided by S.P. Jackson[77]. RPE1 *p53*[-/-] *PAC*[-/-] WT and RPE1 *p53*[-/-] *PAC*[-/-] *BRCA1*[-/-] were a kind

**Fig. 7 | CIP2A-dependent orchestration of DNA repair is required for genome stability and cellular proliferation in the absence of BRCA2. a** Representative images and bar plot illustrating the number of micronuclei per RPE1 *p53*[-/-] FRT/TR WT or SLX4 T1260A knock in cell nucleus. Cells were treated with siCTRL or siCIP2A, 400 nM aphidicolin, in addition to either DMSO or 5 μM ART558 (Polθi) for 18 h (Parental WT (siCTRL + DMSO: *n* = 343, siCIP2A + DMSO: *n* = 394, siCTRL + Polθi: *n* = 372), SLX4 T1260A Cl.1 (siCTRL + DMSO: *n* = 385 and siCTRL + Polθi: *n* = 341) or SLX4 T1260A Cl.2 (siCTRL + DMSO: *n* = 399 and siCTRL + Polθi: *n* = 378) from three independent experiments). Statistical significance was determined using by one-way ANOVA, and the mean of each experiment is displayed as black dots and the mean of all three experiments is presented as bars with SEM. Scale bars are equivalent to 10 μm. **b** Proliferation analysis of RPE1 *p53*[-/-] FRT/TR WT and T1260A cells, performed using the Incucyte live cell analysis system. Cells were treated with DMSO or with 5 μM ART558 (Polθi) in the presence of 400 nM

aphidicolin. The mean confluence divided by confluence at day 0, from three independent experiments, is displayed with SEM. **c** Proliferation analysis of DLD1 *BRCA2*[-/-] cells using the Incucyte live cell analysis system. Cells were treated with siCTRL, siSLX4 or siCIP2A in the presence of DMSO or 5 μM ART558 (Polθi). The mean confluence divided by confluence at day 0, from three independent experiments, is displayed with SEM. **d** Model: Cells enter mitosis with DNA damage, under replicated DNA or recombination intermediates where the CIP2A-TOPBP1 complex is recruited. CDK1 phosphorylation of SLX4 at T1260 regulates interaction with TOPBP1 BRCT 1/2 facilitating SMX component recruitment and MiDAS/BIR to safeguard genome stability. PLK1 phosphorylation of POLQ in mitosis facilitates TOPBP1 interaction and MMEJ[5]. Loss of CIP2A impairs both pathways and leads to simultaneous deficiency in mitotic MiDAS/BIR and MMEJ promoting genome instability. Created in BioRender. Martin, P. (2025) https://BioRender.com/d5kygf2. Source data are provided as a Source Data file.

gift from S.M. Noordermeer[78]. RPE1 *p53*[-/-] FRT/TR, RPE1 *p53*[-/-] *PAC*[-/-] WT, RPE1 *p53*[-/-] *PAC*[-/-] *BRCA1*[-/-] and HEK293TN (RRID:CVCL_UL49) cell lines were cultured in Dulbecco's modified Eagle's medium (DMEM) supplemented with 10% foetal bovine serum (FBS). HCT116-TOPBP1-mAID-CLOVER cells were cultured in McCoy's 5 A medium supplemented with 10% FBS. DLD1 (RRID:CVCL_0248), DLD1 *BRCA2*[-/-] and SUM149PT *BRCA1*[-/-] (RRID:CVCL_3422) cells were kindly provided by C. Lord. DLD1 and DLD1 *BRCA2*[-/-] cells were cultured in RPMI-1640 supplemented 10% FBS with the addition of 2 mM L-glutamine. SUM149PT cells were grown in Ham's F-12 medium supplemented with 5% FBS, 10 μg/mL insulin, 0.5 μg/mL hydrocortisone. U2OS cells (RRID: CVCL_0042) stably integrated with EJ2 (RRID:Addgene_44025[65]), a gift from J. M. Stark and pBIR (RRID:Addgene_49807[59]), a gift from T. D. Halazonetis, were maintained in Dulbecco's modified Eagle's medium (DMEM) supplemented with 10% foetal bovine serum (FBS). All cell lines were maintained in penicillin/streptomycin antibiotics.

Where indicated cells were treated with 400 nM Aphidicolin (Sigma Aldrich, A4487), 5 μM ART558 (S9936, Selleckchem), 500 μM 3-indolacetic acid (Sigma Aldrich, I2886), 60 or 100 ng/ml Nocodazole (Sigma Aldrich, 487928), 7 or 5 μM RO3306 (Sigma Aldrich, 217699). For induction of the expression of proteins under tet repressor control, cells were treated with 10 ng/ ml (RPE1 *p53*[-/-] FRT/TR *SLX4*[-/-]) or 100 ng/ml (DLD1 *BRCA2*[-/-] and SUM149PT *BRCA1*[-/-]) doxycycline for 24 hours (Sigma Aldrich, D3447) where indicated.

Cell lines were regularly tested to confirm the absence of mycoplasma contamination using the MycoAlert Mycoplasma Detection Kit (Lonza, LT07-318).

## Plasmids

pcDNA5-Hygro-FRT/TO-eGFP-SLX4 was kindly provided by Dr. John Rouse. The SLX4 coding sequence was transferred to pcDNA5-Neo-FRT/TR (RRID: Addgene_41000; a gift from Dr. J. Mansfeld) via HiFi DNA assembly following manufacturer's guidelines (NEB, E2621S). pcDNA5-Neo-FRT/TO-eGFP-SLX4 T1260A plasmid was generated by subjecting the pcDNA5-FRT/TO-Neo-eGFP-SLX4 plasmid to site directed mutagenesis using the Q5 site directed mutagenesis kit (NEB, E0552S). pENTR223.1-POLQ (HsCD00353909) was obtained from the DNASU plasmid repository. The POLQ cDNA sequence was amplified from pENTR223.1-POLQ and an eGFP containing fragment was amplified from pcDNA5-FRT/TO-eGFP, the fragments were transferred into the pLVX-TET-ONE-Puro (Takara) backbone via HiFi DNA assembly (NEB, E2621S) to generate pLVX-TET-ONE-eGFP-POLQ. For the generation of an inducible minimal SLX4 T1260 encompassing fragment, a DNA sequence encoding an amino acid stretch spanning the surrounding region of SLX4 T1260 was amplified and cloned into pLVX-TET-ONE using HiFi assembly (NEB, E2621S). pOG44 (Thermo Scientific, V6005-20) was used for Flp-In recombination. pVLX-TET-ONE-PURO (Takara), pMD2.G (RRID:Addgene_12259) and psPAX2 (RRID:Addgene_12260) were used for lentiviral work. AIO-mCherry

(RRID:Addgene_74120) kindly provided by S.P Jackson, was used as backbone for CRISPR knock-out of SLX4. Plasmid constructs encoding GFP-TOPBP1 in the pIRESneo2 backbone (Clontech) were obtained from T.D Halazonetis[79], truncation mutants described previously were used in this study[80] and BRCT1/2 mutant constructs were generated by Q5 site directed mutagenesis (NEB, E0552S). The eSPCAS9.1-ATP1A1-G3 plasmid was a kind gift from Y. Doyon. Plasmids with the pCDNA5, pIRESneo2 or eSPCAS9.1-ATP1A1-G3 backbone were transformed and amplified in dH5α bacteria (Thermo Scientific, EC0112) and those used for lentiviral work were transformed and amplified in NEB stable bacteria (NEB, C3040). For bacterial cell expression a pGEX-6P-1 plasmid containing TOPBP1 fragment coding sequences was used. For insect cell expression, a single construct containing CDK1, Cyclin B and Cks1 was produced using the biGBac system[81].

## Inducible cell line generation

To facilitate Flp-In recombination, 2 μg of The Flp-Recombinase expressing plasmid pOG44 was co-transfected with 6 μg pcDNA5-FRT/TO-eGFP SLX4 WT/T1260A into RPE1 *p53*[-/-] FRT/TR cells, using Lipofectamine 3000 (Invitrogen, L3000001) following manufacturers guidelines. Cells were incubated for 24 h at 37 °C with 5% $CO_2$ followed by addition of 1 mg/ml G418 (Invivogen, ant-gn) to facilitate selection. G418 was replaced every 72 h. After 14 days 100 ng/ml of doxycycline was added to the media, and the cells were incubated a further 24 h at 37 °C with 5% $CO_2$. eGFP-expressing cells were sorted by fluorescence-activated cell sorting (FACS) using the BD Symphony S6 Cell Sorter. Cells were maintained in 500 μg/ ml G418 containing medium during continuous culture.

To facilitate lentiviral transduction and integration of inducible eGFP-POLQ, 10 μg of pLVX-TET-ONE-eGFP-POLQ was Co-transfected with 1.8 μg of pMD2.G (RRID:Addgene_12259) and 4.4 μg of psPAX2 (RRID:Addgene_12260) using Lipofectamine 3000 (Invitrogen, L3000001) following manufacturers guidelines into HEK293TN cells. Cells were incubated for 72 h at 37 °C with 5% $CO_2$ in antibiotic free complete media. At which point the medium was collected, centrifuged at 450 x g for 5 min and filtered using a 0.45 μM syringe filter. The filtered lentivirus−containing medium was then aliquoted and stored at −80 °C. DLD1 BRCA2[-/-] cells were subsequently transduced by the addition of lentivirus−containing medium and 6 μg/ml Polybrene (Sigma Aldrich, TR-1003). Cells were incubated 24 h at 37 °C with 5% $CO_2$ in antibiotic-free medium. Medium was then replaced with medium containing 4 μg/ml puromycin (Invivogen, ant-pr) and replaced every 72 h. After 10 days 100 ng/ ml doxycycline was added to the medium and cells were incubated a further 24 h at 37 °C in 5% $CO_2$ followed by FACS sorting of eGFP positive cells. Cells were maintained in 4 μg/ ml puromycin containing medium during continuous culture.

To facilitate lentiviral transduction and integration of an inducible minimal SLX4 T1260 encompassing fragment, 10 μg of pLVX-TET-ONE-SLX4 T1260 (fragment) plasmid or pLVX-TET-ONE (Empty vector) was

co-transfected with 1.8 μg of pMD2.G and 4.4 μg of psPAX2 using Lipofectamine 3000 (Invitrogen, L3000001) following manufacturers guidelines into HEK293TN cells. Cells were incubated for 72 h at 37 °C with 5% $CO_2$ in antibiotic-free medium. At which point the medium was collected, centrifuged at 450 x g for 5 min and filtered using a 0.45 μM syringe filter. The filtered lentivirus−containing medium was then aliquoted and stored at −80 °C. DLD1 *BRCA2*$^{-/-}$ and SUM149PT *BRCA1*$^{-/-}$ cells were subsequently transduced by the addition of lentivirus− containing medium and 6 μg/ml Polybrene (Sigma Aldrich, TR-1003). Cells were incubated 24 h at 37 °C with 5% $CO_2$ in antibiotic free medium. Medium was then replaced with medium containing 4 μg/ml (DLD1 *BRCA2*$^{-/-}$) or 1 μg/ml (SUM149PT *BRCA1*$^{-/-}$) puromycin (Invivogen, ant-pr) and replaced every 72 h. After 10 days 100 ng/ ml doxycycline was added to the medium and cells were incubated a further 24 h at 37 °C in 5% $CO_2$ followed by FACS sorting of eGFP positive cells. Cells were maintained in 4 μg/ml (DLD1 *BRCA2*$^{-/-}$) or 1 μg/ml (SUM149PT *BRCA1*$^{-/-}$) puromycin-containing medium during continuous culture.

### Generation of SLX4 knock-out cells by CRISPR/Cas9 nickase

To facilitate generation of SLX4 knock-out cells, the following guide RNA sequence containing complementary oligonucleotides with overhangs compatible with a BbsII or BsaI restriction sites were purchased from IDT.

SLX4gRNA1F Sequence: 5′-ACCG TG TCC CAA AGG ATC CTC AAG-3′

SLX4gRNA1R Sequence: 5′-AAAC CT TGA GGA TCC TTT GGG ACA-3′

SLX4gRNA2F Sequence: 5′-ACCG GT AGG ACC AAT TGT GCT GTG-3′

SLX4gRNA2R Sequence: 5′-AAAC CA CAG CAC AAT TGG TCC TAC CGG T-3′

Subsequently, the oligonucleotides were phosphorylated and annealed by combining 1 μl each of 100 μM complementary oligonucleotides with 1 μl of 10x T4 ligation buffer (NEB, B0202S) with 0.5 μl of T4 PNK (NEB, M0201S) in a total reaction volume of 10 μl. Subsequently, the reaction mixture was incubated at 37 °C for 30 minutes followed by 95 °C for 5 min and gradient cooled at −5 °C/ minute to 25 °C. The annealed and phosphorylated oligonucleotides were then dilute 1:250 in dH$_2$O. Then, 100 ng of AIO-mCherry (RRID:Addgene_74120) kindly provided by S.P. Jackson, was digested and ligated by combining 1 μl of diluted and annealed phosphorylated oligonucleotide with 1 mM DTT, 1 mM ATP, 0.5 μl of FastDigest BbsI (Thermo Scientific, ER1011), 0.25 μl of T7 DNA ligase (NEB, M0318S) in a final reaction volume of 10 μl. The reaction mixture was then incubated 37 °C for 5 min then 23 °C for 5 min for six cycles. The reaction mixture was then treated with PlasmidSafe ATP-dependent DNase by combining 5.5 μl of the previous ligation reaction with 1 mM ATP and 0.5 μl of ATP-dependent DNase (Biosearch Technologies, E3101K) in a final reaction volume of 7.5 μl. Subsequently, 3 μl of the treated ligation product was transformed into DH5-Alpha competent cells, plated on LB-Agar/ 50 μg/ml ampicillin containing plates and incubated overnight at 37 °C. Colonies were picked and grown overnight in 3 ml LB-broth containing 50 μg/ml ampicillin. Plasmid was purified using the GeneJET Plasmid Miniprep Kit (Thermo Scientific, K0502). The purified plasmid was then digested, and the second pair of annealed and phosphorylated oligonucleotides were ligated as above, but instead using BsaI-HF (NEB, R3733S), followed by PlasmidSafe treatment and transformation into DH5-Alpha cells. Colonies were picked, grown overnight in LB broth as above, followed by generation of glycerol stocks and isolation of plasmid DNA. The retrieved plasmid DNA was then sequenced by Source Bioscience using 3.2 pmol of the following sequencing primer:

AIOseq: 5′-CTTGATGTACTGCCAAGTGGGC-3′

Glycerol stocks corresponding to plasmids confirmed to contain both guide RNA sequences were then streaked on LB/Agar plates incubated overnight at 37 °C. Colonies were picked and grown overnight in 50 ml of LB-broth containing 50 μg/ ml ampicillin. Plasmid DNA was then purified using the ZymoPURE II Plasmid Midiprep Kit (Zymo Research, D4201) and sequenced by Source Bioscience using the above AIOseq sequencing primer to confirm sequence. RPE1 *p53*$^{-/-}$ FRT/TR cells were transfected with 1.5 μg of plasmid DNA using Lipofectamine 3000 transfection reagent following manufacturer's instructions (Invitrogen, L3000001). After 24 h incubation at 37 °C with 5% $CO_2$ cells were trypsinised and 10,000 mcherry positive cells were bulk fluorescence activated cell sorted using BD FACSymphony S6 Cell Sorter into a well of a 6 well tissue culture dish, then incubated for a further 7 days at 37 °C with 5% $CO_2$. Subsequently, single mCherry negative cells were sorted into 96 well tissue culture plates using the BD FACSymphony S6 Cell Sorter into 96 well tissue culture plates and incubated at 37 °C with 5% $CO_2$ for 14 days. Colonies were then transferred to wells of a 6 well plate and incubated for a further 5 days at 37 °C with 5% $CO_2$. At this stage, cells were trypsinised and 90% of the cell suspension was isolated for western blot analysis for loss of SLX4 expression at protein level while complete medium was added to the remaining 10% and incubated at 37 °C with 5% $CO_2$. Once loss of SLX4 protein was detected by western blot analysis, putative clones were isolated and expanded, followed by genomic DNA extraction using the Monarch genomic DNA purification kit (NEB, T3010). Using forward (5′- CAAC-CACCACCACTACCTAAC-3′) and reverse (5′- GCGAAACCCTGTCTC-TACTAAA-3′) primers flanking the edit site and isolated genomic DNA, PCR was carried out using Q5 high-fidelity polymerase (NEB, M0491) following the manufacturer's instructions. The PCR product was resolved on a 1% agarose/ 1xTBE gel, followed by gel extraction using Zymoclean Gel DNA Recovery Kit (Zymo research, D4002). The purified product was then submitted to Source BioScience for sanger sequencing analysis using the forward PCR primer as a sequencing primer.

### Generation of SLX4 T1260A knock-in cells by CRISPR/Cas9 nuclease marker-free co-selection

RPE1 *p53*$^{-/-}$ FRT/TR SLX4 T1260A cells, were generated by CRISPR Cas9-mediated knock-in via marker-free co-selection, as previously described[82]. Briefly, two guide RNA sequences were chosen proximal to the SLX4 T1260 encoding sequence and assessed for their on-target and off-target efficiency using Benchling (benchling.com) and CRIS-pick (https://portals.broadinstitute.org/gppx/crispick/public) guide RNA design tools. Subsequently, a 200 bp single-stranded ALT-R HDR Donor oligo (IDT) was designed to introduce the SLX4 T1260A point mutation and silent mutations upstream of the PAM sequences to introduce a KPNI restriction endonuclease site. In addition, silent mutations were included within the guide RNA recognition sequences to inhibit further cleavage by the Cas9 sgRNA complex. This single-stranded HDR donor oligo was then purchased from IDT in addition to a single-stranded ALT-R HDR Donor oligo ATP1A1 repair template used to promote editing and resistance to ouabain[82,83]. The eSPCAS9.1-ATP1A1-G3 plasmid (RRID:Addgene_86613), kindly provided by Y. Doyon, was modified previously to include an additional third U6-gRNA cassette with BsmBI sites to create plasmid PLJM787[83]. Complementary single-stranded DNA oligos were purchased from IDT to include overhangs compatible with cloning into BbsI and BsmBI sites located proximal to the available encoded gRNA scaffolds.

Guide RNA 1 oligo pair:
5′- CACCGACGGCTTCTGCTGGCCAGCG − 3′
5′- AAACCGCTGGCCAGCAGAAGCCGTC − 3′
Guide RNA 2 oligo pair:
5′- CACCGAGAACAGTCACGGCTTCTGC-3′
5′-AAACGCAGAAGCCGTGACTGTTCTC-3′
T1260A repair template:

5′-CTCTTTGGGCAGGA-GAGGGGGCTCCCTGGCTGTTCTGTGACCGTGA-GAGCAGCCCCAGCGAGGCCAGCACCACAGA-CACCTCGTGGCTGGTACCCGCtgcaCCGcttGCtagca-gaagcCGTGACTGTTCTTCCCA-GACCCAAATCAGCAGCCTCAGGAGCGGGCTGGCCGTGCAGGCGGT-GACTCAGCACACGCCCAGGG-3′

ATP1A1 repair template:

5′-CAATGTTACTGTGGATTGGAGCGATTCTTTGTTTCTTGGCTTA-TAGCATCAGAGCTGCTACAGAA-GAGGAACCTCAAAACGATGACGTGAGTTCTGTAATTCAGCA-TATCGATTTGTAGTACACATCAGATATCTT-3′

The above RNA encoding oligonucleotides for knock-in were phosphorylated and annealed, followed by cloning into PLJM787, following the same method described for preparation of AOI-mCherry described in the 'Generation of SLX4 knock-out cells by CRISPR/CAS9 nickase' methods.

Subsequently, RPE1 $p53^{-/-}$ FRT/TR cells were electroporated using the Neon Transfection System. Per electroporation shot 300,000 cells were mixed with 500 ng of PLJM787 plasmid, 2pmol of ATP1A1 ALT-R-HDR-Donor oligo (IDT) repair template and 6pmol of the T1260A ALT-R-HDR-Donor oligo (IDT) repair template. Cells were then incubated at 37 °C at 5% $CO_2$. Four days after electroporation, cells were incubated with media containing 250 nM ouabain, ATP1A1 inhibitor (Sigma Aldrich, Cat. No: O3125). After 7 days, ouabain-resistant clones were picked and transferred by trypsinisation into 96-well plates. Each picked clone was duplicated in an additional 96-well plate. While one of the duplicated populations remained in continuous culture, the other duplicated populations were processed for preparation of genomic DNA (gDNA). To purify gDNA cells were washed with 100 μl 1x PBS and then replaced with 100 μl lysis buffer (1 mM $CaCl_2$, 3 mM $MgCl_2$, 1 mM EDTA, 10 mM Tris pH 7.4, 1% Triton X-100, supplemented with fresh 0.2 mg/ml Proteinase K (NEB, P8107S)). Cells were incubated with lysis buffer at 37 °C for 10 min and subsequently transferred to a 96-well PCR plate and sealed with a microplate seal. The plate was transferred to a thermocycler and incubated at 65 °C for 10 min followed by 95 °C for 15 min. The plate was then stored at −20 °C.

To PCR amplify the potentially edited region of SLX4, 2 μl of the gDNA lysate was used for a 25 μl standard PCR reaction using Phusion high fidelity polymerase, following manufacturer's instructions (NEB, M0530S), using forward (5′-AGCAGGAGGATGAGGGGG-3′) and reverse primers (5′-CCGCCTGCACGGCCA-3′) flanking the edited genomic DNA sequence. Following this, 1 μl of KpnI restriction endonuclease was added to the PCR reaction and incubated for 18 h at 37 °C. Incorporation of the repair template was verified with restriction digest of the amplified DNA amplicons after 1% agarose/1xTBE gel electrophoresis analysis. Clones that demonstrated restriction digestion were subjected to a further round of the above PCR followed by agarose gel electrophoresis and gel purification using the Zymoclean Gel DNA Recovery Kit (Zymo research, D4021). The purified product was then submitted for sanger sequencing analysis by Source Bioscience, using the above forward PCR primer as a sequencing primer, to confirm knock-in. Clones that were positive for editing were then expanded from matching continuous cultures under 250 nM ouabain selection, then subjected to single cell sorting into a 96-well tissue culture plate containing 100 μl complete medium, facilitated by the BD FAC-Symphony S6. Single clones were incubated for 10 days at 37 °C in 5% $CO_2$, followed by trypsinisation and transfer to a 10 cm tissue culture dish. Cells were incubated for a further 5 days before harvesting 500,000 cells for gDNA extraction using the Monarch gDNA extraction kit (NEB, T3010S) while maintaining the remaining cells in continuous culture. The gDNA was subjected to Phusion polymerase (NEB, M0530S) PCR as above and the PCR product was then analysed by 1% agarose/1xTBE gel electrophoresis. The PCR product was confirmed to be a single product through gel visualisation using a UV transilluminator and was then purified using the Zymoclean Gel DNA Recovery Kit (Zymo research, D4021). The gel purification product was then analysed by sanger sequencing (Source Bioscience) using the forward primer as a sequencing primer, confirming the desired genomic edit.

## Generation of CIP2A knock-out cells by CRISPR/Cas9 ribonucleoprotein delivery

RPE1 $p53^{-/-}$ FRT/TR cells were subjected to CRISPR via Cas9/sgRNA ribonucleoprotein delivery. Alt-R™ S.p. Cas9-GFP V3 (IDT, 10008100), Alt-R™ CRISPR-Cas9 tracrRNA, ATTO™ 550 (IDT, 1075927) and Alt-R™ CRISPR-Cas9 crRNA targeting exon 1 of the CIP2A locus (CIP2A crRNA: GGAGTCCATTGCACCGGCCGCGG) were acquired from IDT. Subsequently, crRNA and tracrRNA were resuspended in nuclease free duplex buffer at 100 μM. Then 1 μl of each of 100 μM crRNA and tracRNA were diluted in 98 μl of nuclease free duplex buffer, mixed and denatured at 95 °C for 5 min followed by gradient cooling to room temperature, to facilitate crRNA and tracrRNA annealing to form sgRNA. S.p. Cas9 Nuclease V3 was vortexed vigorously and diluted to 1 μM in Optimem (Gibco, 31985062). At which point 24 μl of 1 μM sgRNA was combined with 24 μl of 1 μM Cas9 and 20 μl of Lipofectamine RNAimax (Invitrogen, 13778075) in 732 μl of Optimem, to generate an 800 μl RNP transfection mixture. The RNP transfection mixture was incubated at room temperature for 20 min to facilitate complex formation. At this time, RPE1 $p53^{-/-}$ FRT/TR cells were trypsinised and diluted to $4 \times 10^5$/ml. Subsequently, 800 μl of the RNP transfection mixture was added to a well of a 6 well tissue culture dish, followed by addition of 1600 μl of the diluted cell suspension. The cells and RNP mixture were briefly mixed by pipetting and incubated at 37 °C with 5% $CO_2$ for 24 h. The cells were then trypsinised and eGFP and ATTO550 double-positive cells were bulk cell fluorescence activated cell sorted using the BD FACSymphony S6 Cell Sorter into a well of a 6 well tissue culture plate and then incubated at 37 °C with 5% $CO_2$ for 7 days. After, the cells were subjected to single cell sorting for eGFP/ATTO550 negative cells into wells of a 96-well tissue culture plate using the FACSymphony S6 Cell Sorter and incubated for a further 14 days. Colonies were transferred to individual wells of 6 well plates and incubated at 37 °C with 5% CO2 for a further 5 days. Then cell populations were trypsinised and 90% of the cell suspension was isolated by centrifugation and processed by western blotting to detect CIP2A loss at protein level.

## Cellular fractionation

Cell pellets stored at −80 °C were thawed on ice, followed by addition of ten volumes to 1 NIB-250 (15 mM Tris-HCL (pH 7.5), 60 mM KCl, 15 mM NaCl, 5 mM MgCl2, 1 mM CaCl2, 250 mM sucrose) containing 1 mM DTT, 1x cOmplete, EDTA-free Protease Inhibitor +0.3% NP. The cells were mixed by pipetting gently and then incubated on ice for 10 min. The cell pellet was then centrifuged at 400–600 x g for 5 min at 4 °C. The supernatant was transferred to a separate tube (cytoplasmic fraction). The nuclei pellet was then resuspended in ten volumes of 10:1 NIB-250 without detergent containing 1 mM DTT and 1x cOmplete, EDTA-free Protease Inhibitor. The nuclei were then pelleted by centrifugation at 400–600 x g for 5 min at 4 °C and the supernatant was discarded. The pellet was then resuspended 7 volumes relative to pellet volume of hypotonic buffer (3 mM EDTA, 0.2 mM EGTA containing 1 mM DTT and 1x cOmplete, EDTA-free Protease Inhibitor) and incubated on ice for 30 min, vortexing every 5 min. The insoluble chromatin fraction was collected by centrifugation for 4 min at 1700 x g. The supernatant, was retained, containing the nucleoplasm fraction. The chromatin pellet was washed with hypotonic buffer containing 1 mM DTT and protease inhibitors and the above centrifugation was repeated. The supernatant was discarded and the pellet retained and used as starting material for immunoprecipitation experiments.

## Immunoblotting

Cell lysis was carried out in 1x RIPA buffer (Sigma-Aldrich), supplemented with 1× SIGMAFAST protease inhibitors (Sigma-Aldrich), and 1xPhosStop phosphatase inhibitors (Roche), on ice for 15 min followed by centrifugation at $1.4 \times 10^4$ x g for 20 min and then by collection of supernatant containing cell lysate. Cell lysates were prepared in SDS loading buffer (2% SDS, 10% (v/v) glycerol, 2% 2-Mercaptoethanol and 62.5 mM Tris–HCl, pH 6.8) followed by boiling at 95 °C for 10 min. Protein concentrations were determined by the BCA or by the Bradford assay. Samples were resolved by SDS-PAGE and transferred to nitrocellulose membrane followed by blocking in 5% low fat milk in 1× TBS/ 0.1% Tween-20 for 1 h at room temperature. The Broad Range Prestained Protein Marker (Proteintech, PL00002) was used throughout this study as a molecular weight marker for immunoblots. Membranes were washed 3 × 5 min in 1× TBS/ 0.1% Tween-20 and incubated overnight at 4 °C in the indicated primary antibodies in 5% low fat milk in 1x TBS/ 0.1% Tween-20. Membranes were subsequently washed 3 × 5 min in 1× TBS/ 0.1% Tween-20 and incubated in 5% low fat milk in 1× TBS/ 0.1% Tween-20 containing secondary antibodies for 1 h at room temperature. Membranes were subsequently washed 3 × 5 min in 1× TBS/ 0.1% Tween-20 and developed using Immobilon Western HRP Substrate (Millipore, WBKLS0S00) and imaged using the Azure C280, 300 or 600 instruments (Azure biosystems).

Primary antibodies used were: α-Tubulin (Sigma-Aldrich Cat# T5168, RRID:AB_477579, 1:100000), β-Actin (Sigma-Aldrich Cat# A2066, RRID:AB_476693, 1:1000), MCM2 (Abcam Cat# ab4461, RRID:AB_304470 1:2000), H3 (Abcam Cat# ab1791, RRID:AB_302613, 1:50,000), Vinculin (Thermo Fisher Scientific Cat# MA5-11690, RRID:AB_10976821, 1:1000), GFP (Roche Cat# 11814460001, RRID:AB_390913, 1:500), MUS81 (Santa Cruz Biotechnology, (Santa Cruz Biotechnology Cat# sc-47692, RRID:AB_2147129,1:500), ERCC1 (Santa Cruz Biotechnology Cat# sc-17809, RRID:AB_2278023,1:500), SLX4 (MRC-PPU Cat# S714C, RRID:AB_2752254, 1:500), SLX4 pT1260 (Genscript). TOPBP1 (Santa Cruz Biotechnology Cat# sc-271043, RRID:AB_10610636,1:500), MDC1 (Abcam Cat# ab11171, RRID:AB_297810, 1:1000), CIP2A (Santa Cruz Biotechnology Cat# sc-80659, RRID:AB_1121640, 1:500), POLD3 (Abnova Cat# H00010714-M01, RRID:AB_606803, 1:500), TOP3A (Proteintech Cat# 14525-1-AP, RRID:AB_2205881, 1:3000), BLM (Bethyl Cat# A300-110A, RRID:AB_2064794, 1:500), BRCA1 (Millipore Cat# OP92, RRID:AB_2750876, 1:200), BRCA2 (Cell Signalling Technology Cat# 10741, RRID:AB_2797730, 1:1000).

Secondary antibodies used were anti-mouse IgG-HRP (Dako, P0447, 1:2000), anti-rabbit IgG-HRP (Dako, P0448, 1:5000) and anti-sheep IgG-HRP (Abcam Cat# ab6747, RRID:AB_955453, 1:500).

## Phosphatase treatment of western blot membranes

After western blot transfer, the membrane was washed 3 x for 5 min n 1x TBS/ 0.1% Tween-20 (TBS-T). The western blot membrane was divided in half by cutting the membrane vertically. The divided membrane was then separated into two 50 ml centrifuge tubes containing 10 ml phosphatase reaction buffer (NEB, 1x rCutSmart buffer B6004SVIAL) supplemented with 1 mM MgCl₂. To one tube 1 µg/ml of lambda protein phosphatase and 1 unit/ml of shrimp alkaline phosphatase (rSAP; NEB, M0371) was added. The tubes were then placed in a rotating hybridisation oven at 30 °C for 1 h followed by 2 h at 37 °C. The membranes were then washed once in TBS-T, followed by blocking with 5% low fat milk/ TBS-T for 1 h. Membranes were then incubated with antibodies and developed as described in the above immunoblotting methods.

## RNA interference

To facilitate transient depletion of TOPBP1, SLX4, MDC1, BRCA1, BRCA2, Polθ, POLD3 or CIP2A, cells were transfected with 30 pmol of oligonucleotides using Lipofectamine RNAimax transfection reagent

(Invitrogen, 13778100), according to the manufacturer's reverse transfection protocol. siRNA targeting luciferase was used as a non-targeting control siRNA. For experiments involving depletion of MDC1, BRCA2, Polθ, POLD3 or CIP2A a SMARTpool of four oligonucleotides was used. For experiments involving SLX4 siRNA, SLX4 (1) and (2) oligonucleotides were mixed at equimolar ratios in advance of use. First pulse of siRNA was followed with second pulse after 24 h, following manufacturer's forward transfection protocol and all subsequently experiments were performed from 72 h after the first siRNA pulse.

Small interfering RNA used in this study were as follows:

siRNA targeting luciferase (siCTRL/siLUC): 5′-CGTACGCGGAA-TACTTCGA-3′

KIAA1524 (CIP2A) ID: L-014135-01-0005 (Horizon) consisting of:

CIP2A (1): 5′-ACAGAAACUCACACGACUA-3′
CIP2A (2): 5′-GUCUAGGAUUAUUGGCAAA-3′
CIP2A (3): 5′-GAACAAGGUUGCAGAUUC-3′
CIP2A (4): 5′-GCAGAGUGAUAUUGAGCAU-3′
SLX4 (1): 5′-GCACAAGGGCCCAGAACAA-dT-dT-3′
SLX4 (2): 5′-GCACCAGGUUCAUAUGUA-dT-dT-3′
TOPBP1: 5′-GUAAAUAUCUGAAGCUGUAUU-3′
POLD3: L-026692-01-0005 (Horizon)
Polθ: L-015180-01-0005 (Horizon)
MDC1: L-003506-00-0005 (Horizon)
BRCA1: 5′-ACCAUACAGCUUCAUAAAUAA −3′
BRCA2: L-003462-00-0005 (Horizon)

## Immunofluorescence microscopy

For detection of EdU incorporation in mitosis and the analysis of recruitment of TOPBP1, CIP2A, MUS81, ERCC1, TOPBP1-mAID-Clover, eGFP tagged SLX4 and eGFP tagged Polθ in RPE1 *p53⁻/⁻* FRT/TR and DLD1 *BRCA2⁻/⁻* cells, respectively, cells were grown on 13 mm 1.5 thickness coverslips and treated as described. Subsequently cells were simultaneously fixed and permeabilised in PTEMF buffer (20 mM PIPES pH 6.8, 10 mM EGTA, 0.2% Triton X-100, 1 mM MgCl2 and 4% formaldehyde) for 10 minutes at room temperature. For detection of micronuclei, cells were fixed in 250 mM HEPES pH 7.5, 0.1% Triton X-100, 4% PFA in PBS for 20 minutes at 4 °C.

Coverslips were then incubated for 5 min in 1x PBS with three buffer changes followed by incubation with 0.5% Triton X-100/ PBS for 10 min at room temperature. Coverslips were then incubated for 5 minutes in 1x PBS with three buffer changes. For EdU detection coverslips were transferred to a humidified chamber and incubated in EdU click it reaction buffer containing Alexafluor 488 azide for 1 h at room temperature followed by three 5 min washes in 1x PBS. Coverslips were then incubated in 5% FBS/PBS for 1 h followed by overnight incubation at 4 °C with primary antibody diluted in 5% FBS/PBS. Coverslips were then washed 3 × 5 min in 1x PBS followed by 1 h incubation at room temperature in 5% FBS/PBS containing secondary antibodies. Coverslips were incubated a further three times for 5 min in 1x PBS. Coverslips were then mounted on superfrost microscopy slides using Vectashield mounting medium with DAPI (Vector laboratories, H1200).

Alternatively, for the detection of colocalisation of γH2AX, MUS81/SLX4/CIP2A, FANCD2/ CIP2A, detection of recruitment of γH2AX, 53BP1, MDC1, SLX4, in interphase and mitosis in either DLD1, RPE1 *p53⁻/⁻* FRT/TR, RPE1 *p53⁻/⁻* PAC⁻/⁻ WT or BRCA1⁻/⁻ or the detection of recruitment of SLX4 or MUS81 in RPE1 *p53⁻/⁻* PAC⁻/⁻ BRCA1⁻/⁻ the following procedure was used. Cells were fixed in 2% BSA/PBS for 15 min at room temperature, followed by 3 × 5 min washes in 1xPBS. Coverslips were then incubated in 0.5% Triton-X/PBS for 10 min at room temperature, followed by a further 3 × 5 min 1xPBS washes. Coverslips were then incubated in blocking buffer (2% BSA, 0.05% Tween-20) for 1 h at room temperature, followed by incubation in respective primary antibodies overnight at 4 °C. Coverslips were then washed 3x for 5 min in 1x PBS followed by incubation for 1 h at room temperature in

blocking buffer containing secondary antibodies. Coverslips were then washed 3x for 5 min in 1x PBS and mounted with Vectashield PLUS antifade mounting medium with DAPI (Vector laboratories, H2000).

Primary antibodies used for immunofluorescence microscopy were: mouse anti-MUS81 (Santa Cruz Biotechnology Cat# sc-53382, RRID:AB_2147138,1:500), mouse anti-ERCC1 (Santa Cruz Biotechnology Cat# sc-17809, RRID:AB_2278023, 1:500), mouse anti-SLX4 (Abnova Cat# H00084464-B01P, RRID:AB_1673069, 1:500), mouse anti-TOPBP1 (Santa Cruz Biotechnology Cat# sc-271043, RRID:AB_10610636, 1:500), rabbit anti-TOPBP1 (Bethyl Cat# A300-111A, RRID:AB_2272050,1:500), mouse anti-MDC1 (Abcam Cat# ab50003, RRID:AB_881103, 1:500), mouse anti-CIP2A (Santa Cruz Biotechnology Cat# sc-80659, RRID:AB_1121640, 1:500), rabbit anti-CIP2A (Proteintech Cat# 23199-1-AP, RRID:AB_2918079,1:500), rabbit anti-FANCD2(Novus Cat# NB 100-182, RRID:AB_350110), mouse anti γH2AX (Millipore Cat# 05-636, RRID:AB_309864) or rabbit anti γH2AX (Abcam Cat# ab81299, RRID:AB_1640564) or GFP-Booster Alexa Fluor® 488 (ChromoTek Cat# gb2AF488-10, RRID:AB_2827573, 1:500).

Secondary antibodies used for immunofluorescence microscopy were: Alexa Fluor® 647 AffiniPure™ Donkey Anti-Mouse IgG (H + L) (Jackson ImmunoResearch Labs Cat# 715-605-150, RRID:AB_234086, 1:400), Donkey anti-Rabbit IgG (H + L) Highly Cross-Adsorbed Secondary Antibody, Alexa Fluor™ 647 (Molecular Probes Cat# A-31573, RRID:AB_2536183,1:400), Donkey anti-Mouse IgG (H + L) Highly Cross-Adsorbed Secondary Antibody, Alexa Fluor™ Plus 647 (Thermo Fisher Scientific Cat# A32787, RRID:AB_2762830, 1:500), Donkey anti-Mouse IgG (H+L) Highly Cross-Adsorbed Secondary Antibody, Alexa Fluor 488 (Molecular Probes Cat# A-21202, RRID:AB_141607, 1:400) or Donkey anti-Rabbit IgG (H + L) Highly Cross-Adsorbed Secondary Antibody, Alexa Fluor™ 488 (Thermo Fisher Scientific Cat# A-21206, RRID:AB_2535792, 1:400).

Images were acquired using a Zeiss Axio Observer Z1 Marianas™ Microscope attached with a CSU-SoRa spinning disk unit using either a Photometrics Kinetix sCMOS camera built by Intelligent Imaging Innovations (3i) or a Zeiss Axio Observer Z1 Marianas™ Microscope attached with a CSUX1 spinning disk unit and Hamamatsu Flash 4 CMOS camera built by Intelligent Imaging Innovations (3i) or using an Opera Phenix Plus high content imaging system (Revvity). In each case 405, 488 and 640 laser lines were used in addition to DAPI, GFP/FITC, iRFP/CY5 emission filters.

Quantification was carried out using FIJI (ImageJ, RRID:SCR_003070) software, CellProfiler (Broad Institute), Harmony (Revvity) or using Aivia (Leica Microsystems). Images of mitotic cells were analysed in 3D with maximum Z projections used for representative images in the manuscript to assist with visualisation.For interphase cells 2D images were analysed.

### EdU labelling
RPE1 $p53^{-/-}$ FRT/TR cells were grown on 13 mm coverslips with 1.5 thickness and incubated with 400 nM aphidicolin in complete medium for 18 h at 37 °C in 5% $CO_2$. Aphidicolin was then washed out by two media replacements followed by a third media replacement containing 20 μM EdU and 60 ng/ml nocodazole, followed by incubation at 37 °C in 5% $CO_2$ for a further 30 min. DLD1 WT and DLD1 $BRCA2^{-/-}$ cells were grown on coverslips for 24 hours at 37 °C in 5% $CO_2$ followed by incubation with medium containing 20 μM EdU for 30 min. Cells were then fixed in PTEMF buffer at room temperature for 10 min, followed by three 5 min 1x PBS washes. Subsequently, coverslips were incubated for 10 minutes in 0.5% Triton-X/PBS at room temperature, followed by three 5 min PBS washes. Coverslips were then mounted using Vectashield mounting medium containing DAPI (Vector laboratories, H1200).

Images were acquired using a Zeiss Axio Observer Z1 Marianas™ Microscope attached with a CSU-W spinning disk unit using either a Hamamatsu Flash 4 CMOS camera or a Photometrics Prime 95b

sCMOS camera built by Intelligent Imaging Innovations (3i). During capture, 405 and 488 laser lines were used in addition to DAPI and GFP/FITC emission filters. Quantification was carried assisted by visualisation using FIJI (ImageJ, RRID:SCR_003070) software.

### Cellular proliferation analysis by live cell imaging
For proliferation analysis of DLD1 $BRCA2^{-/-}$ cells after treatment with siRNA as indicated, 72 h post-transfection of the first pulse of siRNA, 1500 cells were seeded per well in a PhenoPlate 96-well optically clear tissue culture plate (Revvity, 6055302). Cells were evenly distributed within the well by pipetting up and down to mix, then incubated at 37 °C with 5% $CO_2$ for 24 h. The following day the media was aspirated and replaced with 200 μl of RPMI 1640 medium containing standard antibiotics, 10% FBS and 2 mM L-glutatmine. Plates were then transferred to an Incucyte S3 Live Cell Analysis System (Sartorius) imaging system, and phase images were captured every 12 h for a total of 120 h. Subsequently, proliferation was analysed using the Incucyte analysis software, normalising to confluence (%) at the assay start point. Data was exported to Microsoft Excel and values were plotted in GraphPad Prism 10.

To analyse cellular proliferation of RPE1 $p53^{-/-}$ FRT/TR and RPE1 $p53^{-/-}$ FRT/TR SLX4 T1260A Cl.1 and 2 cells, 500 cells per well of a 96 well μclear plate (Griener Bio-One, 655096) were seeded in a final volume of 100 μl of DMEM medium containing standard antibiotics and 10% FBS. Cells were mixed by pipetting to ensure even distribution and then incubated 24 h at 37 °C with 5% $CO_2$. Then 2x working dilutions of DMSO, ART558, aphidicolin + DMSO and aphidicolin + ART558 were prepared in complete medium and then 100 μl of drug dilution was added to each well of the plate to achieve final working concentrations of 5 μM ART558 and 400 nM aphidicolin. For the analysis of RPE1 $p53^{-/-}$ FRT/TR and RPE1 $p53^{-/-}$ FRT/TR SLX4 T1260A Cells after treatment with siBRCA1 or siBRCA2, cells were treated with siRNA as indicated. After 72 h cells were trypsinised, followed by seeding 500 cells per well of a 96 well μclear plate (Griener Bio-One, 655096) in a final volume of 100 μl of DMEM medium containing standard antibiotics and 10% FBS. Cells were mixed by pipetting to ensure even distribution and then incubated for 24 h at 37 °C with 5% $CO_2$. Then the plates were transferred to an Incucyte SX5 Live Cell Analysis System (Sartorius) and phase images were captured every 12 h for a total of 120 h. Subsequently, proliferation was analysed using the Incucyte analysis software, normalising to confluence (%) at the assay start point. Data was exported to Microsoft excel and values were plotted in GraphPad Prism 10.

To analyse cellular proliferation of DLD1 $BRCA2^{-/-}$ or SUM149PT $BRCA1^{-/-}$ cells expressing a minimal SLX4 T1260 containing fragment, 1500 cells were seeded per well in a PhenoPlate 96-well optically clear tissue culture plate (Revvity, 6055302). Cells were evenly distributed within the well by pipetting and then incubated at 37 °C with 5% $CO_2$ for 24 h. The following day the medium was aspirated and replaced with 200 μl of RPMI 1640 medium containing standard antibiotics, 10% FBS and 2 mM L-glutamine (DLD1 BRCA2-/- SLX4 T1260 fragment cells or empty vector control expressing) or S Ham's F-12 medium supplemented with 5% FBS, 10 μg/mL insulin, 0.5 μg/mL hydrocortisone and penicillin/streptomycin antibiotics for SUM149PT $BRCA1^{-/-}$ (SLX4 T1260 fragment or empty vector control expressing) containing 1 μg/ml doxycycline. Plates were then transferred to an Incucyte S3 Live Cell Analysis System (Sartorius) imaging system and phase images were captured every 12 h for a total of 144 h. Subsequently, proliferation was analysed using the Incucyte analysis software, normalising to confluence (%) at the assay start point divided by the by the number of objects detected. Data was exported to Microsoft excel and values were plotted in GraphPad Prism 10.

### Collection of asynchronous, S-phase and M-phase synchronised cells
To facilitate enrichment of HEK293TN cells in S phase or M phase, $2 \times 10^6$ cells were seeded in 15 cm tissue culture dishes and incubated at

37 °C with 5% $CO_2$ for 24 h. For enrichment of cells in S-phase, cells were incubated with complete medium containing 2 mM thymidine (Sigma Aldrich, T9250) for 18 hours. The medium was replaced by three buffer changes and then cells were incubated for 9 h at 37 °C with 5% $CO_2$ to facilitate cell cycle progression. The medium was then replaced with 2 mM thymidine containing complete medium. The cells were then incubated for a further 15 h at 37 °C with 5% $CO_2$. Thymidine containing medium was then removed followed by two further medium changes to facilitate thymidine wash-out. Cells were incubated for a further 3 h at 37 °C with 5% $CO_2$ and harvested by trypsinisation followed by inactivation by addition of complete medium and then by centrifugation at 300 x g for 3 min. The supernatant was aspirated, and the cells were resuspended in 1 ml 1x PBS. The cells were again centrifuged at 300 x g for 3 min, followed by aspiration of the supernatant. Then the cell pellet was snap frozen on dry ice. For enrichment of mitotic cells, after 24 h incubation at 37 °C with 5% $CO_2$, medium was replaced with complete medium containing 100 ng/ml nocodazole and incubated at 37 °C with 5% $CO_2$ for 18 h. At which point the medium was collected, followed by 5 ml 1x PBS wash, collection and then trypsinisation. This was followed by inactivation by addition of complete medium. Cells were pelleted by centrifugation at 300 x g for 3 minutes, the supernatant was aspirated, and cells were resuspended in 1 ml 1x PBS. The 300 x g for 3 min centrifugation step was repeated, followed by aspiration of the supernatant. The cell pellet was then snap frozen on dry ice. Asynchronous cells were collected in parallel by trypsinisation followed inactivation by addition of serum containing medium, then centrifugation at 300 x g for 3 min. Then the supernatant was aspirated, and cells were also washed once in 1 ml 1x PBS, centrifuged at 300 x g for 3 min. The supernatant was aspirated and the cell pellet snap frozen on dry ice. In all cases before the final centrifugation a 500 μl aliquot of cell suspension was fixed in ice-cold 70% ETOH followed by propidium iodide staining and flow cytometry analysis to confirm cell cycle stage by DNA content analysis (see below).

## Flow Cytometry

HEK293TN cells were pelleted by centrifugation at 300 x g for 3 min and 1x PBS was removed. The cell pellet was then loosened by flicking the tube and 5 ml 70% ice cold ethanol was added slowly while continually vortexing the sample at ~1000 rpm. The sample was then placed on ice for 30 minutes and stored until the day of processing at −20 °C. On the day of analysis, the cells were pelleted by centrifugation at 300 x g for 3 mins. The 70% ethanol was aspirated and 1 ml of 1x PBS was added. The sample was centrifuged again at 300 x g for 3 mins and the PBS was then aspirated. The previous PBS wash step was carried out twice more. The sample was then resuspended in 2 ml of 1x PBS with 1 μg/ml RNAse A (Roche, RNASEA-RO- 10109142001) and 1 μg/ml of propidium iodide. The sample was incubated in the dark at 4 °C for 4 hours, then vortexed and transferred to a 5 ml flow cytometry tube through a cell strainer. The sample was then analysed using the BD LSR II Analyzer flow cytometry instrument. Cells were gated by analysis of forward scatter area and side scatter area, followed by gating of single cells by gating forward scatter height against forward scatter area. A histogram was then plotted of the PE: TxRed-H PI intensity per event. The data was then analysed and visualised using FlowJo (FlowJo LCC) analysis software, comparing 7915 events per condition.

## Co-immunoprecipitation

For TOPBP1 Co-immunoprecipitation experiments, 20 μl of Dynabead protein G slurry were washed 3x in IP2 buffer (200 mM NaCl, 0.2% Igepal CA-630, 1 mM MgCl2, 10% glycerol, 5 mM NaF, 2 mM EDTA, 50 mM Tris-HCl, pH 7.5). Beads were then blocked in IP2 + 1% BSA + 1x SIGMAFAST protease inhibitors (Sigma-Aldrich) for 1 h at room temperature. Blocking buffer was removed and replaced with 1 ml IP2 buffer containing either 3 μg of anti-TOPBP1 (A300-111A, Bethyl) or 3 μg

of anti-rabbit IgG control (AP132, Sigma-Aldrich). The beads and antibody mixture were rotated at 4 °C overnight. The next day, cell pellets were removed from −80 storage and thawed on ice in 2.4 ml IP buffer 1 (100 mM NaCl, 0.2% Igepal CA-630, 1 mM MgCl2, 10% glycerol, 5 mM NaF, 50 mM Tris-HCl, pH 7.5), supplemented with complete PhosSTOP phosphatase inhibitor (PHOSS-RO, Roche), 1x SIGMAFAST protease inhibitor and 25 U ml−1 Benzonase (Novagen). Cell pellets were resuspended by pipetting and then rotated at 4 °C on a carousel for 90 min, to assist lysis. Benzonase was then inhibited by adjusting the NaCl concentration to 200 mM and by the addition of 2 mM EDTA, then rotated for a further 30 min at 4 °C on a carousel. Cell suspensions were then centrifuged at 16,000 x g for 25 min at 4 °C. Soluble supernatant (lysate) from the above centrifugation was then transferred to individual 1.5 ml microcentrifuge tubes. Then an additional 20 μl of Dynabead protein G bead slurry was transferred to a 1.5 ml microcentrifuge tube and washed 3x in IP2 buffer, assisted by a 1.5 ml magnetic separation rack. The cell suspension was then transferred to the tubes containing the washed Dynabead protein G beads and incubated at 4 °C for 1 hour to facilitate preclearing of the lysate. The cell supernatant was then transferred to a new 1.5 ml microcentrifuge tube, and the BCA assay was then carried out to determine protein concentration. The antibody mix was then removed from Dynabead protein G beads that were incubated overnight with IgG or anti-TOPBP1 antibodies. The antibody bound beads were then washed 3x in 500 μl IP2. Then the antibody bound Dynabead protein G beads were transferred to 2 ml tubes and cell lysate was combined with IP2 buffer to bring each sample to 1 mg/ ml final concentration. The tubes were then rotated at 4 °C on a carousel for 120 min. The supernatant was then removed and 500 μl IP2 buffer was added to wash the beads. This step was repeated once. IP2 buffer was then removed, and beads were washed with 500 μl 1x PBS. The 1x PBS was removed, and the PBS was step was repeated once more. The 1x PBS was removed and discarded and then beads bound by antibody-protein complexes were stored at −20 °C prior to IP-MS analysis.

## Mass-spectrometry proteomics

Beads were re-suspended in 100 μL of 100 mM TEAB, reduced with 10 μL of 50 mM TCEP, and alkylated with 5 μL of 200 mM freshly prepared iodoacetamide. The mixture was incubated at room temperature for 30 minutes in the dark. Proteins were digested overnight with trypsin (500 ng/μL in 0.1% formic acid) at 37 °C with shaking. The digested peptides were collected, dried, and cleaned using the Pierce™ High pH Reversed-Phase Peptide Fractionation Kit (Thermo Scientific). For TMT, peptides were labelled with TMTpro reagents according to the manufacturer's instructions (ThermoFisher scientific). TMT labelled samples were combined, vacuum-dried, fractionated using the same kit, and dried again before MS analysis. Samples were re-suspended in 0.1% TFA prior to analysis.

For label-free samples, LC-MS analysis was performed using a Dionex UltiMate 3000 UHPLC system coupled to an LTQ Orbitrap Lumos mass spectrometer (Thermo Scientific). Chromatographic separation was achieved on an EASY-Spray C18 column (75 μm × 50 cm, 2 μm) at 50 °C. The mobile phase consisted of 0.1% formic acid (A) and 80% acetonitrile with 0.1% formic acid (B). The gradient elution was: 0–150 minutes up to 38% B, 150–160 min up to 95% B, 160–165 minutes isocratic at 95% B, 165–175 min re-equilibration to 5% B, and 175–185 min isocratic at 5% B.

Data-independent acquisition (DIA) was performed on a high-resolution Orbitrap mass spectrometer in positive ion mode. An MS1 survey scan was acquired between 8 and 109 minutes at 60,000 resolution with a scan range of m/z 400–900. Quadrupole isolation was used with a Standard AGC target and a custom maximum injection time of 300 ms. DIA was conducted using 42 variable-width isolation windows across m/z 400–900, with precursor ions fragmented by HCD at 30% collision energy. Fragment ions were detected in the

Orbitrap at 15,000 resolution over m/z 145–1450. DIA windows were calculated to ensure complete coverage and minimal overlap of the precursor mass range.

For DIA-NN searches, raw data files were deconvoluted and converted to mzML files using MS Converter with filter settings peakPicking (vendor msLevel=1-) and titleMakerData. The data was searched against the human proteome (UniProtKB). The analysis employed a library-free workflow. For precursor ion generation, the options FASTA digest for library-free search/library generation and Deep learning spectra, RTs, and IMs prediction were enabled. The digestion parameters allowed for a maximum of one missed cleavage by trypsin, with a maximum of two variable modifications per peptide. Carbamidomethylation of cysteine residues was specified as a fixed modification, while methionine oxidation. Peptide length was restricted to a range of seven to 30 amino acids. The precursor charge range was set from 2 to 4, with the m/z range for both precursors and fragments set from 300 to 1300. The mass accuracy for both parent and fragment ions was manually adjusted to 10 ppm. A false discovery rate (FDR) threshold of 1% was applied at both the protein and peptide levels, and the match between runs feature was enabled to improve peptide identification across different samples.

Tandem mass tag (TMT) labelling of peptides was conducted according to manufacturer's instructions using TMTpro regents (Thermo Fischer Scientific). TMT labelled samples were analysed using a Thermo Orbitrap Ascend mass spectrometer. MS1 scans were performed over a mass range of m/z 400–1600 at 120,000 resolution in the Orbitrap, with standard AGC settings and automatic injection times. Ions with charge states +2 to +6 were included. Dynamic exclusion was set to 45 seconds with a repeat count of 1, a ± 10 ppm mass tolerance, and isotopes were excluded from further analysis.

MS2 spectra were acquired in the ion trap using a Turbo scan rate, with 32% HCD collision energy and a maximum injection time of 35 ms. Real-time database searching against Homo sapiens (canonical and isoforms) was conducted using the Comet search engine, considering tryptic peptides with a maximum of 1 missed cleavage. Static modifications were set for carbamidomethylation of C ( + 57.0215 Da) and TMTpro labelling on K and N-termini ( + 304.207 Da). Variable modifications included deamidation of N/Q ( + 0.984 Da) and oxidation of M ( + 15.9949 Da), allowing up to 2 variable modifications per peptide. A close-out feature was enabled, limiting to 4 peptides per protein.

SPS10-MS3 scans were performed on selected precursors using the Orbitrap at 45,000 resolution with 55% HCD collision energy, a 200% normalised AGC target, and a 200 ms maximum injection time. Data were collected in centroid mode with single microscan acquisition. For protein identification and quantification, Proteome Discoverer 3.0 (Thermo Scientific, Proteome Discoverer, RRID:SCR_014477) with SequestHT and Comet search engines was used. Spectra were searched against the UniProt (RRID:SCR_002380)

Homo sapiens database with a precursor mass tolerance of 20 ppm and a fragment mass tolerance of 0.02 Da. Peptides were considered fully tryptic, allowing up to two missed cleavages. Static modifications included TMT at N-termini/K and carbamidomethylation at C residues. Dynamic modifications included methionine oxidation and deamidation of N/Q. Peptide confidence was assessed using Percolator with a 1% FDR using target-decoy validation. Quantification employed the TMT quantifier node with a 15 ppm integration window using the most confident centroid peak at the MS2 level. Only unique peptides with a signal-to-noise ratio >3 were considered for quantification. Data were normalised to total protein loading, and relative abundances were calculated by dividing normalised values by the average abundance across all TMT channels per biological replicate.

Statistical analysis and data visualisation were performed using Python (RRID:SCR_008394). Gene ontology enrichment analysis was performed using ShinyGO (PMID: 31882993).

Samples generated from three independent experiments were analysed in label-free and TMT labelled mass-spectrometry experiments. Co-IP experiments were performed with IgG-coated beads in parallel with anti-TOPBP1-coated beads as negative control. Sample preparation was optimised via preliminary co-immunoprecipitation assays, followed by silver stain of gel replicate samples to confirm the pull-down of complex samples. Data were validated by Co-IP western blot experiments described in this study.

## GFP-Trap® Agarose Co-immunoprecipitation

HEK293TN cells were seeded to be ~25–30% confluence the next day in a 15 cm tissue culture dish. The following day cells were transfected using Lipofectamine 2000 transfection reagent (Thermo Fisher Scientific) with 24 µg of plasmid DNA encoding eGFP-SLX4 or eGFP-TOPBP1 expression cassettes following manufacturer's instructions.

Forty eight hours post transfection cells were harvested after treatments as described, snap frozen on dry ice and stored at −80 °C. Cell pellets were removed from −80 storage and thawed on ice in 2.4 ml IP buffer 1 (100 mM NaCl, 0.2% Igepal CA-630, 1 mM MgCl2, 10% glycerol, 5 mM NaF, 50 mM Tris-HCl, pH 7.5), supplemented with EDTA Free SIGMAFAST protease inhibitor (Sigma Aldrich) and 25 U ml−1 Benzonase (Novagen). Cells were resuspended and rotated at 4 °C for 90 min. Benzonase was then inhibited by adjusted NaCl concentration to 200 mM and with the addition of 2 mM EDTA, followed by rotation for a further 30 min at 4 °C on a carousel. The cell suspension was then centrifuged at 16,000 x g for 25 min at 4 °C. In parallel, 20 µl agarose binding control bead (bab, Chromotek) slurry was washed with 500 µl IP buffer 2 (200 mM NaCl, 0.2% Igepal CA-630, 1 mM MgCl2, 10% glycerol, 5 mM NaF, 2 mM EDTA, 50 mM Tris-HCl, pH 7.5). The binding control beads were then centrifuged at 2500 x g for 2 min. The supernatant was then removed and the IP2 wash was repeated twice more. Cell lysates (soluble supernatant) were then collected added to the agarose binding control beads in 15 ml centrifuge tubes and subsequently rotated at 4 °C on carousel for 60 min. The agarose binding control beads were then pelleted by centrifugation at 2000 x g for 2 min, the supernatant was then transferred to a new tube. BCA assay was then carried out to determine the protein concentration of the pre-cleared cell lysate. Next, 20 µl GFP_TRAP_A bead (gta, Chromotek) slurry was washed with 500 µl IP buffer 2 (200 mM NaCl, 0.2% Igepal CA-630, 1 mM MgCl2, 10% glycerol, 5 mM NaF, 2 mM EDTA, 50 mM Tris-HCl, pH 7.5). The beads were then centrifuged at 2500 x g for 2 min. The supernatant was removed and the IP2 wash step was repeated twice more. Cell lysates were diluted to 1 mg/ml final concentration in IP buffer 2 and added to the washed GFP_TRAP_A beads in 15 ml conical tubes, followed by rotation at 4 °C on a carousel for 120 min. The 15 ml conical tubes were then centrifuged at 2000 x g for 2 min at 4 °C and the supernatant was removed. The beads were then suspended in 500 µl IP2 buffer and transferred to a new 1.5 ml microcentrifuge tube. The sample was then centrifuged at 2500 x g for 2 min at 4 °C. The supernatant was removed and the IP2 wash was repeated twice more. Bound protein complexes were eluted from the GFP_TRAP_A beads by addition of 50 µl 2x SDS buffer (120 mM Tris/Cl pH 6.8, 20% glycerol, 4% SDS, 0.04% bromophenol blue, 10% β-mercaptoethanol) followed by incubation at 95 °C for 10 min. Tubes were then placed on ice for 10 min followed by centrifugation for 2500 x g for 2 min. The supernatant, now containing eluted material, was transferred to a new 1.5 ml microcentrifuge tube ready for downstream western blot analysis.

## BRCT 1 recognition motif bioinformatics search

Consensus motifs for BRCT1 and 2 of TOPBP1[26] were scanned over the SLX4 sequence using the ExPASy ScanProsite server[84]. Potential sites were checked against Phosphosite plus[53] to see if they were previously documented phosphorylation sites. The sequence conservation of the sites was also checked using the Proviz Server[85].

## Alphafold3 modelling

Amino acid sequences for TOPBP1(1-300) and SLX4(1240-1280) with pT1260 modification were submitted to the Alphafold3 web server[86]. Images of the complex, including superposition with known TOPBP1-RAD9 structure, were produced using PyMOL (version2.2.2; PyMOL, RRID:SCR_000305). The Alphafold3 structure was deposited to ModelArchive and available at https://modelarchive.org/doi/10.5452/ma-vgsnn.

## TOPBP1(BRCT0-2 1-290) purification

*E.coli* BL21(DE3) cells were transformed with pGEX-6P-1 TOPBP1(BRCT0-2 1-290) WT and K155E, K250E and K155E/K250E mutant variants for protein expression. Cell pellets were resuspended in lysis buffer containing 25 mM HEPES pH 7.5, 200 mM NaCl and 0.5 mM TCEP supplemented with 50U Turbo DNase, disrupted by sonication, and the resulting lysate clarified by centrifugation at 36,000 x g for 60 min at 4 °C. The supernatant was applied to a 5 ml HiTrap GST column, then washed with buffer containing 50 mM HEPES pH 7.5, 1000 mM NaCl, 0.25 mM TCEP, before the retained protein was eluted by application of the lysis buffer supplemented with 20 mM glutathione. This was concentrated using a Vivaspin with a 30 kDa MWCO and applied to a Superdex 200 16/60 size exclusion column equilibrated in 25 mM HEPES pH 7.5, 150 mM NaCl, 1 mM EDTA, 0.5 mM TCEP, 0.002% (v/v) Tween-20.

## CDK1-CyclinB-CKS1 purification

A single construct containing for expression of CDK1, Cyclin B and CKS1 was produced using the biGBac system[81]. SF9 cells were transfected with this bacmid to produce baculovirus for protein expression, and following two rounds of viral amplification, cells were infected at a cell density of $1.5 \times 10^6$ using 5% viral stock for expression and harvested after 48 hours. Cell pellets were resuspended in lysis buffer containing 25 mM HEPES pH 7.5, 200 mM NaCl and 0.5 mM TCEP supplemented with 50U Turbo DNase and cOmplete EDTA free protease tablets, disrupted by homogenisation and sonication, and the resulting lysate clarified by centrifugation at 36,000 x g for 60 min at 4 °C. The supernatant was applied to a 5 ml HiTrap TALON column, then washed with lysis buffer supplemented with 5 mM imidazole, before retained protein was eluted by application of the lysis buffer supplemented with 250 mM imidazole. This was applied to a 1 ml HiTrap StrepXT column, washed with lysis buffer, before elution of retained material with lysis buffer supplemented with 50 mM Biotin. The eluted material was concentrated using a Vivaspin with a 50 kDa MWCO and applied to a Superdex200increase 10/300 size exclusion column equilibrated in 10 mM HEPES pH 7.5, 150 mM NaCl, 5% (v/v) glycerol, 0.5 mM TCEP, 0.002% (v/v) Tween-20.

## Peptide pull-down experiments

Combinations of biotinylated SLX4 peptide corresponding to amino acids 1247–1267 (with or without Thr-1260 phosphorylated), active CDK1-CyclinB-CKS1, and ATP MgCl2, were mixed and incubated at 30 °C for 4 hours in reaction buffer 10 mM HEPES pH 7.5, 150 mM NaCl, 0.5 mM TCEP, 5% (v/v) Glycerol, 0.002 % (v/v) Tween-20. These reactions were then mixed with StreptactinXT magnetic beads equilibrated in reaction buffer and incubated for 15 min at 25 °C to allow biotinylated peptides to bind. Beads were washed with reaction buffer and subsequently GST-TOPBP1(BRCT0-2 1-290) protein was added and incubated for a further 15 minutes at 25 °C prior to again washing with reaction buffer to remove unbound material. Samples were eluted but the addition of reaction buffer supplemented with 25 mM biotin and analysed by PAGE and Coomassie staining.

Biotin-SLX4_T1260 'Biotin'-GYGSEASTTDTSWLVPATPLASRSR
Biotin-SLX4_pT1260 'Biotin'-GYGSEASTTDTSWLVPA(pT)PLASRSR

## In vitro fluorescence polarisation

Fluorescein-labelled peptides corresponding to amino acids 1253-1267 or 1470-1483 of SLX4 incorporating pT1260 or pT1476 respectively, were incubated at a concentration of 100 nM at room temperature with increasing concentrations of TOPBP1 WT or mutant BRCT module variants[26] in 25 mM HEPES pH 7.5, 150 mM NaCl, 1 mM EDTA, 0.25 mM TCEP, 0.002% (v/v) Tween 20 in a black 96-well polypropylene plate. Additional experiments performed with dephosphorylated peptide were achieved though incubation with Lambda phosphatase supplemented with 1 mM MnCl2 to remove the phosphorylation on the peptide. Fluorescence polarisation was measured in a CLARIOstar multimode microplate reader. Binding curves were calculated assuming a single binding site with non-specific binding component, using Graphpad Prism10.2.3 (GraphPad Prism, RRID: SCR_002798). Binding affinities are presented as calculated values alongside the 95% CI values. The presented graphs represent data plotted with the non-specific binding component subtracted. Each curve is produced from the mean of three independent experiments, with displayed error bars representing SEM.

Flu-SLX4_pT1260 'Flu'-GYGTSWLVPA(pT)PLASRSR
Flu-SLX4_pT1476 'Flu'-GYGSPGLLDT(pT)PIRGSCT

## ALT-EJ and BIR reporter assays

To facilitate transient depletion of CIP2A, SLX4, POLD3 or Polθ U2OS EJ2[65] or U2OS BIR[59] cells were transfected with 30 pmol of oligonucleotides using Lipofectamine RNAimax transfection reagent (Invitrogen, 13778100), according to the manufacturer's forward transfection protocol.

crRNA (EJ2 – CTAATTACCCTGTTATCCCT, BIR - AAGATTACCCTG TTATCCCT) targeting I-SceI recognition sequence was annealed with tracrRNA- Atto 550 (IDT) according to manufacturer's protocol.

Cas9 (IDT, 7 pmol) and annealed crRNA–tracrRNA (8.4pmol) were incubated in 100 µl Opti-MEM for 10 min at room temperature to facilitate RNP formation, followed by addition of 4 µl of Lipofectamine RNAimax and further incubation for 10 min. 250,000 cells 72 h post siRNA transfection were then transfected with RNP complex and analysed by FACS 24 h post transfection. The number of GFP positive cells per 30,000 (EJ2) or 10,000 (pBIR) Atto positive cells was determined and repair efficiency was normalised to siCTRL samples.

## Statistics and reproducibility

Statistical analyses were carried out using GraphPad Prism 10 (GraphPad Software Inc.). Student's t-test, One-way ANOVA, Two-way ANOVA, or Mann–Whitney U test were used to determine statistical significance as indicated in the figure legends. Western blot analysis of co-immunoprecipitation, siRNA depletion or knock-out experiments were carried out at least twice, where similar results were observed. For CDK1-Cyclin B-CKS1-dependent phosphorylation and pull-down analysis by SDS-PAGE, the analysis was conducted twice with similar observations.

## Reporting summary

Further information on research design is available in the Nature Portfolio Reporting Summary linked to this article.

## Data availability

Reagents generated in this study are available upon request to the corresponding authors. The mass spectrometry proteomics data have been deposited to the ProteomeXchange Consortium via the PRIDE[87] partner repository with the dataset identifier PXD068371. The Alphafold 3 model of the TOPBP1 BRCT1-2 SLX4pT1260 peptide interaction is available at ModelArchive (modelarchive.org) with the accession code ma-vgsnn [https://modelarchive.org/doi/10.5452/ma-vgsnn].

Due to 3D imaging data set size, microscopy imaging data is available upon request to the corresponding authors. The remaining data generated in this study are available within the article and its supplementary data files. Source data are provided with this paper.

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

## Acknowledgements

We thank D. Cortez (Vanderbilt University) for HCT116-osTIR1-TOPBP1-mAID-Clover cells, C. Lord for DLD1 and DLD1 *BRCA2*[-/-] cells and SUM149PT cells, and S. Jackson for RPE1 *p53*[-/-] FRT/TR cells. Also, Thanos D. Halazonetis for providing U2OS-pBIR reporter cells and J.Stark for providing U2OS-EJ2 reporter cells. In addition, we thank S.M. Noordemeer for providing RPE1 *p53*[-/-] *PAC*[-/-] WT and *BRCA1*[-/-] cells. We also thank J. Rouse for provision of *SLX4* cDNA containing plasmids, W. Yang, C. Lord, T.Trakarnphornsombat and A. Radzisheuskaya for support through provision of specific reagents. We acknowledge the Chelsea Light Microscopy facility (Kai Betteridge, Ross Scrimgeour and Queenie Lai) at The Institute of Cancer Research for providing expert support on microscopy and image analysis. Their support was instrumental in

achieving the results presented in this paper (https://doi.org/10.5281/zenodo.14851951). Additionally, the ICR Flow Cytometry Facility for assistance with cell sorting, cell cycle, BIR and MMEJ reporter analysis (H.Ale, Y. Semochkina and B. Satam). We also thank G. Benstead-Hume and S. Haider for bioinformatical analysis and scientific discussion. Additionally, we are grateful to M. Fernández-Casañas for experimental advice related to use of CDK1 inhibitor (RO3306) use. Schematic graphics were created using BioRender.com. We thank C. Zierhut, G. Coster and J. Pines for feedback on the manuscript ahead of submission. Finally, we thank M. Van Vugt and L. De Haan for scientific discussion. This work was funded by an MRC Research Grant to W.N and P.M (MR/X018547/1) and a BBSRC Discovery fellowship to P.M (BB/T009608/1). Work in W.N's lab was also supported by a CR-UK Programme grant [A24881] and by CRUK Radnet grant (CRUK RRCOER-Jun24/100006). S.K was supported by Bekker NAWA fellowship (BPN/BEK/2024/1/00159). J.M lab and J.V were funded by a Cancer Research UK Senior Cancer Research Fellowship to J.M. (RCCSCF-Nov22/100001). Funding supporting Z.K. and J.S.C. was provided by Wellcome Trust [223745/Z/21/Z] and from ICR core funding. Work carried out by M.D, A.O and L.P was supported by a Cancer Research UK Programme Grant (C302/A24386) and a Royal Society Research Grant (RG\R2\232017). Work in the J.D lab was supported by Cancer Research UK (C7905/A25715).

## Author contributions

P.M. and W.N. conceived the project. P.M. and W.N. designed and analysed experiments. M.D., J.N., Z.K., J.S.C., J.D., L.P., and A.O. contributed through experimental design, analysis, review and acquisition of funding. P.M. performed most of the experiments. J.N., M.D., Z.K., J.V., N.J., S.K., M.L., A.K., K.L., and A.O. contributed to specific experiments. Z.K. and J.C. assisted with proteomics experimental design and analysis. M.D., A.O., and L.P. assisted with in vitro experimental design, protein purification, fluorescence polarisation and CDK1 phosphorylation reconstitution experiments. J.V. and J.M. assisted with CRISPR knock-in design and execution. W.N. and P.M. wrote the manuscript, and all authors reviewed the manuscript ahead of submission.

## Competing interests

P.M., M.D., and W.N. are inventors on a filed patent application (GB 2415863.6 and GB2418894.8) covering targeting of the SLX4-TOPBP1 interaction for cancer therapy. The other authors declare no competing interests.
