## [Transparent Peer Review file · Nature Communications]

The CIP2A-TOPBP1 axis facilitates mitotic DNA repair via MiDAS and MMEJ

Corresponding Author: Dr Peter Martin

Version 0:

Reviewer comments:

Reviewer #1

(Remarks to the Author)

In this manuscript, the authors performed cell cycle-specific proteomics to identify TOPBP1 interacting proteins and found that the interaction of TOPBP1 and the components of the SMX complex (SLX1-SLX4/MUS81-EME1/XPF-ERCC1) is significantly enriched in mitotic cells. They further showed that TOPBP1 along with CIP2A facilitates the recruitment of the SMX complex to form foci in mitotic cells upon aphidicolin treatment or in BRCA2-deficient cells. They identified the BRCT1/2 domains of TOPBP1 mediating the interactions with SLX4 upon SLX4 phosphorylation at T1260 by CDK1. Furthermore, they showed that the SLX4-T1260A mutant is defective in MiDAS, suggesting that the recruitment of SMX by TOPBP1 through SLX4-T1260p binding is important for promoting MiDAS.

Technically, this report is strong with well-controlled experiments. Mapping BRCT domains for the interaction of TOPBP1 with SLX4 is new. Combining inactivation of Pol and SLX4 to retreat BRCA-deficient cancer cells has therapeutic implications. However, several important points described in this manuscript have been published (listed below). Additionally, given that the essential role of SMX is to generate DSBs to initiate MiDAS, the observation of a defect in BIR using the BIR reporter assay after CIP2A and SLX4 depletion—following I-SceI cleavage (where DSBs are already generated)—suggests that the role of CIP2A and SLX4 in BIR is more than just generating DSBs, but the related mechanism is not discussed. Overall, this study has brought some new aspects and provides incremental insights into DSB repair in mitosis with potential treatment applications.

- The role of TOPBP1 in MiDAS (or unscheduled DNA synthesis in mitosis): *Journal of Cell Biology* 210, 565-582 (2015)
- TOPBP1 interacts with SLX4 through CDK1 phosphorylation site T1260: *Genes & Development* 28, 1604-1619 (2014).
- The role of CIP2A in recruiting TOPBP1 and Pol in mitotic cells: *Nature cancer* 2, 1357-1371 (2021); *Nature Communications* 13, 4143 (2022); *Nature* 621, 415-422 (2023).
- CIP2A-TOPBP1 as a synthetic lethal target for BRCA-deficient cancer: *Nature cancer* 2, 1357-1371 (2021)

Specific comments:

1. Authors showed that TOPBP1 is colocalized with MUS81 and SLX4 in mitotic cells. For comparison, the foci formation of MUS81, SLX4 and TOPBP1 needs to be analyzed in interphase cells after aphidicolin treatment and with BRCA2 deficiency.
2. Some experiments need to be repeated in BRCA1-deficient cells. For instance, Figure 2A, 2C, 2D, 6D, 6F, 7C
3. TOPBP1 BRCT 0-1-2 domains also mediate the RAD9 pS387 interaction. It is not clear under what conditions BRCT 0-1-2 bind to RAD9 versus SLX4-T1260. Does the binding of RAD9 with TOPBP1 only occur in interphase cells?
4. Similarly, MDC1 also binds to TOPBP1 through BRCT 1-2, and both MDC1 and TOPBP1 form foci in mitotic cells. It would be informative to check whether MDC1 forms foci after aphidicolin treatment and in BRCA1 deficient cells, and whether TOPBP1 recruitment depends on MDC1 and its interaction with TOPBP1. Discussion would be needed for the regulation of MDC1/TOPBP1 and TOPBP1/SLX4 interactions, especially if TOPBP1 recruitment would require MDC1 interaction.
5. It is interesting that the major binding of MUS81 to TOPBP1 is independent of SLX4 interaction with TOPBP via T1260p. It is important to map the MUS81 binding sites on TOPBP1.
6. Since a substantial interaction between MUS81 and TOPBP1 is independent of SLX4-T1260, it needs to explain why MUS81 co-localization with CIP2A is significantly disrupted in the SLX4-T1260A knock-in cells (Fig. 5B). MUS81 should have an SLX4-T1260-independent recruitment route to the TOPBP1/CIP2A foci through its binding to TOPBP1.
7. After obtaining MUS81-binding sites on TOPBP1, the effect of MUS81 binding and SLX4 binding to TOPBP1 on MiDAS and micronuclei formation should be determined (Fig. 5C, 5D). MUS81 binding to TOPBP1, independent of SLX4-T1260p,

should also generate DSBs in mitosis at under-replicated DNA regions to initiate MiDAS. In Fig. 5D, SLX4-KO cells show more micronuclei compared to SLX4-T1260A-KI cells. Does MUS81 play a role in this?

8. In the BIR assay (Fig.6E), the readout is after I-SceI cleavage, and BIR frequency is reduced in CIP2A- and SLX4-depleted cells. The defects in MiDAS after CIP2A and SLX4 depletion are expected to attribute to impaired cleavage at under-replicated DNA regions, preventing the generation of DSBs required to promote MiDAS. Since DSBs are already generated in the BIR assay shown in Fig.6E, the mechanisms underlying the observed BIR deficiency upon CIP2A or SLX4 loss remain unclear.

In addition, CIP2A is located in the cytoplasm of interphase cells, and the BIR assay is performed in asynchronous cells. It is unclear whether the observed BIR defect originates from the mitotic stage, which is pretty short in asynchronous cells making it unlikely that the defect is caused by mitosis.

9. The cellular proliferation (as shown in 7B) of the T1260A-KI mutant expressed in BRCA1- and BRCA2-deficient cells (or with BRCA1/2 knockdown) needs to be assayed in combination with Pol θ inhibitor.

Minor points:

1. In the figures showing foci formation, the y axis indicates the number of colocalizing foci per prometaphase cell. It is not clear what percentage of the cell population contains foci. e.g. Figure 2A, 2B, 2C, 2D, 5A, 5B, 6C, 6D, 6F.
2. In mitotic cells after aphidicolin treatment or with BRCA2 deficiency, do SLX4 and MUS81, CIP2A colocalize with H2AX?

Reviewer #2

(Remarks to the Author)

This study uncovers a crucial mechanism for repairing mitotic DNA double-strand breaks, focusing on the CIP2A-TOPBP1 complex. This complex plays a key role in regulating DSB repair by facilitating two main pathways: mitotic DNA synthesis (MiDAS) and microhomology-mediated end-joining (MMEJ). During mitosis, CDK1-dependent phosphorylation of SLX4 at Thr1260 promotes interaction between SLX4 and the CIP2A-TOPBP1 complex, which facilitates the recruitment of SLX4, MUS81, and ERCC1 to mitotic chromatin. The CIP2A-TOPBP1 complex also recruits Pol θ to mitotic chromatin, aiding MMEJ pathway. Importantly, loss of CIP2A impairs both MiDAS and MMEJ pathways. The study highlights that SLX4, Pol θ , and CIP2A are essential for the proliferation of cells deficient in BRCA1/2. Overall, this research provides a valuable framework for understanding mitotic DSB repair and suggests that the CIP2A-TOPBP1 axis could be a promising therapeutic target for treating BRCA1/2-deficient cancers. The study is well-controlled, logical, results are exciting and represents an important advancement in our understanding of mitotic repair. However, one notable limitation of this study is that the authors primarily assessed the proteins' roles in the context of mitosis, despite many of these proteins having crucial functions in S and G2 phases as well. To strengthen the conclusions, it would be important to investigate whether the observed interactions and mechanisms are indeed exclusive to mitosis or if they also occur in other cell cycle phases. This additional context would be valuable before definitively labeling these processes as solely involved in mitotic DNA repair.

The authors have demonstrated that TOPBP1 co-localizes with SMX complex components and is required for their recruitment to chromatin in mitosis. This finding is significant; however, it is well-established that these proteins also play crucial roles and form foci during interphase. To strengthen the conclusion that this interaction is specific to mitosis, it would be valuable to investigate whether SLX4/MUS81 recruitment during S phase is also TOPBP1-dependent or not. This additional analysis would help distinguish between a mitosis-specific function and a more general role of TOPBP1 in regulating SMX complex interactions throughout the cell cycle.

The authors report an increase in mitotic colocalization of TOPBP1 with MUS81/ERCC1 upon BRCA2 knockout or aphidicolin treatment. To further strengthen this observation, it would be valuable to provide a quantitative comparison of TOPBP1 foci colocalization with MUS81/ERCC1 between mitosis and S/G2 phases. Specifically, what percentage of TOPBP1 foci colocalize with MUS81/ERCC1 in mitosis compared to S phase under these conditions?

To substantiate the authors' claim that SLX4 and MUS81 act downstream of TOPBP1, it would be important to assess whether SLX4 foci formation depends on TOPBP1 recruitment in both S/G2 and mitosis.

Considering that the authors have shown that the interaction between SLX4 and TOPBP1 can be prevented by adding a CDK inhibitor in nocodazole-arrested cells, it would be interesting to investigate whether TOPBP1 degradation in nocodazole-arrested cells, as opposed to asynchronous cells, can prevent SLX4/MUS81 foci formation.

Upon CDK1 treatment, do TOPBP1, SLX4, and MUS81 form foci? The authors suggest that TOPBP1 is upstream of SLX4 and MUS81; however, the absence of SLX4-TOPBP1 interaction does not prevent the SLX4-MUS81-ERCC1 interaction. Does this imply that upon CDK1 inhibition, cells will exhibit SLX4-MUS81 colocalizing foci independent of TOPBP1?

Upon CDK1 inhibition, TOPBP1 does not interact with SLX4, MUS81, or ERCC1; however, SLX4 interacts with MUS81 and ERCC1. Although in Figure 5, cells expressing the Thr1260A mutant do not appear to form any foci of MUS81, could the authors please discuss whether upon CDK1 inhibition SLX4-MUS81 interaction occurs in the soluble fraction? In that case, it may be necessary to perform the IP experiments using only the chromatin fraction to clarify these interactions.

Does CIP2A colocalize with FANCD2? Additionally, does CIP2A only form foci in mitosis, or is it also present during S/G2 phases?

Is the recruitment of Pol θ to the TOPBP1-CIP2A complex dependent on PLK1 or CDK1?

The model suggests that the SMX complex is recruited to chromatin through interactions with TOPBP1 and CIP2A, and this process is dependent on CDK1-mediated phosphorylation of SLX4. However, SLX4 is recruited to chromatin at stalled replication forks much earlier than mitotic entry, while MUS81 and EME1 are recruited to SLX4 upon mitotic entry. The authors need to discuss this conundrum.

Reviewer #3

(Remarks to the Author)

In this manuscript, the authors show that CIP2A-TOPBP1 interactions drives the recruitment of SLX4 and MUS81 to mitotic chromatin. Interestingly, CDK1-driven interaction between BRCT domains of TOPBP1 and phosphorylation of threonine1260 of SLX4 is critical for MiDAS. In parallel, the authors show that CIP2A promotes the recruitment of Pol θ to facilitate mitotic MMEJ. The authors show that loss of Pol θ mediated MMEJ and phosphor SLX4 mediated MiDAS could account for the synthetic lethality observed between CIP2A and BRCA1/2. This is a well performed study and provides mechanistic insights into the role of CIP2A in mediating multiple repair pathways.

Comments:

1. In Fig 2A and 2B, there is significantly higher amount of both MUS81 and ERCC1 which do not co-localize with TOPBP1. Why is that so?
2. In fig 5A in the representative pictures, the CIP2A foci seem to be more diffused un the SLX4 mutants. Is that true? If that the case, the authors should come up with an explanation as to why this is the case.
3. Throughout the manuscript, the authors measure micronuclei as a measure of genome instability. They should use additional assays to measure genome instability such as metaphase spreads or neutral comet assays to monitor repair efficiency.
4. How do the authors think both MiDAS and MMEj being controlled by CIP2A. is there site locus specific repair effects controlled CIP2A?

Version 1:

Reviewer comments:

Reviewer #1

(Remarks to the Author)

The authors have performed new experiments and adequately addressed the reviewers' comments. The manuscript is much improved.

Reviewer #2

(Remarks to the Author)

The authors have addressed the comments thoroughly and satisfactorily. The manuscript shows substantial improvement.

Reviewer #3

(Remarks to the Author)

The authors have addressed all my concerns sufficiently and I recommend this manuscript for publication.

Dear Reviewers

We thank you all for your constructive and insightful feedback, which has been instrumental in refining the core findings and expanding the scope of our study. Your suggestions have enabled us to strengthen the mechanistic conclusions, uncover new dimensions of mitotic genome maintenance, and better contextualise the roles of key regulatory components under replication stress. As a result of these contributions, we believe the manuscript now presents a significantly enhanced and more comprehensive narrative, and is well-positioned for consideration at Nature Communications.

Point-by-point responses:

Reviewer: 1

In this manuscript, the authors performed cell cycle-specific proteomics to identify TOPBP1 interacting proteins and found that the interaction of TOPBP1 and the components of the SMX complex (SLX1-SLX4/MUS81-EME1/XPF-ERCC1) is significantly enriched in mitotic cells. They further showed that TOPBP1 along with CIP2A facilitates the recruitment of the SMX complex to form foci in mitotic cells upon aphidicolin treatment or in BRCA2-deficient cells. They identified the BRCT1/2 domains of TOPBP1 mediating the interactions with SLX4 upon SLX4 phosphorylation at T1260 by CDK1. Furthermore, they showed that the SLX4-T1260A mutant is defective in MiDAS, suggesting that the recruitment of SMX by TOPBP1 through SLX4-T1260p binding is important for promoting MiDAS. Technically, this report is strong with well-controlled experiments. Mapping BRCT domains for the interaction of TOPBP1 with SLX4 is new. Combining inactivation of Pol θ and SLX4 to retreat BRCA-deficient cancer cells has therapeutic implications. However, several important points described in this manuscript have been published (listed below). Additionally, given that the essential role of SMX is to generate DSBs to initiate MiDAS, the observation of a defect in BIR using the BIR reporter assay after CIP2A and SLX4 depletion—following I-SceI cleavage (where DSBs are already generated)—suggests that the role of CIP2A and SLX4 in BIR is more than just generating DSBs, but the related mechanism is not discussed. Overall, this study has brought some new aspects and provides incremental insights into DSB repair in mitosis with potential treatment applications.

- *The role of TOPBP1 in MiDAS (or unscheduled DNA synthesis in mitosis): Journal of Cell Biology 210, 565-582 (2015)*
- *TOPBP1 interacts with SLX4 through CDK1 phosphorylation site T1260: Genes & 648 development 28, 1604-1619 (2014).*
- *The role of CIP2A in recruiting TOPBP1 and Pol θ in mitotic cells: Nature cancer 2, 1357-1371 (2021); Nature Communications 13, 4143 (2022); Nature 621, 415-422 (2023).*
- *CIP2A-TOPBP1 as a synthetic lethal target for BRCA-deficient cancer: Nature cancer 2, 1357-1371 (2021)*

Specific comments:

1. *Authors showed that TOPBP1 is colocalized with MUS81 and SLX4 in mitotic cells. For comparison, the foci formation of MUS81, SLX4 and TOPBP1 needs to be analysed in interphase cells after aphidicolin treatment and with BRCA2 deficiency.*

Our reply: We thank the reviewer for this suggestion. As requested, we have now analysed TOPBP1, SLX4 and MUS81 foci formation in interphase RPE1 p53^{-/-} FRT/TR cells treated with and without aphidicolin and also DLD1 WT compared to BRCA2^{-/-} cells. These new data are now presented in the New Supplementary Figures S4 and S5.

2. *Some experiments need to be repeated in BRCA1-deficient cells. For instance, Figure 2A, 2C, 2D, 6D, 6F, 7C*

Our reply: We are extremely grateful to the reviewer for raising this point, as it has broadened our study to an additionally relevant genetic background. As requested, we have now performed additional analysis with respect to TOPBP1 colocalisation with SLX4 and MUS81 foci in

prometaphase WT and BRCA1^{-/-} cells (Fig S3 B and C). We have also demonstrated the dependency of MUS81 recruitment in BRCA1^{-/-} on TOPBP1 in mitosis (Fig S6C) and the independency of TOPBP1 foci formation on SLX4 (Fig S6E). We also analysed the ability of BRCA1^{-/-} cells to proliferate under conditions of SLX4 depletion with DMSO or DNA polymerase theta inhibition treatment compared to CIP2a depletion (Fig R1C). We additionally demonstrate that in BRCA1^{-/-} cells MDC1 foci are elevated in comparison to their WT parental control (Fig S2B) and demonstrate that MDC1 depletion in this genetic background impairs, but does not abolish, mitotic recruitment of TOPBP1 (Fig. S2E). We again, thank the reviewer for prompting us to expand our study to establish the relevance of these mechanisms in BRCA1^{-/-}.

3. TOPBP1 BRCT 0-1-2 domains also mediate the RAD9 pS387 interaction. It is not clear under what conditions BRCT 0-1-2 bind to RAD9 versus SLX4-T1260. Does the binding of RAD9 with TOPBP1 only occur in interphase cells?

Our reply: We thank the reviewer for raising this important mechanistic question. Indeed, TOPBP1 BRCT 1-2 domains can interact with both RAD9 pS387 and phosphorylated SLX4 Thr1260. To directly address the reviewer's question, we analysed our TOPBP1 Co-immunoprecipitation mass-spectrometry data set. Our LC-MS mass spectrometry analysis of TOPBP1 interactors showed enrichment of the RAD9 TOPBP1 interaction in S-phase when compared to M phase, but we did not observe absence of the RAD9 interaction in mitosis (Fig R1A). Consistently, immunofluorescence analysis demonstrated RAD9 foci formation and colocalisation with TOPBP1 during prometaphase (Fig. R1B). We observe that RAD9 foci however infrequently colocalise with TOPBP1 in mitosis (Fig R1B). We observe that RAD9 TOPBP1 colocalising foci increase in response to aphidicolin treatment (Fig. R1B).

Our mass-spec data suggests a cell-cycle dependency in TOPBP1 binding with the RAD9–TOPBP1 interaction predominantly occurring in S-phase (Fig 1A and Fig R1A), whereas the SLX4 Thr1260–TOPBP1 interaction is specifically promoted in mitosis by CDK1 phosphorylation. Such a temporally regulated switch likely underpinned by unique kinases may coordinate distinct functional roles for TOPBP1 during interphase (e.g DNA replication checkpoint via RAD9) versus mitosis (e.g SMX complex-dependent DNA repair via SLX4). Interestingly, recent work from the Michael laboratory indicates TOPBP1 functions as an oligomer, likely enabling higher order complex formation and simultaneous interactions with multiple protein partners^{1,2}, consistent with the large focus formation we observe in mitosis. We do not rule out a role for RAD9 in mitosis we instead speculate that higher order complex formation of TOPBP1-CIP2A in addition to highly coordinated regulation of TOPBP1's interactions by distinct phosphorylation events will enable distinct sub-complex establishment as required for the appropriate substrate. We subsequently discuss this in the text and thank the reviewer for prompting us to do so, given this is an interesting aspect of the CIP2A-TOPBP1 complex. While characterising the role of the mitotic TOPBP1-RAD9 interaction would be interesting, we believe that it is beyond the scope of the present study and is a candidate for follow up research.

4. Similarly, MDC1 also binds to TOPBP1 through BRCT 1-2, and both MDC1 and TOPBP1 form foci in mitotic cells. It would be informative to check whether MDC1 forms foci after aphidicolin treatment and in BRCA1 deficient cells, and whether TOPBP1 recruitment depends on MDC1 and its interaction with TOPBP1.

Our reply: To directly address the reviewer's suggestion, we have now analysed MDC1 and TOPBP1 foci formation in mitotic cells, after aphidicolin treatment and in both BRCA1 or 2-deficient backgrounds (Fig S2B-D). Additionally, we assessed the impact of MDC1 depletion on mitotic TOPBP1 localisation (Fig S2E,F and I). Importantly, the molecular basis of the TOPBP1-MDC1 interaction has been extensively studied, establishing that CK2-dependent phosphorylation of MDC1 creates a conserved binding site recognized by the TOPBP1 BRCT1/2 domains. Disruption of this interaction selectively impairs TOPBP1 recruitment to mitotic—but not interphase—DNA damage sites, leading to mitotic genome instability in response to ionising radiation³. This is consistent with a model where distinct phosphorylation events enable a

nuanced response to specific stimuli, likely supported by TOPBP1's ability to oligomerise and form higher order complexes^{2,4,5}. As indicated above we have now added discussion of this to the text.

5. It is interesting that the major binding of MUS81 to TOPBP1 is independent of SLX4 interaction with TOPBP via T1260p. It is important to map the MUS81 binding sites on TOPBP1.

Our reply: We appreciate the reviewer highlighting this important point. In our initial experiments where we co-immunoprecipitated MUS81 with TOPBP1 BRCT1-2 from whole-cell lysates (Fig S8A), indeed MUS81 binding to TOPBP1 appeared independent of the phosphate binding pockets in BRCT 1 and 2 of TOPBP1. To explore this more deeply and to determine whether MUS81 binds instead independently in the N-terminal region of TOPBP1 we carried out further immunoprecipitation analysis. In an attempt to rule out an interaction with a similar region within BRCT0, we created a mutant TOPBP1 construct containing mutations within the N-terminal hydrophobic patch of BRCT 0 equivalent to the phosphate binding pocket, which had no impact on MUS81 binding (Fig R1D). Additional immunoprecipitation experiments using the isolated TOPBP1 BRCT0 fragment also revealed no binding to MUS81 or SLX4 (Fig R1E). In combination with our previous analysis indicating that MUS81 binds to the BRCT0-2 module of TOPBP1 (Fig 3B) this data suggested the BRCT1-2 portion of TOPBP1 is indeed important for MUS81 binding. We therefore, revisited the possibility that the phosphate binding pockets in BRCT1 and 2 may indeed be crucial for MUS81 binding and to explore this we carried out further analysis by immunoprecipitation, but instead specifically from the chromatin fraction of mitotic synchronised cells expressing WT or the BRCT1/2 phosphate binding pocket binding mutant (K155E and K250E, previously demonstrated to impair SLX4 binding (Fig S8A)) and analysed both SLX4 and MUS81 binding (Fig 4A). This analysis demonstrates that the TOPBP1 BRCT1/2 mutant (which abolishes interaction with phosphorylated SLX4) also partially reduced the chromatin residency of TOPBP1 and its interaction with MUS81 and ERCC1 (Fig. 4A and S8A). Moreover, our experiments involving acute CDK1 inhibition further support that CDK1-dependent phosphorylation of SLX4 Thr1260 is required for stable chromatin recruitment of MUS81 and SLX4 in mitosis (Fig. 4C- F, Fig 5B, Fig. S8C and D). Thus, our data now suggest that the chromatin-based recruitment of MUS81 to TOPBP1 in mitosis critically depends on BRCT1/2 domains and CDK1 dependent SLX4 phosphorylation at Thr1260.

We speculate that the persistent interaction between MUS81 and the TOPBP1 BRCT1/2 mutant observed in whole-cell lysates (Fig S8A) may reflect an ability of TOPBP1 and MUS81 to physically interact in the soluble compartment of the cells, which may or may not be relevant in context of stabilisation of genome integrity, or physiologically relevant *in situ*. Importantly, these interactions are not sustained under chromatin-associated conditions, highlighting the importance of the BRCT1/2 domains and SLX4 phosphorylation in the assembly and stability of the SMX complex at DNA damage sites in mitosis. Given our comprehensive analysis and findings, mapping additional candidate binding interfaces of MUS81 within TOPBP1 remains beyond the current scope but represents an interesting direction for future structural and biochemical investigations.

6. Since a substantial interaction between MUS81 and TOPBP1 is independent of SLX4-T1260, it needs to explain why MUS81 co-localization with CIP2A is significantly disrupted in the SLX4-T1260A knock-in cells (Fig. 5B). MUS81 should have an SLX4-T1260-independent recruitment route to the TOPBP1/CIP2A foci through its binding to TOPBP1.

Our reply: As clarified in our response to point 5, the stable chromatin recruitment and co-localisation of MUS81 with CIP2A/TOPBP1 complexes in mitosis critically depends on SLX4 phosphorylation at Thr1260 and intact TOPBP1 BRCT1/2 domains. Thus, although MUS81 appeared to interact with TOPBP1 independently of SLX4 in whole-cell lysates, our chromatin fractionation, chromatin immunoprecipitation, and CDK1 inhibition data (Fig. 4A-F, 5A S8A-E)

demonstrate that stable chromatin residency and proper mitotic localisation of MUS81 strictly require phosphorylated SLX4-Thr1260 interaction with TOPBP1.

7. After obtaining MUS81-binding sites on TOPBP1, the effect of MUS81 binding and SLX4 binding to TOPBP1 on MiDAS and micronuclei formation should be determined (Fig. 5C, 5D). MUS81 binding to TOPBP1, independent of SLX4-T1260p, should also generate DSBs in mitosis at under-replicated DNA regions to initiate MiDAS. In Fig. 5D, SLX4-KO cells show more micronuclei compared to SLX4-T1260A-KI cells. Does MUS81 play a role in this?

Our reply: We thank the reviewer for raising these interesting points. As addressed in our replies to points 5 and 6, our chromatin fractionation and CDK1 inhibition analyses strongly indicate that MUS81 chromatin recruitment in mitosis critically depends on CDK1 dependent phosphorylation activity, SLX4 phosphorylation at Thr1260 and intact TOPBP1 BRCT1/2 domains (Fig. 4A-F, 5A S8A-E). Thus, the apparently independent MUS81 binding observed in whole-cell lysate immunoprecipitation experiments likely reflects interactions occurring independent of chromatin context, if physiological, rather than representing an alternative stable chromatin recruitment mechanism.

We agree that SLX4 knockout cells exhibit a greater micronuclei phenotype compared to SLX4 Thr1260Ala knock-in cells (Fig. 5D). This observation aligns with the broader roles of SLX4 outside mitosis (e.g., interphase replication fork repair), beyond its specific phosphorylation-dependent interaction with TOPBP1 required for MiDAS in mitosis. Given these findings, dissecting additional, potential MUS81 binding sites on TOPBP1, distinct from SLX4, remains beyond the scope of the current manuscript but merits future exploration.

8. In the BIR assay (Fig.6E), the readout is after I-SceI cleavage, and BIR frequency is reduced in CIP2A- and SLX4-depleted cells. The defects in MiDAS after CIP2A and SLX4 depletion are expected to attribute to impaired cleavage at under-replicated DNA regions, preventing the generation of DSBs required to promote MiDAS. Since DSBs are already generated in the BIR assay shown in Fig.6E, the mechanisms underlying the observed BIR deficiency upon CIP2A or SLX4 loss remain unclear.

In addition, CIP2A is located in the cytoplasm of interphase cells, and the BIR assay is performed in asynchronous cells. It is unclear whether the observed BIR defect originates from the mitotic stage, which is pretty short in asynchronous cells making it unlikely that the defect is caused by mitosis.

Our reply: We thank the reviewer for raising these insightful points. Indeed, although the BIR assay utilises an initial Cas9/I-SceI-induced double-strand break (DSB), the observed requirement for CIP2A and SLX4 suggests roles beyond initial DSB formation, this in itself is interesting and provides details. While SLX4-MUS81 is known for its nuclease activity at under-replicated regions, the SMX complex is also well-established to facilitate the resolution of recombination intermediates, including Holliday junctions and migrating D-loops that are critical in the later stages of the BIR pathway. Importantly, MUS81 has been shown to cleave migrating D-loops, allowing multiple cycles of re-engagement during BIR, thereby promoting the completion of DNA synthesis downstream of the initial break⁶⁻¹¹. Thus, we propose that the requirement for CIP2A and SLX4 in the BIR assay likely reflects their roles in resolving these BIR intermediates rather than in generating the initiating DSB itself.

We agree with the reviewer that asynchronous cell populations could complicate interpretation regarding a strictly mitotic BIR repair event. That said, we emphasise that our primary readout for MiDAS is the established EdU incorporation assay, performed in mitotic cells. Subsequently, we additionally generated CIP2A^{-/-} cells and demonstrated that they display impaired MUS81-TOPBP1 recruitment in mitotic cells in response to aphidicolin induced replication stress (Fig 6A) consistent with the roles of CIP2A-MUS81 and TOPBP1 in MiDAS^{4, 9-12}. Furthermore, we analyse the ability of the newly generated CIP2A^{-/-} cells to incorporate EdU in mitosis in response to

aphidicolin induced replicative stress (Fig 6B). Again, we observe defects in the ability of these cells to incorporate EdU in mitosis as read-out of MiDAS activity. Furthermore, we detect colocalisation of CIP2A with γ H2AX and FANCD2 (established marker of persistent replication associated lesions, common-fragile sites and facilitator of MiDAS)¹³⁻¹⁶ (Fig 6E and F) under these conditions. Together this data providing strong evidence for CIP2A's mitotic role in driving BIR-like MiDAS (Fig.6A-F and S12A). In addition, recent independent evidence from Michael Lisby's laboratory supports our findings by demonstrating that TOPBP1 loss and SLX4 T1260A negatively impacts BIR-like MiDAS, further strengthening our conclusions regarding CIP2A's mitotic function as an upstream regulator^{4,9}. Furthermore, recent studies by the Ceccaldi¹⁷ and Sfier¹⁸ laboratories identified the critical role of Pol θ recruitment in mediating mitotic MMEJ. Our data showing impaired Pol θ recruitment following CIP2A depletion align well with these findings, underscoring the central regulatory function of CIP2A in coordinating mitotic DNA repair through MiDAS and MMEJ. However, we cannot fully rule out the possibilities that CIP2A may act both in *cis* and *trans* to support BIR across the cell cycle or that the BIR reporter system used largely assays mitotic repair events, we have subsequently highlighted this in our discussion.

9. *The cellular proliferation (as shown in 7B) of the T1260A-KI mutant expressed in BRCA1- and BRCA2-deficient cells (or with BRCA1/2 knockdown) needs to be assayed in combination with Pol θ inhibitor.*

Our reply: We thank the reviewer for raising this insightful question. As requested by the reviewer, we have now performed proliferation analyses of the SLX4 T1260A-knock in mutant in combination with Pol θ inhibitor treatment after BRCA1- and BRCA2-depletion. This data is included in the new Supplementary Figure S15B,C and D of our revised manuscript.

Minor points:

1. *In the figures showing foci formation, the y axis indicates the number of colocalizing foci per prometaphase cell. It is not clear what percentage of the cell population contains foci. e.g. Figure2A, 2B, 2C, 2D, 5A, 5B, 6C, 6D, 6F.*

We have provided this data for the reviewer in Fig R2A-I.

2. *In mitotic cells after aphidicolin treatment or with BRCA2 deficiency, do SLX4 and MUS81, CIP2A colocalize with γ H2AX?*

We thank the reviewer for this comment as it prompted us to provide further insight into the mitotic DNA damage response. Subsequently, in RPE1 p53^{-/-} cells treated with aphidicolin we observed elevated colocalisation of γ H2AX with either CIP2A, SLX4 or MUS81. In addition, we observe increased colocalisation between γ H2AX and either CIP2A or SLX4 in BRCA2 deficient DLD1 cells in comparison to WT. This data is presented in Fig.6E, S13F-I.

We are deeply grateful to Reviewer 1 for their thoughtful engagement with our work and their evident broader appreciation of mitotic DNA repair processes. Their insightful suggestions have been instrumental in guiding us to deepen our mechanistic understanding of these mitotic DNA repair processes. The requested analyses and expanded discussion have, in our view, substantially enhanced the manuscript, allowing us to elucidate novel aspects of these pathways with greater clarity. We sincerely thank the reviewer for their valuable contribution to the refinement of our study.

Reviewer: 2

This study uncovers a crucial mechanism for repairing mitotic DNA double-strand breaks, focusing on the CIP2A-TOPBP1 complex. This complex plays a key role in regulating DSB repair by facilitating two main pathways: mitotic DNA synthesis (MiDAS) and microhomology-mediated end-joining (MMEJ). During mitosis, CDK1-dependent phosphorylation of SLX4 at Thr1260

promotes interaction between SLX4 and the CIP2A-TOPBP1 complex, which facilitates the recruitment of SLX4, MUS81, and ERCC1 to mitotic chromatin. The CIP2A-TOPBP1 complex also recruits Polθ to mitotic chromatin, aiding MMEJ pathway. Importantly, loss of CIP2A impairs both MiDAS and MMEJ pathways. The study highlights that SLX4, Polθ, and CIP2A are essential for the proliferation of cells deficient in BRCA1/2. Overall, this research provides a valuable framework for understanding mitotic DSB repair and suggests that the CIP2A-TOPBP1 axis could be a promising therapeutic target for treating BRCA1/2-deficient cancers. The study is well-controlled, logical, results are exciting and represents an important advancement in our understanding of mitotic repair. However, one notable limitation of this study is that the authors primarily assessed the proteins' roles in the context of mitosis, despite many of these proteins having crucial functions in S and G2 phases as well. To strengthen the conclusions, it would be important to investigate whether the observed interactions and mechanisms are indeed exclusive to mitosis or if they also occur in other cell cycle phases. This additional context would be valuable before definitively labeling these processes as solely involved in mitotic DNA repair.

The authors have demonstrated that TOPBP1 co-localizes with SMX complex components and is required for their recruitment to chromatin in mitosis. This finding is significant; however, it is well-established that these proteins also play crucial roles and form foci during interphase. To strengthen the conclusion that this interaction is specific to mitosis, it would be valuable to investigate whether SLX4/MUS81 recruitment during S phase is also TOPBP1-dependent or not. This additional analysis would help distinguish between a mitosis-specific function and a more general role of TOPBP1 in regulating SMX complex interactions throughout the cell cycle.

Our reply: We thank the reviewer for their appreciation of our study and for raising these insightful points. Indeed, we now include additional data demonstrating elevated recruitment of SLX4, MUS81, and TOPBP1 to chromatin in interphase cells upon replicative stress induced by aphidicolin treatment or BRCA2 deficiency (Fig. S4 and S5). Interestingly, our quantitative analysis shows elevated colocalisation of SLX4/MUS81 with TOPBP1 during mitosis compared to interphase (Fig. S5A-D), underscoring mitotic enrichment and requirement of these interactions in mitosis.

Notably, our analysis of interphase cells also revealed increased SLX4 and MUS81 foci formation in Cyclin A2 positive cells upon TOPBP1 depletion in BRCA2-deficient and aphidicolin-treated RPE1 cells (Fig. S5E-H). Collectively, these findings indicate that while TOPBP1 plays a central role in SMX complex component recruitment in mitosis—dependent on CDK1-mediated phosphorylation of SLX4 at Thr1260—additional TOPBP1-independent mechanisms may operate to recruit SMX components during interphase under conditions of replication stress. We have included this detailed analysis and discussion in the revised manuscript to clarify these important points and are thankful to the reviewer for prompting us to expand our study in this manner.

The authors report an increase in mitotic colocalization of TOPBP1 with MUS81/ERCC1 upon BRCA2 knockout or aphidicolin treatment. To further strengthen this observation, it would be valuable to provide a quantitative comparison of TOPBP1 foci colocalization with MUS81/ERCC1 between mitosis and S/G2 phases. Specifically, what percentage of TOPBP1 foci colocalize with MUS81/ERCC1 in mitosis compared to S phase under these conditions?

Our reply: We appreciate the reviewer's suggestion to provide a quantitative comparison of TOPBP1–MUS81/ERCC1 colocalisation between S/G2 and mitosis under conditions of BRCA2 loss or aphidicolin-induced replication stress. We now include comparative analysis of the percentage of interphase colocalising foci compared to mitotic colocalising foci of MUS81/SLX4 with TOPBP1 in the revised manuscript (New Fig. S5A–D), demonstrating a marked mitotic enrichment.

In relation to these data, we are conscious of the reviewers request to analyse S/G2 colocalising foci specifically in comparison to mitosis rather than interphasic compared to mitosis. We became aware that including an additional S/G2 marker (Cyclin A2) was technically unfeasible with our established models, reagents and microscopes. To be specific the use of species-specific

antibodies (SLX4, MUS81 (mouse), TOPBP1 (rabbit) and Cyclin A2 (rabbit)) prevented us from conducting multiplexing of three antibodies in our RPE1 p53^{-/-} cells and DLD1 WT/BRCA2^{-/-} cells. Additionally, given the small but important spectral overlap between 488/555 and 555/647 wavelengths of Alexa fluor antibody excitation/emission when using our spinning disc microscope systems that were used for analysis throughout the manuscript, we wished to avoid crosstalk between channels and possible misreporting limiting use to 488 + 647 fluors.

Consequently, we decided to provide data that quantitatively demonstrates that TOPBP1–SMX component complex interactions (Fig. 1A-C) and colocalisations (Fig S5) are enhanced during mitosis when compared to interphase, especially in response to aphidicolin treatment (S5D and E) and in a BRCA2-deficient setting (Fig S5A and B) underscoring the mitotic engagement of TOPBP1 with SMX components.

Consistently, we find that TOPBP1 depletion does not impair SLX4 or MUS81 foci formation under replication stress in interphase, suggesting the existence of TOPBP1-independent recruitment pathways in interphase. However, the increase in colocalisation in mitosis highlights TOPBP1's role in orchestrating SMX complex-mediated activity in mitosis. These findings, in our view, reinforce our conclusion that TOPBP1 functions in mitosis as a scaffold for SMX complex assembly and activity under conditions of replicative stress.

To substantiate the authors' claim that SLX4 and MUS81 act downstream of TOPBP1, it would be important to assess whether SLX4 foci formation depends on TOPBP1 recruitment in both S/G2 and mitosis.

Our reply: We thank the reviewer for this valuable suggestion. In our revised manuscript, we present new data showing that SLX4 and MUS81 foci formation in mitosis critically depends on TOPBP1, as both immunofluorescence analyses and chromatin fractionation followed by IP demonstrate severe impairment in the absence of TOPBP1 or upon mutation of TOPBP1 BRCT1/2 domains, respectively (Fig. 2C–E, 4A, S3, S4, S5 and S8A). This provides strong evidence for mitotic dependency of SLX4 and MUS81 recruitment on TOPBP1.

In interphase (S/G2 phases), our analyses reveal increased rather than decreased SLX4 and MUS81 foci formation upon TOPBP1 depletion (Fig. S5). This indicates potential compensatory or alternative mechanisms for SLX4 and MUS81 recruitment to chromatin outside mitosis, distinct from their strict mitotic requirement for TOPBP1.

Together, these data support the conclusion that SLX4 and MUS81 act downstream of TOPBP1 in mitosis, and that the regulatory dependency of these factors changes depending on the cell-cycle stage. We have included these clarifications in the revised manuscript to address this important point.

Considering that the authors have shown that the interaction between SLX4 and TOPBP1 can be prevented by adding a CDK inhibitor in nocodazole-arrested cells, it would be interesting to investigate whether TOPBP1 degradation in nocodazole-arrested cells, as opposed to asynchronous cells, can prevent SLX4/MUS81 foci formation.

Our reply: We appreciate the reviewer's valuable suggestion. Indeed, we believe we have addressed this point experimentally by performing a two-hour acute degradation of TOPBP1 specifically in nocodazole-arrested cells (Fig. 2E and S6G). This analysis demonstrates a near-complete loss of MUS81 foci formation upon acute TOPBP1 depletion in G2/M, directly supporting our conclusion that mitotic recruitment of MUS81 (and by inference SLX4, as previously demonstrated) requires TOPBP1. Furthermore, we now demonstrate that acute CDK1 inhibition in mitosis disrupts TOPBP1 and MUS81 foci formation in mitosis (Fig S8D and E) Thus, we believe that these existing data, complementary to our analysis of the SLX4 T1260A mutant that cannot be phosphorylated by CDK1 and analysis of foci formation after acute CDK1 inhibition adequately address the reviewer's request and demonstrate mitotic stabilisation/recruitment of SLX4-MUS81.

Upon CDK1 treatment, do TOPBP1, SLX4, and MUS81 form foci? The authors suggest that TOPBP1 is upstream of SLX4 and MUS81; however, the absence of SLX4-TOPBP1 interaction does not prevent the SLX4-MUS81-ERCC1 interaction. Does this imply that upon CDK1 inhibition, cells will exhibit SLX4-MUS81 colocalizing foci independent of TOPBP1?

Our reply: We thank the reviewer for raising this important question. Our new data demonstrates that acute CDK1 inhibition significantly impairs chromatin residency and mitotic foci formation of TOPBP1, SLX4, and MUS81 (Fig S8C–E). These results in combination to our analysis of the SLX4 T1260A mutant (Fig.4 and 5), strongly suggest that CDK1-dependent phosphorylation of SLX4-Thr1260 and its subsequent interaction with TOPBP1 are critical for stable mitotic recruitment of the SLX4-MUS81-ERCC1.

While our co-immunoprecipitation data indicates that SLX4 can still associate with MUS81 and ERCC1 independently of phosphorylated SLX4-TOPBP1 interaction in solution (whole-cell lysates), the chromatin-based recruitment of SLX4 and stable foci formation of TOPBP1 and MUS81 strictly depend on intact CDK1 activity (Fig 8C-E) and the CDK1 regulated SLX4 (pT1260)-TOPBP1 (BRCT1/2) interaction in mitosis (Fig.4). Thus, upon acute CDK1 inhibition, we observe a reduction in TOPBP1 and MUS81 foci formation in mitotic cells (Fig S8D and E). We have clarified these points in our revised manuscript to explicitly address the reviewer's query.

Upon CDK1 inhibition, TOPBP1 does not interact with SLX4, MUS81, or ERCC1; however, SLX4 interacts with MUS81 and ERCC1. Although in Figure 5, cells expressing the Thr1260A mutant do not appear to form any foci of MUS81, could the authors please discuss whether upon CDK1 inhibition SLX4-MUS81 interaction occurs in the soluble fraction? In that case, it may be necessary to perform the IP experiments using only the chromatin fraction to clarify these interactions.

Our reply: We appreciate the reviewer raising this thoughtful point. Indeed, our co-immunoprecipitation analysis shows that upon CDK1 inhibition, TOPBP1 interaction with SLX4, MUS81, or ERCC1 is impaired (Fig. 4C and 4D), whereas SLX4 maintains interactions with MUS81 and ERCC1 in whole-cell lysates (Fig 4C). However, as discussed above, the interactions observed in soluble fractions (whole-cell lysates) may represent chromatin independent, indirect or non-physiological binding events, distinct from those stabilized on chromatin under physiological mitotic conditions (Fig 4A, 5A,B, S8 C,D, E and S11A).

Our mitotic cellular fractionation experiment followed by eGFP-SLX4 co-immunoprecipitation from the chromatin fraction after treatment with DMSO or CDK1i (Fig.S8C) demonstrates that stable recruitment of SLX4 and MUS81 to mitotic chromatin critically depends on CDK1-mediated phosphorylation. Consistent with this, the SLX4 Thr1260Ala mutant, which disrupts the TOPBP1 interaction, fails to support chromatin-based MUS81, ERCC1 or SLX4 foci formation (Fig. 5A, B and S11A). These findings strongly suggest that while SLX4–MUS81 interactions may persist transiently in solution, the assembly of functional SMX components at chromatin-bound damage sites strictly requires CDK1-dependent phosphorylation and interaction with TOPBP1.

Moreover, our observations (data not shown) suggests that extended pan-CDK1 substrate inhibition not only disrupts specific phosphorylation events but also significantly alters protein dynamics and can cause mitotic exit complicating further experimental analysis.

Does CIP2A colocalize with FANCD2? Additionally, does CIP2A only form foci in mitosis, or is it also present during S/G2 phases?

Our reply: We are extremely grateful for the reviewer raising these queries. We have now analysed the colocalisation between CIP2A and FANCD2 on mitotic chromatin. Our data demonstrate substantial colocalisation of CIP2A with FANCD2 in mitosis (Fig. 6F), supporting a model in which CIP2A participates in resolving FANCD2-marked, replication-associated DNA lesions during mitosis which is complementary to the additional analysis within the manuscript.

Regarding cell-cycle specificity, our analysis (Fig. S12) reveals that CIP2A is predominantly cytoplasmic in interphase cells, including those in S/G2 phases, with little evidence of focal nuclear accumulation. Strikingly, CIP2A shows specific focal chromatin localisation exclusively in mitotic cells, supporting a specialised chromatin associated mitotic function.

We wish to point out that in figure S12 we fix cells with 2% PFA and proceed to permeabilisation to retain the soluble compartment of the cells in our analysis, this is distinct from some of the additional analysis we conduct throughout the manuscript where we fix cells with PTEMF buffer (containing Triton-X which acts to extract the soluble material prior to completion of fixation). Consequently, images in Fig.S12 capture the cytosolic and likely, additional non-chromatin bound CIP2A to provide a complete picture of CIP2A localisation. We compare in this figure (Fig.S12) our newly generated CRISPR knock out CIP2A^{-/-} cells to provide control for CIP2A immunofluorescence staining, in turn further supporting the validation of the models. We have incorporated these data and observations into the revised manuscript to address the reviewer's query.

Is the recruitment of Polθ to the TOPBP1-CIP2A complex dependent on PLK1 or CDK1?

Our reply: We thank the reviewer for this intuitive question. We wish to highlight that recent work from the Ceccaldi laboratory has provided an extensive characterisation of Polθ recruitment to mitotic chromatin through its interaction with TOPBP1, determining their interaction and the subsequent recruitment to mitotic chromatin is driven by a PLK1-mediated phosphorylation of Polθ at S1482, S1486, S1488 and S1493 that enables its interaction with BRCT7/8 of TOPBP1 to facilitate chromatin loading and MMEJ¹⁷. We have now highlighted this within the manuscript discussion to improve clarity.

In addition, sequential phosphorylation, first by CDK1 priming for further PLK1 phosphorylation, is a well-characterised regulatory mechanism for mitotic substrate modification¹⁹⁻²¹. This mechanism suggests CDK1 acting upstream, creating phospho-epitopes that facilitate subsequent PLK1 docking and activity¹⁹⁻²¹. In our manuscript, we have demonstrated that CIP2A depletion impairs Polθ recruitment, consistent with a functional link between CIP2A, TOPBP1, and Polθ recruitment in mitosis. However, direct analysis of the relative contribution of CDK1 and PLK1 to Polθ recruitment is beyond the current scope given the previous extensive characterisation¹⁷.

The model suggests that the SMX complex is recruited to chromatin through interactions with TOPBP1 and CIP2A, and this process is dependent on CDK1-mediated phosphorylation of SLX4. However, SLX4 is recruited to chromatin at stalled replication forks much earlier than mitotic entry, while MUS81 and EME1 are recruited to SLX4 upon mitotic entry. The authors need to discuss this conundrum.

Our reply: We thank the reviewer for raising this important point. Indeed, as the reviewer correctly notes, previous work has established that SLX4 associates with stalled replication forks already during S-phase, while MUS81–EME1 recruitment and activation of the SMX complex predominantly occurs upon entry into mitosis. Our data are consistent with this established temporal regulation, showing that mitotic recruitment of MUS81 and stable formation of functional SMX complex components on chromatin requires CDK1-mediated phosphorylation of SLX4 at Thr1260 and subsequent interaction with TOPBP1 BRCT1/2 domains (Fig. 4 and 5). We propose that while initial SLX4 chromatin localisation at stalled forks in interphase is independent of Thr1260 phosphorylation and TOPBP1 (New Figures S5E-H and S9C), this phosphorylation event in mitosis specifically triggers the association of SLX4 with TOPBP1 and facilitates stable chromatin recruitment of SLX4, MUS81 and ERCC1 (Fig S4A-E, 5A, B, S8C and S11A). Thus, CDK1-dependent phosphorylation serves as a critical regulatory step through stabilisation on mitotic chromatin, temporally coupling SLX4-dependent repair activities to mitotic

entry, as required for resolving persistent replication intermediates specifically in mitosis in a distinct manner to the mechanisms employed for SLX4-MUS81 recruitment in interphase.

We thank Reviewer 2 for their insightful comments, which have been pivotal in refining the mechanistic interpretation of our findings. Their input led us to reinforce the conclusion that the TOPBP1–SLX4-driven pathway operates specifically during mitosis and to elucidate the regulatory role of CDK1 in modulating these components. Particularly impactful was the suggestion to investigate FANCD2–CIP2A colocalisation. This analysis has revealed a previously underappreciated dimension of CIP2A’s involvement in the cellular response to persistent replication stress-associated lesions, contextualised by FANCD2’s established role in this process. These findings not only strengthen the mechanistic framework of our study but also broaden its relevance to understanding mitotic genome maintenance under stress conditions.

Reviewer: 3

In this manuscript, the authors show that CIP2A-TOPBP1 interactions drives the recruitment of SLX4 and MUS81 to mitotic chromatin. Interestingly, CDK1-driven interaction between BRCT domains of TOPBP1 and phosphorylation of threonine1260 of SLX4 is critical for MiDAS. In parallel, the authors show that CIP2A promotes the recruitment of Polθ to facilitate mitotic MMEJ. The authors show that loss of Polθ mediated MMEJ and phosphor SLX4 mediated MiDAS could account for the synthetic lethality observed between CIP2A and BRCA1/2. This is a well performed study and provides mechanistic insights into the role of CIP2A in mediating multiple repair pathways.

Comments:

1. In Fig 2A and 2B, there is significantly higher amount of both MUS81 and ERCC1 which do not co-localize with TOPBP1. Why is that so?

Our reply: We thank the reviewer for highlighting this observation. We agree that in Figures 2A and 2B, some MUS81 and ERCC1 foci do not fully colocalize with TOPBP1. This may arise through a number of reasons, likely through nonspecific binding and antibody limitations, which are common challenges associated with available antibodies. To provide a clearer and more precise analysis of SMX complex recruitment, we generated cells expressing eGFP-SLX4 (Fig. 5A). Furthermore, to address the reviewer's concern, we have also generated eGFP-MUS81 expressing cells which demonstrated colocalisation of eGFP-MUS81 with TOPBP1, providing improved clarity and specificity (Fig.R3A).

While these additional experiments support our conclusions surrounding SMX component localisation relative to TOPBP1, we cannot fully rule out and therefore speculate that there is possibility that a pool of SMX components may act independently on mitotic chromatin not in proximity with TOPBP1.

2. In fig 5A in the representative pictures, the CIP2A foci seem to be more diffused un the SLX4 mutants. Is that true? If that the case, the authors should come up with an explanation as to why this is the case.

Our reply: We thank the reviewer for this observation. To objectively address the apparent difference in CIP2A foci diffusion in Figure 5A, we performed quantitative imaging analysis (measuring bounding width and height of CIP2A foci in 3D) across hundreds of CIP2A foci from images in Fig.5A comparing wild-type eGFP-SLX4 and eGFP-SLX4 T1260A expressing cells, utilising the Aivia analysis software (Leica). This analysis revealed that counter to the reviewer’s impression there is a statistically significant decrease in CIP2A foci bounding width and height (in

our view a measure of diffusion in this context) in eGFP-SLX4 T1260A cells when compared to WT. These data are presented in Fig.R3B. While this represents an interesting observation further investigation of this phenomenon is beyond the scope of our manuscript.

3. Throughout the manuscript, the authors measure micronuclei as a measure of genome instability. They should use additional assays to measure genome instability such as metaphase spreads or neutral comet assays to monitor repair efficiency.

Our reply: We appreciate the reviewer's valuable suggestion, prompting us to provide additional read-outs for genome instability. In addition to micronuclei assays, we have now included quantitative analyses of γ H2AX and 53BP1 foci formation as complementary measures of genome instability (Fig. S14E and F). γ H2AX and 53BP1 foci serve as well-established markers of persistent DNA damage and DNA double-strand breaks, respectively. As such these markers are well characterised to label sites of persistent DNA damage/breaks, thereby providing robust validation of our findings and further evidence of genome instability under TOPBP1-SLX4, or Pol θ disruption. These additional data reinforce our conclusions regarding the impact of disrupting the TOPBP1-SLX4-Pol θ axis on genome stability. We also wish to highlight that additional analysis by the Stucki and Durocher laboratories provide supporting and extensive insight into genome instability phenotypes associated with CIP2A loss^{22, 23}

4. *How do the authors think both MiDAS and MMEJ being controlled by CIP2A. is there site locus specific repair effects controlled CIP2A?*

Our reply: We thank the reviewer for raising this important mechanistic question. Our findings suggest that CIP2A-TOPBP1 functions as a central mitotic hub, coordinating both MiDAS and MMEJ by recruiting TOPBP1, SLX4, MUS81, and Pol θ specifically to chromatin in mitosis. Notably, our data showing substantial colocalisation of CIP2A with FANCD2 (Fig. 6F) strongly support the idea that CIP2A-mediated recruitment of these repair factors is targeted to specific genomic loci—such as under-replicated regions or fragile sites—marked by FANCD2^{13-16, 24}. One hypothesis would be that CIP2A, may play a critical role in stabilising mitotic signalling events driven by phosphorylation and in turn the localisation of factors that bind to TOPBP1 in a phosphorylation dependent manner in mitosis and those required for MiDAS and MMEJ. This may be perhaps in part through its canonical function in inhibition of protein phosphatase 2a (PP2A)²⁵, Given that CIP2A-TOPBP1 structures on mitotic chromatin are large and involve localisation of multiple factors (MDC1, TOPBP1, CIP2A, SLX4, MUS81, ERCC1, Pol θ) we speculate that it is likely higher order complex² formation and coordinated phospho-regulation may provide nuanced and loci/substrate specific responses. We appreciate the reviewer's interest in this aspect of our study and have expanded our discussion to include this in the text.

We are grateful to Reviewer 3 for their intrigue and their feedback, which has led us to generate additional data demonstrating that loss of SLX4 phosphorylation at T1260, particularly in the context of Pol θ inhibition, results in heightened genome instability. These findings underscore the functional importance of this phosphorylation event in maintaining mitotic genome integrity under stress. We also thank the reviewer for prompting us to broaden our discussion of CIP2A's role in response to replication stress and locus specific implications. Indeed our newly generated data demonstrating CIP2A localisation at γ H2AX or FANCD2-marked sites is indicative of a response at key regions within the genome. This has allowed us to more fully explore the potential implications of CIP2A's recruitment in the context of replication stress responses, highlighting its possible role in coordinating mitotic resolution of persistent DNA lesions.

1. Montales, K., Kim, A., Ruis, K. & Michael, W.M. Structure-function analysis of TOPBP1's role in ATR signaling using the DSB-mediated ATR activation in *Xenopus* egg extracts (DMAX) system. *Scientific Reports* **11**, 467 (2021).
2. Kim, A. *et al.* Biochemical analysis of TOPBP1 oligomerization. *DNA repair* **96**, 102973 (2020).
3. Leimbacher, P.-A. *et al.* MDC1 interacts with TOPBP1 to maintain chromosomal stability during mitosis. *Molecular cell* **74**, 571-583. e578 (2019).
4. Bagge, J., Oestergaard, V.H. & Lisby, M. in *Seminars in cell & developmental biology*, Vol. 113 57-64 (Elsevier, 2021).
5. Wardlaw, C.P., Carr, A.M. & Oliver, A.W. TopBP1: A BRCT-scaffold protein functioning in multiple cellular pathways. *DNA repair* **22**, 165-174 (2014).
6. Wyatt, H.D., Laister, R.C., Martin, S.R., Arrowsmith, C.H. & West, S.C. The SMX DNA repair tri-nuclease. *Molecular cell* **65**, 848-860. e811 (2017).
7. Osman, F. & Whitby, M.C. Exploring the roles of Mus81-Eme1/Mms4 at perturbed replication forks. *DNA repair* **6**, 1004-1017 (2007).
8. Mayle, R. *et al.* Mus81 and converging forks limit the mutagenicity of replication fork breakage. *Science* **349**, 742-747 (2015).
9. Bagge, J. *et al.* TopBP1 coordinates DNA repair synthesis in mitosis via recruitment of the nuclease scaffold SLX4. *Communications Biology* **8**, 1005 (2025).
10. Bhowmick, R., Minocherhomji, S. & Hickson, I.D. RAD52 facilitates mitotic DNA synthesis following replication stress. *Molecular cell* **64**, 1117-1126 (2016).
11. Minocherhomji, S. *et al.* Replication stress activates DNA repair synthesis in mitosis. *Nature* **528**, 286-290 (2015).
12. Ying, S. *et al.* MUS81 promotes common fragile site expression. *Nature cell biology* **15**, 1001-1007 (2013).
13. Leung, W. *et al.* FANCD2-dependent mitotic DNA synthesis relies on PCNA K164 ubiquitination. *Cell reports* **42** (2023).
14. Traband, E.L. *et al.* Mitotic DNA synthesis in untransformed human cells preserves common fragile site stability via a FANCD2-driven mechanism that requires HELQ. *Journal of Molecular Biology* **435**, 168294 (2023).
15. Madireddy, A. *et al.* FANCD2 facilitates replication through common fragile sites. *Molecular cell* **64**, 388-404 (2016).
16. Chan, K.L., Palmai-Pallag, T., Ying, S. & Hickson, I.D. Replication stress induces sister-chromatid bridging at fragile site loci in mitosis. *Nature cell biology* **11**, 753-760 (2009).
17. Gelot, C. *et al.* Polθ is phosphorylated by PLK1 to repair double-strand breaks in mitosis. *Nature* **621**, 415-422 (2023).
18. Brambati, A. *et al.* RHINO directs MMEJ to repair DNA breaks in mitosis. *Science* **381**, 653-660 (2023).
19. Neef, R. *et al.* Choice of Plk1 docking partners during mitosis and cytokinesis is controlled by the activation state of Cdk1. *Nature cell biology* **9**, 436-444 (2007).
20. Parrilla, A. *et al.* Mitotic entry: the interplay between Cdk1, Plk1 and Bora. *Cell Cycle* **15**, 3177-3182 (2016).
21. Colicino, E.G. & Hehny, H. Regulating a key mitotic regulator, polo-like kinase 1 (PLK1). *Cytoskeleton* **75**, 481-494 (2018).
22. Adam, S. *et al.* The CIP2A-TOPBP1 axis safeguards chromosome stability and is a synthetic lethal target for BRCA-mutated cancer. *Nature cancer* **2**, 1357-1371 (2021).

23. De Marco Zompit, M. *et al.* The CIP2A-TOPBP1 complex safeguards chromosomal stability during mitosis. *Nature Communications* **13**, 4143 (2022).
24. Clay, D.E., Bretscher, H.S., Jezuit, E.A., Bush, K.B. & Fox, D.T. Persistent DNA damage signaling and DNA polymerase theta promote broken chromosome segregation. *Journal of Cell Biology* **220**, e202106116 (2021).
25. Nagelli, S. & Westermarck, J. CIP2A coordinates phosphosignaling, mitosis, and the DNA damage response. *Trends in Cancer* (2024).

Rebuttal figure legend

Fig.R1

(A) Dot plot illustrating mean \log_2 fold change (FC) of proteins detected in TOPBP1 Co-IP samples from M phase (M) versus S phase cells (S), by-MS label-free mass-spectrometry demonstrating the enrichment of the TOPBP1-RAD9A interaction. (A) Representative images and violin plot showing the number of RAD9A and TOPBP1 colocalising foci in RPE1 p53^{-/-} FRT/TR treated without or with 400 nM aphidicolin for 18 hours followed by synchronisation with 60 ng/ml nocodazole for 2 hours (from two individual experiments, statistical significance was determined with Mann-Whitney test). (C) Proliferation analysis using the incucyte SX5 live cell imaging system, of RPE1 p53^{-/-} FRT/TR BRCA1^{-/-} cells treated with siCTRL, siSLX4 or siCIP2A followed by treatment with DMSO or 5 μ M ART558 (Pol θ i). (D). Western blot analysis of eGFP-TOPBP1 WT and K71/72E mutant co-immunoprecipitation samples from HEK293TN cells synchronised with 100 ng/ml Nocodazole for 18 hours. (E) Western blot analysis of eGFP-TOPBP1 WT and BRCT0-2 and BRCT0 C-terminal truncation mutant co-immunoprecipitation samples from HEK293TN cells synchronised with 100 ng/ml Nocodazole for 18 hours.

Figure R1

Fig.R2

(A) Bar plot showing the percentage of MUS81/TOPBP1 colocalising foci in DLD1 WT and BRCA2^{-/-} cells synchronised with 60 ng/ml nocodazole for 2 hours (from three individual experiments, absence of statistical significance was determined by two-tailed unpaired t-test). (B) Bar plot showing the percentage of ERCC1/TOPBP1 colocalising foci in DLD1 WT and BRCA2^{-/-} cells synchronised with 60 ng/ml nocodazole for 2 hours (from three individual experiments, absence of statistical significance was determined by two-tailed unpaired t-test). (C) Bar plot showing the percentage of MUS81 foci in DLD1 WT and BRCA2^{-/-} cells treated with siCTRL, siTOPBP1, siSLX4 synchronised with 60 ng/ml nocodazole for 2 hours (from three individual experiments, absence of statistical significance was determined by two-tailed unpaired t-test). (D) Bar plot showing the percentage of prometaphase nuclei with TOPBP1 foci in DLD1 BRCA2^{-/-} cells treated with siCTRL, siSLX4 synchronised with 60 ng/ml nocodazole for 2 hours (from three individual experiments, absence of statistical significance was determined by two-tailed unpaired t-test). (E) Bar plot showing the percentage of prometaphase nuclei with eGFP-SLX4 and CIP2A colocalising foci in RPE1 p53^{-/-} FRT/TR SLX4^{-/-} +eGFP-SLX4 WT or T1260A cells treated without or with 400 nM aphidicolin for 18 hours followed by synchronisation with 60 ng/ml nocodazole for 2 hours (from three individual experiments, statistical significance was determined by two-tailed unpaired t-test). (F) Bar plot showing the percentage of prometaphase nuclei with MUS81 and CIP2A colocalising foci in RPE1 p53^{-/-} FRT/TR WT or SLX4 T1260A knock in cells treated without or with 400 nM aphidicolin for 18 hours followed by synchronisation with 60 ng/ml nocodazole for 2 hours (from three individual experiments, statistical significance was determined by two-tailed unpaired t-test). (G). Bar plot showing the percentage of prometaphase nuclei with eGFP-SLX4 and TOPBP1 colocalising foci in RPE1 p53^{-/-} FRT/TR SLX4^{-/-} +eGFP-SLX4 WT cells treated with siCTRL or siCIP2A without or with 400 nM aphidicolin for 18 hours followed by synchronisation with 60 ng/ml nocodazole for 2 hours (from three individual experiments, statistical significance was determined by two-tailed unpaired t-test). (H). Bar plot showing the percentage of prometaphase nuclei with EdU foci in DLD1 BRCA2^{-/-} treated with siCTRL or siCIP2A followed by synchronisation with 60 ng/ml nocodazole for 2 hours (from three individual experiments, statistical significance was determined by two-tailed unpaired t-test). (I) Bar plot showing the percentage of prometaphase nuclei with eGFP-Polθ foci in DLD1 BRCA2^{-/-} +eGFP-Pol cells after induction with 100 ng/ml doxycycline for 24 hours, 48 hours after either siCTRL or siCIP2A siRNA treatment followed by synchronisation with 60 ng/ml nocodazole for 2 hours (from three individual experiments, statistical significance was determined by two-tailed unpaired t-test).

Figure R2

Fig.R3

(A) Images of eGFP-MUS81 and TOPBP1 colocalising foci in prometaphase nucleus in DLD1 WT cells after induction with 100 ng/ml doxycycline for 24 hours, demonstrating colocalisation between eGFP-MUS81 and TOPBP1

(B). Violin plot showing the mean bounding height of CIP2A foci per prometaphase nucleus of RPE1 p53^{-/-} FRT/TR SLX4^{-/-} +eGFP-SLX4 cells after induction of expression with 10 ng/ml doxycycline for 24 hours (from three individual experiments, statistical significance was determined by Mann-Whitney test). (C) Violin plot showing the mean bounding width of CIP2A foci per prometaphase nucleus of RPE1 p53^{-/-} FRT/TR SLX4^{-/-} +eGFP-SLX4 cells after induction of expression with 10 ng/ml doxycycline for 24 hours (from three individual experiments, statistical significance was determined by Mann-Whitney test).

a.

b.

c.